# Gut microbiota regulates acute myeloid leukaemia via alteration of intestinal barrier function mediated by butyrate

Ruiqing Wang [1], Xinyu Yang [1], Jinting Liu [1], Fang Zhong [2], Chen Zhang [1], Yuhong Chen [1], Tao Sun [1], Chunyan Ji [1] & Daoxin Ma [1]✉

The gut microbiota has been linked to many cancers, yet its role in acute myeloid leukaemia (AML) progression remains unclear. Here, we show decreased diversity in the gut microbiota of AML patients or murine models. Gut microbiota dysbiosis induced by antibiotic treatment accelerates murine AML progression while faecal microbiota transplantation reverses this process. Butyrate produced by the gut microbiota (especially *Faecalibacterium*) significantly decreases in faeces of AML patients, while gavage with butyrate or *Faecalibacterium* postpones murine AML progression. Furthermore, we find the intestinal barrier is damaged in mice with AML, which accelerates lipopolysaccharide (LPS) leakage into the blood. The increased LPS exacerbates leukaemia progression in vitro and in vivo. Butyrate can repair intestinal barrier damage and inhibit LPS absorption in AML mice. Collectively, we demonstrate that the gut microbiota promotes AML progression in a metabolite-dependent manner and that targeting the gut microbiota might provide a therapeutic option for AML.

[1] Department of Hematology, Qilu Hospital of Shandong University, Jinan, Shandong 250012, P. R. China. [2] Department of Endocrinology and Metabolism, Shandong Provincial Hospital affiliated with Shandong University, Jinan, Shandong 250012, P. R. China. ✉email: daoxinma@sdu.edu.cn

Acute myeloid leukaemia (AML) is a haematological malignancy characterised by excessive proliferation of immature myeloid cells. Although chemotherapy has been proven to be effective against this malignancy[1], long-term survival is modest[2]. The long-term survival and prognosis of AML are closely related to many factors in AML progression. Thus, improved outcomes may depend on novel treatment strategies derived from a better understanding of the mechanism of AML progression.

The progression of AML is governed by many factors, including genetic and homoeostatic environmental changes. Accumulating evidence suggests that the human gut microbiota can affect the homoeostatic environment and is linked to the development of inflammation, autoimmune diseases, and even cancers[3–6]. With further research, the relationship between gut microbiota and haematological diseases has gradually been reported. In a typical and stable environment, the gut microbiota regulates and sustains normal steady-state haematopoiesis[7]. Recently, it was reported that structural changes in the gut microbiota can occur in acute lymphoblastic leukaemia patients and that the microbiota diversity is significantly reduced[8]. Shifts in the intestinal microbiota following allogeneic haematopoietic cell transplantation were found to be associated with the development of gastrointestinal graft-versus-host disease (GVHD), suggesting that the intestinal microbiota may be an essential factor in GVHD[9]. However, the role played by the gut microbiota in AML progression has not been studied.

Some microbial metabolites are exclusively derived from the gut microbiota and are not host derived, which mediates the biological effects of the gut microbiota. Microbial metabolites such as short-chain fatty acids (SCFAs) have been implicated in haematological diseases. Payen et al. revealed a dramatic decrease in the levels of the main SCFAs, acetate, propionate, and butyrate, in severe GVHD patients[10]. In addition, butyrate, as an important SCFA[11–13], has been reported as a chief energy source for intestinal epithelial cells, and administration of butyrate can effectively alleviate GVHD[14–16]. This finding suggests that the gut microbiota may play an important role in diseases through metabolic pathways, especially in GVHD. However, related research on SCFAs in AML is still lacking.

The existence of the gut microbiota, along with its nutritious SCFA metabolites, can facilitate the growth of intestinal epithelial cells and regulate their differentiation and repair[17–19]. The intestinal barrier is the sole gate where entry of the gut microbiota or metabolites into the blood can be prevented. Its integrity and whether it can function normally are directly linked to fluctuations in the amounts of bacteria and their metabolites in the blood. Cui et al. found that the physical, chemical, immunological, and microbiological barriers in the intestinal tract constitute the complete intestinal barrier, which plays an important defensive role against the invasion of harmful substances from the intestines[20]. Another study confirmed that in haematological diseases, patients with GVHD showed significant incapacitation of gut barrier functionality compared with healthy volunteers[21]. The extent of integrity and permeability of the intestinal barrier often determine how the gut microbiota and their metabolites affect human systemic immunity[22]. The intestinal barrier tends to be disregarded in AML, and focusing on the intestinal barrier may provide novel insight for improving the therapeutic effect of AML.

In this study, we found that the gut microbes of AML patients and mice were significantly disordered and that gut microbiota dysbiosis further aggravated the progression of AML. SCFA metabolome analyses showed that the concentration of butyrate was significantly decreased in the intestinal contents of AML patients, which resulted in reduced intestinal barrier function and increased absorption of LPS into the blood. Moreover, the increased LPS content promoted AML progression, while the addition of intestinal butyrate protected intestinal epithelial cells and decreased AML severity. Our study on the relationship between AML and intestinal barrier function has confirmed that intestinal health is also worthy of attention in the process of AML progression, which might be useful for the diagnosis and treatment of AML.

## Results

**The diversity and composition of the gut microbiota are significantly altered in AML patients.** Sixty-one adults were enrolled from 2018 to 2020. Stool, peripheral blood, and bone marrow (BM) included in this study were obtained from 31 newly diagnosed (ND) and untreated AML patients and 30 healthy controls (Supplementary Table 1). A total of 2461 operational taxonomic units (OTUs) in all faecal samples of the 61 individuals were identified. We first confirmed that the sequencing depth of the faecal samples was sufficient for further analysis through the dilution curve method (Supplementary Fig. 1a). Then, we analysed the Shannon α-diversity indices of these samples. A significant difference in Shannon diversity was observed between AML patients and healthy controls ($P = 0.0275$), which demonstrated a significantly lower diversity in the faeces of AML patients than in those of healthy controls (Fig. 1a). Principal coordinate analysis (PCoA) demonstrated a significant difference in β-diversity using the weighted UniFrac distance between the AML and control groups ($P = 0.006$), suggesting that the microbial composition in AML patients was significantly shifted regarding OTUs (Fig. 1b). To identify the detailed alterations in the bacteria, we examined the relative abundances of these taxa. At the phylum level, *Firmicutes* was significantly depleted and *Bacteroidota* was enriched in AML patients compared with healthy controls. At the genus level, five bacterial genera displaying significantly different abundances between AML patients and healthy controls were observed. *Bacteroides* ($P = 0.03456$) was enriched, while *Faecalibacterium* ($P = 0.0005214$), *Roseburia* ($P = 0.0002254$), *Subdoligranulum* ($P = 0.00212$), and *Bifidobacterium* ($P = 0.02883$) were significantly depleted in AML patients (Fig. 1c and Supplementary Fig. 1b). Moreover, among these differential microbiotas, we observed a significantly positive correlation between the significantly different bacterial taxa *Faecalibacterium* and *Roseburia* ($P = 0.0009$) (Fig. 1d). The results of linear discriminant analysis effect size (LEfSe) further confirmed that both *Faecalibacterium* (LDA = 4.538) and *Roseburia* (LDA = 4.107) were lower in the faeces of AML patients compared to the control group (Fig. 1e). To further determine the relationship between bacteria and disease, we analysed the correlation between the abundances of *Faecalibacterium* and *Roseburia* and the clinical characteristics of AML patients. The association and correlation between *Faecalibacterium* and WBC were determined by Spearman correlation. Our results demonstrated that *Faecalibacterium* was significantly negatively correlated with the level of peripheral white blood cells using Spearman correlation and multiple linear regression. (Table 1 and Supplementary Table 3), while *Faecalibacterium* was also negatively correlated with the percentage of BM blast cells using Spearman correlation in AML patients (Table 1). Importantly, the abundance of *Faecalibacterium* and the number of OTUs were much higher in favourable-risk AML patients (Fig. 1f and Table 2). To confirm the consistency of the degree of risk and the patient's final results, the achievement of complete remission (CR) after the first therapy was used as the outcome index. The results showed that the CR rate of the patients with favourable risk was higher than that of the patients with unfavourable risk

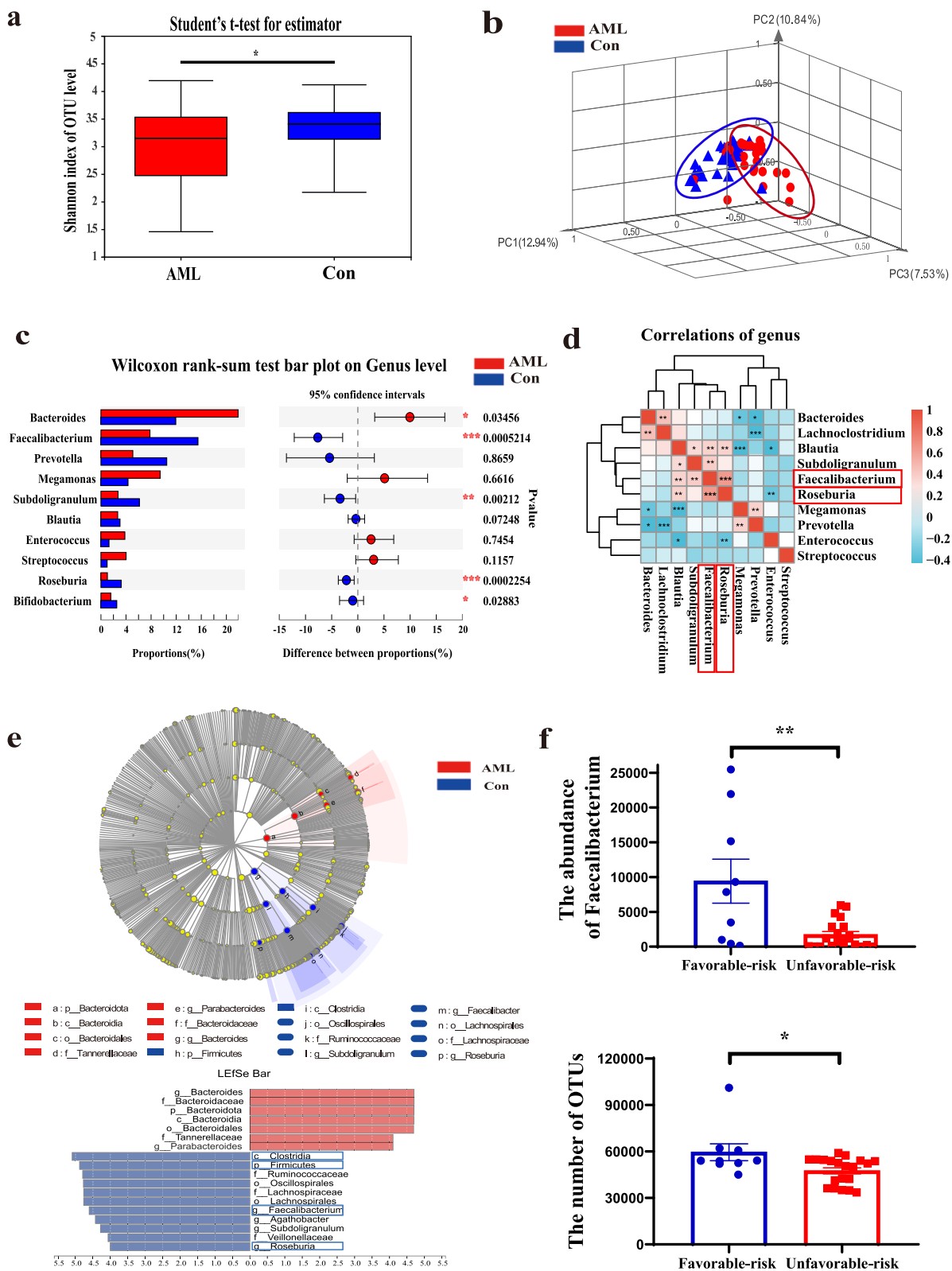

($P = 0.015$) (Supplementary Table 3). In addition, we analysed the data with a more sophisticated analysis taking into account potential bias, such as age and sex, which might be confounding factors for the gut microbiota results. When taking into account age and gender, the microbiota diversity in AML was still decreased (Supplementary Fig. 6c, d). These results suggested that the low diversity of gut microbiota and the reduced abundance of

*Faecalibacterium* might play an important role in the progression of AML.

**AML causes bacterial dysbiosis, and gut microbiota dysbiosis aggravates the progression of AML.** To clarify the role of the microflora in AML, we established MLL-AF9 AML mice and performed 16S ribosomal RNA (rRNA) sequencing of mouse

**Fig. 1 The diversity and composition of the gut microbiota are significantly altered in AML patients.** Total bacterial DNA was isolated from the intestinal content, and 16S rRNA genes were sequenced. **a** The diversity and richness of the gut microbiota in AML patients (AML) and healthy controls (Con). Unpaired $t$-tests were used to compare the Shannon index ($n = 61$). Data were presented as standard boxplots (with the box encompassing Q1–Q3, the median denoted as a central horizontal line in the box, and the whiskers covering the data within ±1.5 IQR). **b** PCoA of a weighted UniFrac distance analysis ($n = 61$). **c** Relative taxon abundance comparison among the AML and control groups ($n = 61$). **d** Spearman correlations between the intestinal content of the 10 genera in AML patients and healthy controls. *Faecalibacterium* and *Roseburia* were significantly correlated (red positive correlation, blue negative correlation). **e** Cladogram generated from linear discriminant analysis effect size (LEfSe) and the LDA score. **f** The abundance of *Faecalibacterium* and OTUs was reduced in the unfavourable-risk group ($n = 20$) compared with the favourable-risk group ($n = 9$). $P$ values were determined using two-tailed $t$-test in (**a**, **f**) and using Wilcoxon rank test in **c**. Error bars represent mean ± SEM in (**a**, **c**, **f**). *$P = 0.02178$ (**a**), **$P = 0.0016$, *$P = 0.015$ (**f**). Source data are provided as a Source Data file.

**Table 1 Correlation analysis between differential gut microbiota and WBC, bone marrow blast cells.**

| Variables | Correlations | | | |
| --- | --- | --- | --- | --- |
| | WBC count | | Percentage of bone marrow blast cells | |
| | *r* | *P* | *r* | *P* |
| *Faecalibacterium* | −0.493 | 0.007 | −0.412 | 0.027 |
| *Roseburia* | −0.452 | 0.014 | −0.248 | 0.195 |

*r*: correlation coefficient.
Italic: bacterial species name.
Correlations were identified with Spearman's rank correlation test.
A two-tailed *P* value <0.05 was considered statistically significant.

**Table 2 Association between different genera and unfavourable prognosis.**

| Variables | Parameter estimates | | | |
| --- | --- | --- | --- | --- |
| | *B* | *P* | Exp(*B*) | 95% confidence interval for Exp(*B*) |
| *Faecalibacterium* | −0.543 | 0.04 | 0.581 | 0.345–0.976 |
| *Roseburia* | −0.13 | 0.93 | 0.987 | 0.737–1.322 |
| OTUs | −0.14 | 0.058 | 0.986 | 0.972–1.00 |

The reference category is: unfavourable; *B*: regression coefficient; Exp(*B*): the exponent of *B*.
Italic: bacterial species name.
The risk classification standard is listed in Supplementary Table 2.
Correlations were identified with Binary logistic regression.
A two-tailed *P* value <0.05 was considered statistically significant.

stool. Our results showed that the gut microbiota diversity of AML mice decreased significantly with the progression of disease compared with that of healthy control mice (Fig. 2a). Consistently, we obtained similar results in another two AML murine models (AML1-ETO murine models and C1498) (Supplementary Fig. 2c, d). To further verify whether the disordered gut microbes could affect disease progression, we first successfully ablated the gut microbiota in mice after administration of antibiotics (Supplementary Fig. 2a) and then established MLL-AF9 AML mice and evaluated their AML progression (Fig. 2b). After sacrificing the mice, the FACS results showed that the percentage of GFP$^+$ MLL-AF9 leukaemia cells in the BM, spleen, or peripheral blood was much higher in antibiotic-treated AML mice than in control PBS AML mice (Fig. 2c and Supplementary Fig. 2e). Furthermore, we used haematoxylin and eosin (HE) staining and Ki67 immunohistochemical staining methods to evaluate the infiltration and proliferation of leukaemia cells in the spleen. As expected, leukaemia cell infiltration and proliferation were aggravated in antibiotic-treated AML mice (Fig. 2d). Importantly, antibiotic-treated AML mice had a significantly shorter lifespan than PBS-treated control AML mice [median 15 (range 12–17)

days vs median 19 (range 18–21) days; $P = 0.0328$] (Fig. 2e). Furthermore, to verify the influence of normal murine intestinal flora on the progression of AML in vivo, we resumed and preserved the wild-type (WT) gut microbiota of AML mice via oral gavage of faecal suspension from normal mice, while oral gavage of PBS served as a control. Next, we determined the murine leukaemia burden. Our results showed that the frequencies of GFP+ AML cells in peripheral blood, BM, and spleen of AML mice gavaged with normal mouse faecal suspension were decreased significantly compared with AML mice gavaged with the PBS control (Fig. 2g). The decreased size and weight of the spleen and mitigated leukaemia cell infiltration in AML mice gavaged with the normal mouse faecal suspension were also verified (Fig. 2f, h). These data suggested that the preservation of WT microflora could effectively delay the progression of AML in the mouse model. Altogether, these results demonstrated that AML caused dysbacteriosis and that destroying the diversity of the gut microbiota accelerated AML progression.

**FMT delays the development of AML.** As shown above, *Faecalibacterium* correlated with a favourable prognosis and might be considered a beneficial bacteria in AML patients. At the species level, the abundance of *F. prausnitzii* was TOP 1 in control, and significantly decreased in AML (Supplementary Fig. 2b). Therefore, we used *F. prausnitzii* to study the effect of *Faecalibacterium* in butyrate generation and on AML. We studied the function of *Faecalibacterium* through bacterial transplantation. The results showed that *Faecalibacterium* gavage obviously relieved splenomegaly (Supplementary Fig. 3a, b). In terms of tumour load, we observed a significant reduction in the BM and a reduced tendency in the spleen and peripheral blood (Supplementary Fig. 3c). The overall survival of AML mice treated with *Faecalibacterium* gavage was not significantly different from that of control AML mice (Supplementary Fig. 3d). Moreover, to further address the cause-and-effect relationship between *Faecalibacterium* and AML, we performed an faecal microbiota transplantation (FMT) experiment with faeces from AML patients supplemented with or without *Faecalibacterium*. Our results showed that spleen sizes and the leukaemia cell frequencies in the BM and spleen were significantly reduced in AML-FMT mice supplemented with *Faecalibacterium* compared with AML-FMT mice without *Faecalibacterium*. The leukaemia cell frequencies in peripheral blood also showed a decreasing trend but did not reach a significant difference (Supplementary Fig. 3e–g). These results demonstrated that *Faecalibacterium* could decrease the leukaemia burden in AML mice, but the inhibiting effect of a single *Faecalibacterium* supplement was weaker than the transplantation of intact gut microbiota. The full play of *Faecalibacterium* inhibitory effect on AML relies on intact gut microbiota.

Next, in order to observe the influence of the intact gut microbiota on the progression of AML, FMT was performed using the complete stool of AML patients or healthy controls to clarify the causal relationship between the gut microbiota

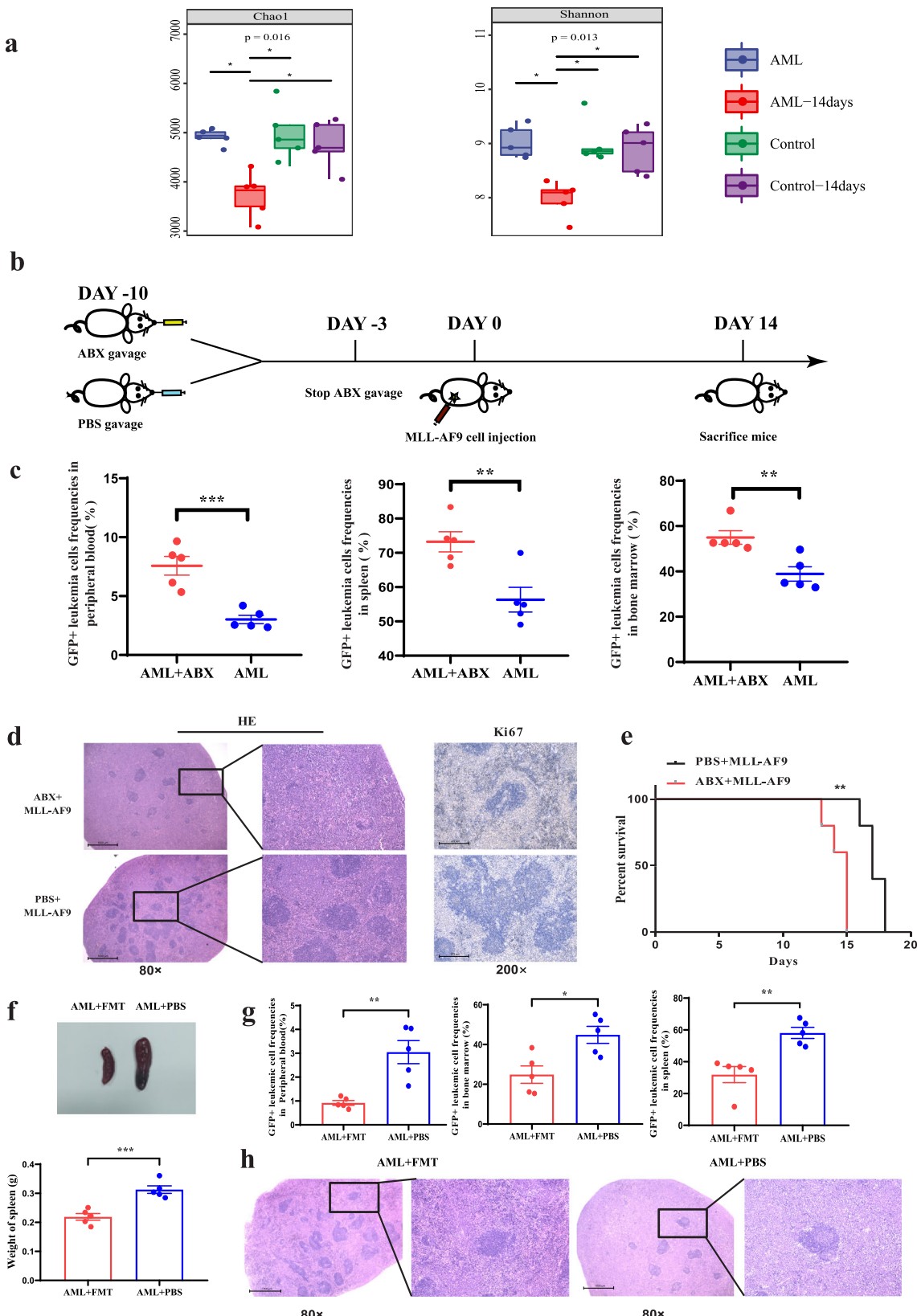

imbalance and AML progression. The experimental procedure is shown in Fig. 3a. PCoA showed that the sample points representing the gut microbiota from the donor individuals and the corresponding recipient mice almost completely overlapped, indicating that bacterial colonisation in recipient mice was successful after FMT (Supplementary Fig. 4a). Our results

demonstrated that the leukaemia load in the spleen, BM, and peripheral blood of AML-FMT mice (antibiotic-treated AML mice followed by FMT with faecal hydration liquid from AML patients) increased significantly compared with that in the spleen, BM, and peripheral blood of Con-FMT mice (antibiotic-treated AML mice followed by FMT with faecal hydration liquid from

**Fig. 2 AML causes bacterial dysbiosis, and gut microbiota dysbiosis aggravates the progression of AML. a** The diversity and richness of the gut microbiota in control mice (without cell injection), control-14days mice (without cell injection after 14 days), AML mice (AML cell injection) and AML-14days (mice after AML cell injection after 14days). Unpaired *t*-test were used for comparing the Chao1 and Shannon index (*n* = 5 per group). Data were presented as standard boxplots (with the box encompassing Q1–Q3, the median denoted as a central horizontal line in the box, and the whiskers covering the data within ±1.5 IQR). **b** Schematic diagram of the mouse experimental process. **c** Leukaemia cells (GFP+ cells) in the spleen, peripheral blood, and bone marrow from ABX AML mice (*n* = 5) and control PBS AML mice (*n* = 5). Details of the gating strategy are described in Supplementary Fig. 11b. **d** Haematoxylin and eosin–stained histopathology sections and Ki67 immunohistochemical staining of a representative spleen, the ABX AML group, and the PBS AML group. All microscopic analyses were performed at an original magnification of ×80 or ×200, scale bar = 1000 and 275 μm. **e** Kaplan–Meier survival curve of AML mice (*n* = 5 per group). **f** The photographs and weights of spleens from AML-FMT mice (*n* = 5) and AML-PBS mice (*n* = 5). **g** The leukaemia cells (GFP+ cells) in spleen, peripheral blood and bone marrow from AML-FMT mice (*n* = 5) and AML-PBS mice (*n* = 5). Details of the gating strategy are described in Supplementary Fig. 11b. **h** HE histopathology sections of a representative spleen, AML-FMT group and AML-PBS group. All microscopic analyses were performed (original magnification ×80 or ×200), scale bar = 1000 and 275 μm. *P* values were determined using Dunn's test in **a**. *P* values were determined using unpaired two-tailed *t*-test in **c**, **f**, **g**. *P* values were determined using Gehan–Breslow–Wilcoxon test in **e**. Error bars represent mean ± SEM in **a**, **c**, **f**, **g**, **e**. *\*P* = 0.016 Chao1, \**P* = 0.013 Shannon (**a**), \*\*\**P* = 0.0008 PB, \*\**P* = 0.0068 SP, \*\**P* = 0.0087 BM (**c**), \*\**P* = 0.0043 (**e**), \*\*\**P* = 0.0007 (**f**), \*\**P* = 0.0025 PB, \*\**P* = 0.0028 SP, \**P* = 0.0144 BM (**g**). Source data are provided as a Source Data file.

healthy people), as shown by the spleen weight, HE sections, Ki67 staining (Fig. 3b, d), and the percentage of GFP+ leukaemia cells (Fig. 3c and Supplementary Fig. 4b). To observe the dynamic changes in spleen leukaemia burden, a dual-luciferase assay demonstrated that the dual-luciferase activity in the AML-FMT group was significantly higher than that in the Con-FMT group, indicating that AML-FMT treatment aggravated AML progression in vivo (Fig. 3e). Moreover, AML-FMT mice had a significantly shorter lifespan than Con-FMT mice [median 17 (range 16–18) days vs median 21 (range 19–22) days (*P* = 0.0358)] (Fig. 3f). Similar results were also demonstrated in the FMT experiment with mouse faeces (Supplementary Fig. 9). Taken together, these results demonstrated that the transplantation of intact gut microbiota plays a greater role in inhibiting the progression of AML than the application of *Faecalibacterium* alone, and supplementation with *Faecalibacterium* decreased the leukaemia burden in AML-FMT mice.

**AML patients exhibit profound alterations in gut microbial metabolites.** Next, we studied the specific mechanism of the involvement of the gut microbiota in AML progression. As gut metabolites are important bridges for the gut microbiota to regulate disease progression, untargeted metabolomics analysis based on ultra-performance liquid chromatography was first performed to explore the alteration of intestinal metabolic profiles in AML patients. We collected paired stool, blood, and BM samples from each patient for ultra-performance liquid chromatography analysis. A total of 540 and 578 peak features were identified in positive ion mode (ES+) and negative ion mode (ES−), respectively. These peak features were clustered by orthogonal partial least squares discriminant analysis (OPLS-DA) and cross-validated scores plot, showing that the samples from the AML and control groups were clearly separated from each other (Fig. 4a). We further analysed the peak features using MS/MS and identified the top 25 abundant metabolites using the combination of precise molecular weight and structural information from the compound structure database. Of these, propionic acid and butyric acid were significantly different in the stool samples between the healthy and AML groups (Fig. 4b). Next, targeted SCFA metabolomics analysis was applied to quantitatively determine intestinal SCFAs by GC-MS, and the results showed that propionic acid (*P* = 0.0196) and butyric acid (*P* = 0.0065) levels were significantly decreased in the stools of AML patients compared with those in the stools of healthy controls (Fig. 4c). Importantly, correlation analyses of the gut microbiota and SCFA levels revealed that *Faecalibacterium* (*P* = 0.0279) was positively correlated with butyric acid (Fig. 4d). These data indicated that *Faecalibacterium* was most significantly downregulated in the gut

microbiota of AML patients, which demonstrated a close relationship with the metabolism of butyric acid in these hosts. To provide more insight into the role played by *Faecalibacterium* in butyrate generation, we cultured *Faecalibacterium* in vitro and determined the content of butyrate in the supernatant of *Faecalibacterium*. The results showed that the concentration of butyrate in the supernatant of *Faecalibacterium* was significantly higher than that in controls (Supplementary Fig. 5g). Then, we administered *Faecalibacterium* to mice via gavage and detected stool butyrate. Our results showed that the stool butyrate content was much higher in mice gavaged with *Faecalibacterium* than in controls (Supplementary Fig. 5h). Therefore, we confirmed that butyric acid was a direct metabolite of *Faecalibacterium* and that *Faecalibacterium* could cause an increase in butyrate. In addition, we found no significant difference in propionic acid or butyric acid concentration in peripheral blood or BM of AML patients compared with controls, which indicated that they might not affect AML cells directly in the blood or BM (Supplementary Fig. 5a, b). To identify the key point that could build a bridge between intestinal bacteria and blood, we predicted the functional changes in intestinal bacteria in AML with PICRUSt. The results showed that AML patients and healthy controls had significant differences and identified abundant bacterial functions in the COG database. For example, DNA-binding transcriptional and fatty acid biosynthesis (COG0416, COG1309) were decreased in AML patients (Fig. 4e). Further analysis of its downstream functions through the KEGG database revealed the obvious difference in the relative abundance of butanoate metabolism (Fig. 4f). The above evidence suggested that butyrate might play an important role in the progression of AML.

**Microbiota-derived butyrate gavage delays AML progression.** Next, to further explore the effect of butyrate on AML progression in vivo, we administered butyrate to mice via intra-gastric gavage every day starting 7 days prior to injection of AML cells and continued the administration of butyrate for another 14 days (Fig. 5a). Then, we determined the effect of butyrate on the leukaemia load and survival time in AML mice. We found that butyrate-treated AML mice presented with more alleviated splenomegaly than control AML mice (Fig. 5b). Moreover, the results showed that the percentage of GFP+ leukaemia cells in the BM, peripheral blood, or spleen of butyrate-treated AML mice was lower than that in the BM, peripheral blood, or spleen of the control AML mice (Fig. 5c and Supplementary Fig. 5c). Under the microscope, leukaemia cell infiltration and proliferation in the spleen were mitigated in butyrate-treated AML mice. In addition, the normal spleen anatomical structure, such as the white pulp, was better

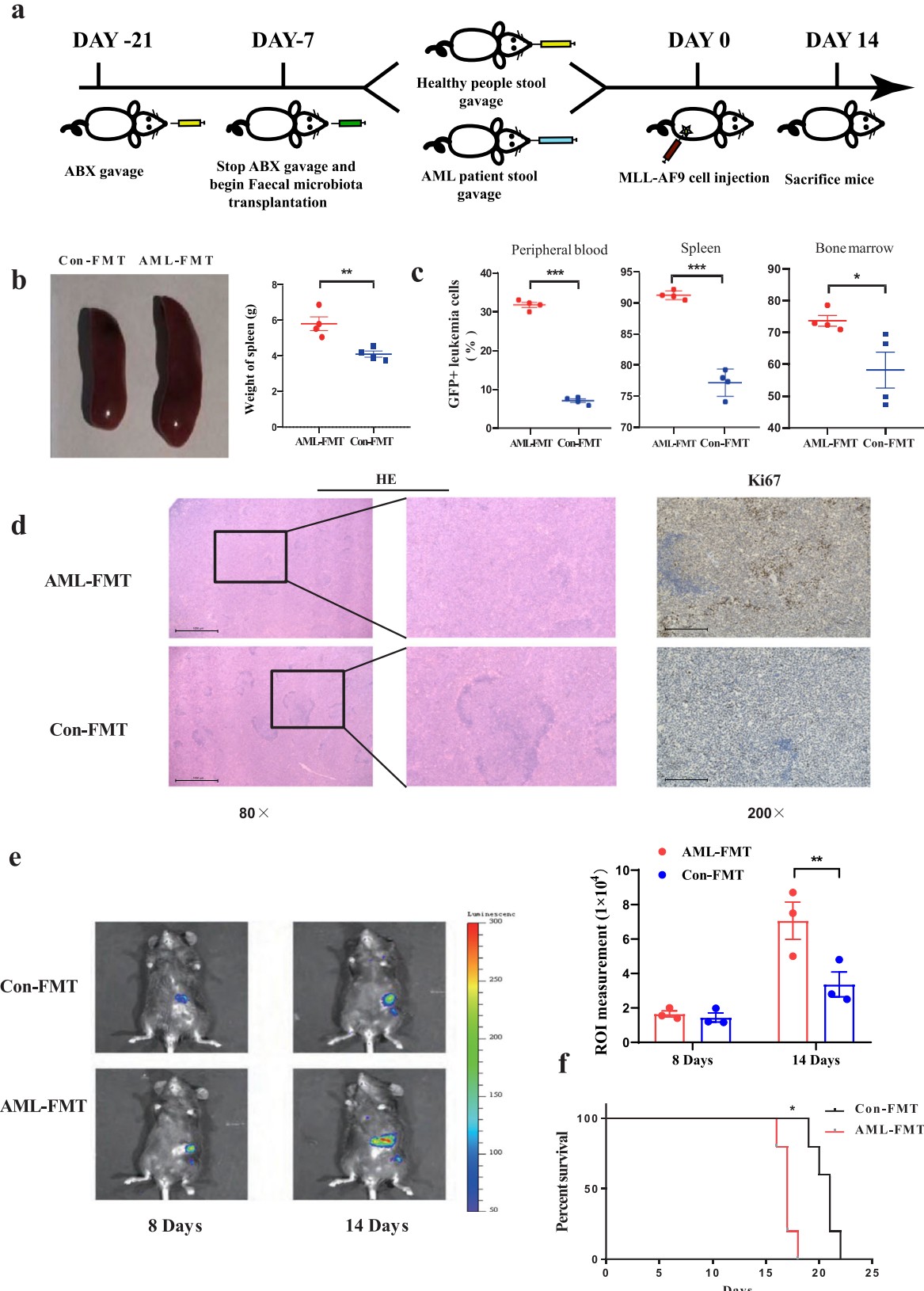

preserved in the butyrate-treated AML mice (Fig. 5d). To observe the dynamic changes in spleen leukaemia burden, a dual-luciferase assay demonstrated that the dual-luciferase activity in the butyrate administration group was significantly lower than that in the control, indicating that butyrate administration alleviated AML progression in vivo (Fig. 5f).

Moreover, butyrate administration in AML mice led to a significantly longer lifespan than that of control AML mice [median 20 (range 19–23) days vs median 16 (range 15–17) days ($P = 0.0479$)] (Fig. 5e). Moreover, consistent with the results in AML patients, the blood concentration of butyrate showed no obvious change in mice that received butyrate

**Fig. 3 FMT delays the development of AML. a** Schematic diagram of the mouse AML-FMT process. **b** Photographs and weights of spleens from AML-FMT mice ($n = 4$) and Con-FMT mice ($n = 4$). **c** Leukaemia cells (GFP+ cells) in the spleen, peripheral blood, and bone marrow from AML-FMT mice ($n = 4$) and Con-FMT mice ($n = 4$). Details of the gating strategy are described in Supplementary Fig. 11b. **d** HE histopathology sections and Ki67 immunohistochemical staining of a representative spleen, the AML-FMT group, and the Con-FMT group. All microscopic analyses were performed at an original magnification ×80 or ×200, scale bar = 1000 and 275 μm. **e** On days 8 and 14 after the injection, the load of Luciferase expressing MLL-AF9 cells in mice was analysed by IVIS ($n = 3$ per group). **f** Kaplan–Meier survival curve of AML mice ($n = 5$ per group). *P* values were determined using unpaired two-tailed *t*-test and error bars represent mean ± SEM in **b**, **c**, **e**. *P* values were determined using Gehan–Breslow–Wilcoxon test and error bars represent mean ± SEM in **f**. **P = 0.0069 (**b**), ***P < 0.0001 PB, ***P < 0.0001 SP, *P = 0.0114 BM (**c**), *P = 0.0473 (**e**), **P = 0.0039 (**f**). Source data are provided as a Source Data file.

administration (Supplementary Fig. 5d), which suggested that butyrate delayed AML progression not via direct contact to kill leukaemia cells but by other mechanisms.

**Butyrate reverses intestinal barrier damage in mice with AML.** After butyrate is produced in the intestinal tract, it is mainly absorbed by intestinal epithelial cells as an energy source, which promotes intestinal barrier integrity. Therefore, we tested the intestinal barrier function in AML mice and experimentally verified its relationship with AML progression. Our results showed that the colon length of AML mice was significantly shorter than that of the control group (Supplementary Fig. 5e), and intestinal permeability to FITC-dextran was significantly increased in AML mice compared with that in control mice, suggesting leukaemia-induced intestinal damage. Furthermore, we damaged the intestinal barrier by using low-dose and short-term DSS and the result showed that the AML progression in DSS treatment mice significantly accelerated than that in the control AML group (Supplementary Fig. 10). These results further validate that GI damage promotes AML progression. Next, we determined the effects of butyrate on intestinal integrity, which has not been studied in AML. The results show that butyrate-treated AML mice showed significantly ameliorated intestinal damage compared with the control AML mice and only mildly increased FITC-dextran permeability compared with normal mice (Fig. 6a). Next, we examined the intestinal histology damage. HE staining results showed that there was no apparent microstructure change except the intestinal recess became a little shallow in AML mice (Supplementary Fig. 6e). Ki67 expression was shown to decrease in AML mice, suggesting intestinal epithelial cells proliferation was inhibited in AML mice (Supplementary Fig. 6f). As for the cell–cell junction integrity, we utilised transmission electron microscopy to examine the ability of butyrate to preserve cellular junctions in AML mice. As expected, a significantly larger gap between intestinal epithelial cells was found in AML mice (Fig. 6d, middle panel). In contrast, the integrity of the intestinal epithelial cell junction was preserved in both normal (Fig. 6d, left panel) and butyrate-treated mice (Fig. 6d, right panel). Moreover, to explore the mechanism of the alteration in intestinal permeability, we evaluated the expression of ZO-1, TJP2, occludin, claudin-1, claudin-2, claudin-3, claudin-4, and claudin-8, which are important intestinal tight junction proteins (TJPs). The results demonstrated that the expression of the cell–cell junction protecting proteins ZO-1 and claudin-1 was downregulated while that of the cell–cell junction-inhibiting protein claudin-2 was upregulated in AML mice at both mRNA and protein levels, and these changes were reversed in butyrate-treated AML mice (Fig. 6b, c). In addition, we found that butyrate treatment significantly increased the mRNA expression of occludin and claudin-3 in intestinal epithelial cells, although no significant change was found between normal and AML mice. Western blot results also showed that occludin expression was increased significantly in butyrate-treated mice, which was consistent with the mRNA expression level (Supplementary Fig. 6a, b).

These findings suggested that the intestinal barrier damage caused by AML might not be due to occludin expression changes, but butyrate could protect the intestinal barrier by further increasing occludin expression. Next, we assessed the in-situ protein expression of claudin-1, claudin-2, and ZO-1 by immuno-fluorescence assays to evaluate the changes in intestinal epithelial tight junctions. The results showed that the expression levels of claudin-1 and ZO-1 were decreased in the intestine of AML mice compared with normal or butyrate-treated AML mice. In contrast, claudin-2 was enriched in the intestine of AML mice (Fig. 6e). Despite our findings, the detailed molecular mechanisms associated with damage or protection of the intestinal barrier by butyrate in AML requires further study.

**Damage to the intestinal barrier accelerates bacterial-derived LPS leakage into the blood, and LPS exacerbates the progression of AML.** The intestinal barrier is the key to protect the body from the harmful effects of gut microbiota, and impairment of its function may lead to the displacement of harmful intestinal substances into the circulatory system. LPS is the main harmful product of the gut microbiota, and the intestinal barrier is the only way for LPS to enter the blood. We used enzyme-linked immunosorbent assay (ELISA) to detect the LPS concentration in the plasma of peripheral blood, and the results showed that the LPS concentration in the plasma was significantly higher in the AML group than in the control group in humans (Fig. 7a). Moreover, we also determined the LPS concentration in the plasma of butyrate-treated AML and FMT-treated AML mice, and the results showed that butyrate and Con-FMT treatment significantly reduced the LPS concentration in mouse plasma (Fig. 7b, c). In addition, to confirm that the increase in the LPS concentration was due to a decrease in barrier function, LPS gavage was administered to mice. The results showed that the concentration of LPS in the peripheral blood was significantly higher in AML mice than in normal mice at the same dose of LPS (Fig. 7d). Furthermore, we determined the serum LPS concentration throughout the whole antibiotics-gavaging AML murine experiment. The result demonstrated that LPS levels in antibiotic mix (ABX)-gavaging AML mice were significantly higher than those of control AML mice during the whole experiment, which provides better support for the role of LPS in AML progression. (Supplementary Fig. 7). Therefore, these results confirmed that the concentration of bacterial-derived LPS was negatively correlated with intestinal barrier function, that a damaged intestinal barrier resulted in enhanced bacteria-derived LPS leakage into blood, and that butyrate treatment and Con-FMT treatment reversed this effect.

To elucidate the biological role of LPS in leukaemia cells in vitro, CCK-8 was used to evaluate cell proliferation, and flow cytometry analysis via the Annexin V-FITC/PI staining method was performed to analyse apoptosis. Our results demonstrated that LPS obviously decreased cell apoptosis (Fig. 7e) and promoted the proliferation of MLL-AF9 cells (Fig. 7f). We further determined the related proteins by western blot and found

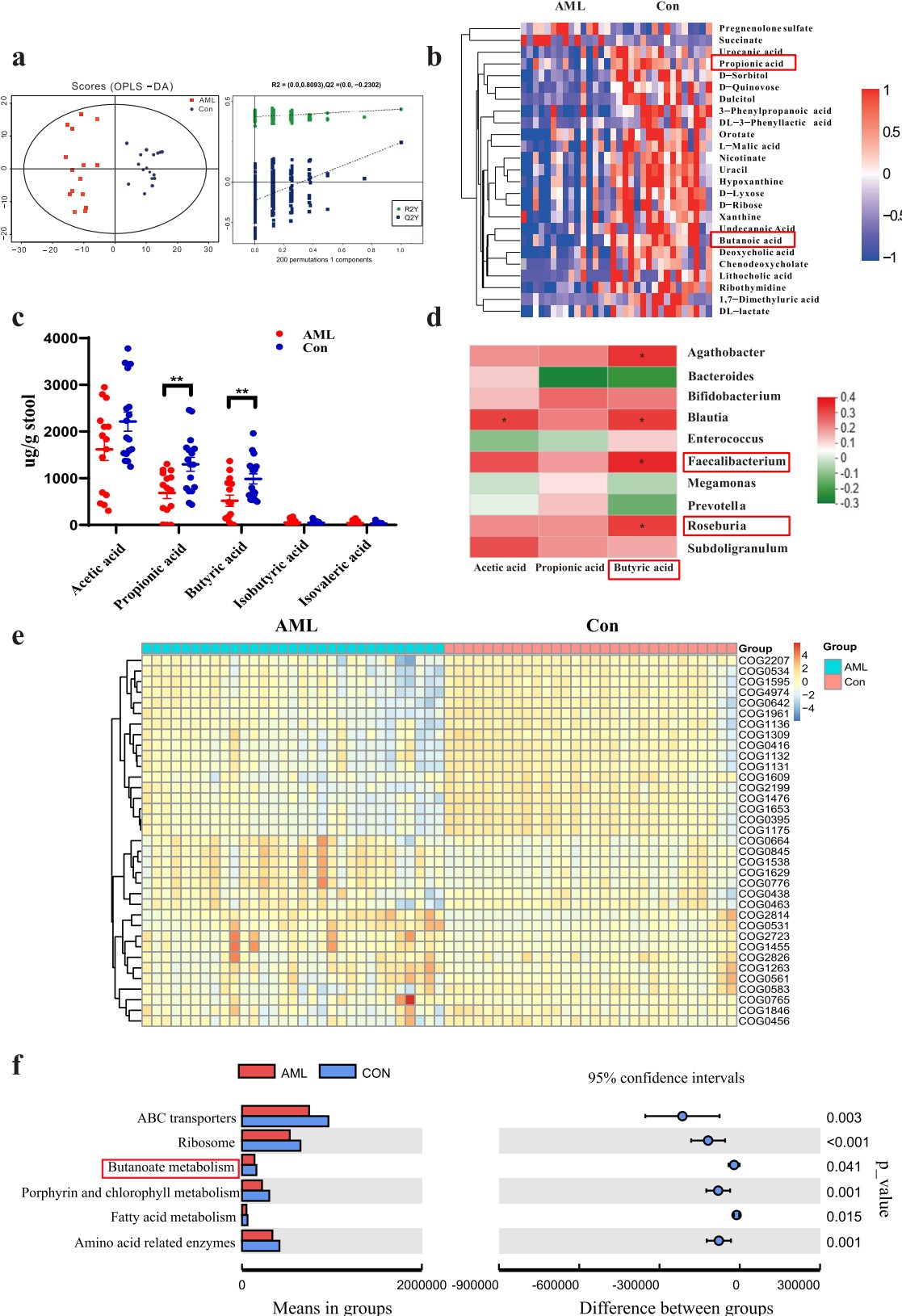

that LPS stimulation significantly upregulated the protein expression of Bcl-2 and downregulated that of BAX and cleaved caspase-3 proteins (Fig. 7g). To explore the possible promoting effects of LPS in vivo, LPS administration was initiated via vein injection into mice (Supplementary Fig. 5f). Our results showed that the LPS-treated AML mice presented with more severe

splenomegaly than the control AML mice, and the percentages of GFP⁺ leukaemia cells in the peripheral blood, spleen, and BM were higher in LPS-treated than in control AML mice (Fig. 7h, i). Through microscopy, leukaemia cell infiltration into the spleen was observed to show an aggravated tendency in LPS-treated compared with control AML mice (Fig. 7j). Similarly, LPS-treated

**Fig. 4 AML patients exhibit profound alterations in gut microbial metabolites. a** Plot of the OPLS-DA score and permutation tests for the OPLS-DA score plots from the untargeted metabolomics analysis of stool samples from AML patients ($n = 15$) and healthy controls ($n = 17$). **b** Heatmap of the relative abundances of the top 25 most abundant metabolites that significantly changed in the AML group. The colour bar indicates the Z score, which represents the relative abundance. A Z score $<0$ ($>0$) indicated that the relative abundance was lower (higher) than the mean. **c** The concentrations of propionic acid, butyric acid, and acetic acid in faecal samples of AML patients ($n = 15$) and controls ($n = 17$) were determined by GC-MS. **d** Heatmap of the Spearman correlation analysis between the gut microbiota and the metabolite. **e** The functional abundance distribution histogram of samples from AML patients and healthy controls in the COG database using PICRUSt software (the top 35 samples were selected by the maximum sorting method). **f** Significant differences in metagenomic functions in AML patients compared with healthy controls (the mean is used to measure the centre of the error bar, corrected $P < 0.05$ and confidence intervals $= 95\%$, $n = 31$ AML patients and $n = 30$ healthy control, biologically independent samples). $P$ values were determined using unpaired two-tailed $t$-test and error bars represent mean ± SEM in **c**. \*\*$P = 0.001423$ AML vs Control Propionic acid, \*\*$P = 0.00673$ AML vs Control Propionic Butyric acid (**c**). Source data are provided as a Source Data file.

AML mice had a significantly shorter lifespan than control AML mice [median 14 (range 13–16) days vs median 18 (range 16–20) days ($P = 0.0408$)] (Fig. 7k), which suggested that modulation of the LPS content could be effective for AML treatment. To validate the generalisability of our findings, we used the AML1-ETO murine model to perform several key functional experiments, including antibiotics treatment, butyrate supplement or LPS injection, and all these results are consistent with those from MLL-AF9 murine model (Supplementary Fig. 8).

## Discussion

The gut microbiota and its metabolism have been reported to be associated with many cancers[23–25]. It has been reported that an imbalance in gut microbial diversity can accelerate the development of ovarian cancer[26], while a diversified gut microbiota is beneficial for the prognosis and chemotherapy sensitivity of colon cancer[27]. As the most important haematological tumour, increasing attention has been given to the relationship between leukaemia and gut microbiota. Although some progress was made recently regarding the gut microbiota in several leukaemia types[28,29], it remains unclear whether gut microbes are mechanistically involved in AML progression. In this study, we found that AML could induce intestinal barrier damage, which is associated with altered gut microbiota and SCFAs in patients. The injured intestinal barrier and increased permeability in AML further promote the release of LPS into the blood circulation, which accelerates AML cell proliferation and disease progression. Our results might provide an insight into developing new targets for AML treatment.

We found significant alterations in gut microbiota diversity in AML, and this change was related to the risk stratification of the disease. This finding indicated that while AML caused an imbalance in the gut microbiota, the disordered microbiota also affected the progression of the disease. To better serve clinical treatment, it is particularly important to study the mechanism of the flora in the progression of AML. Blocking the positive feedback between AML and bacterial dysbiosis mechanism will help to solve the unexplained infection in patients with AML and improve the clinical treatment effect. Regarding the species of bacteria, the abundance of *Faecalibacterium* decreased significantly in treatment-naive AML patients. *Faecalibacterium* is the best-characterised microbiota constituent that is involved in the immune microenvironment[30,31]. According to reports, the role of *Faecalibacterium* in tumours is gradually becoming appreciated. *Faecalibacterium* also suppresses the proliferation and promotes the apoptosis of breast cancer cells, although these effects disappear after adding recombinant human IL-6[32]. However, in our study, after administration of *Faecalibacterium*, AML mice showed no significant difference in survival or leukaemia infiltration, which indicated that the antitumour effect of *Faecalibacterium* was limited. Therefore, we further performed FMT in AML mice and found that AML-FMT aggravated the progression

of AML compared with that in Con-FMT mice. A recent study by Yu et al.[33] reported that the bacterial diversity was increased and no depletion of *Faecalibacterium* was found in AML, which is different from our results. These differences may be due to the regional and gender diversities which have been reported by other studies[34–37].

A large number of reports have shown that most SCFAs in the human body, including butyrate, are derived from the metabolism of intestinal bacteria[38,39], and it is well known that butyrate is a common metabolite of *Faecalibacterium* and *Roseburia*[40]. In our experiment, AML patients had a significantly lower faecal butyric acid content than healthy people, but no significant difference was found in the peripheral blood and BM. Consistent with our results, other reports have shown that butyric acid is mainly metabolised in the intestinal tract, and the change in intestinal concentration cannot cause corresponding changes in peripheral blood[41]. Therefore, the effect of intestinal butyric acid on AML did not occur via direct contact to kill leukaemia cells in the blood or BM. Conversely, butyric acid, as the main energy source of intestinal epithelial cells, plays an important role in the regulation and function of the intestinal barrier[42–44], and its deficiency could lead to disruption of the gut barrier and cause multiple diseases[45,46]. Zheng et al. showed that butyrate has received particular attention for its beneficial effects on intestinal homoeostasis and energy metabolism. With anti-inflammatory properties, butyrate enhances intestinal barrier functions and mucosal immunity[47]. Butyrate plays key roles not only in the local intestinal barrier but also at the systemic level. At both sites, the mode of action is through modulating signalling pathways involving nuclear NF-kB and inhibition of histone deacetylase[48]. Although it has been reported that chemotherapy drugs can affect intestinal barrier function in AML patients[49] and blast crisis chronic myeloid leukaemia murine model has intestinal barrier function alteration[50], its role and mechanism in AML progression have not been reported. Our results showed that the intestinal barrier function of AML mice was decreased and that the expression of TJPs (claudin-1 and ZO-1) in intestinal epithelial cells was decreased and that of claudin-2 was increased in AML mice. Furthermore, administration of butyrate to AML mice effectively reversed intestinal barrier damage. Here, our research confirmed the role of butyrate in protecting the intestinal barrier in the progression of AML. This finding suggests that supplementation with butyrate may be of great significance in AML treatment or post-chemotherapy care. Interestingly, in addition to our research, G protein-coupled receptors for butyrate and their role in modulating the immune system have been reported, which is worthy of further study.

As the main component of intestinal bacteria, LPS affects the immune microenvironment in the human body[51]. The intestinal barrier is the main way through which LPS can enter blood circulation. In this study, we found that intestinal barrier damage in primary AML patients caused more LPS to enter the blood and

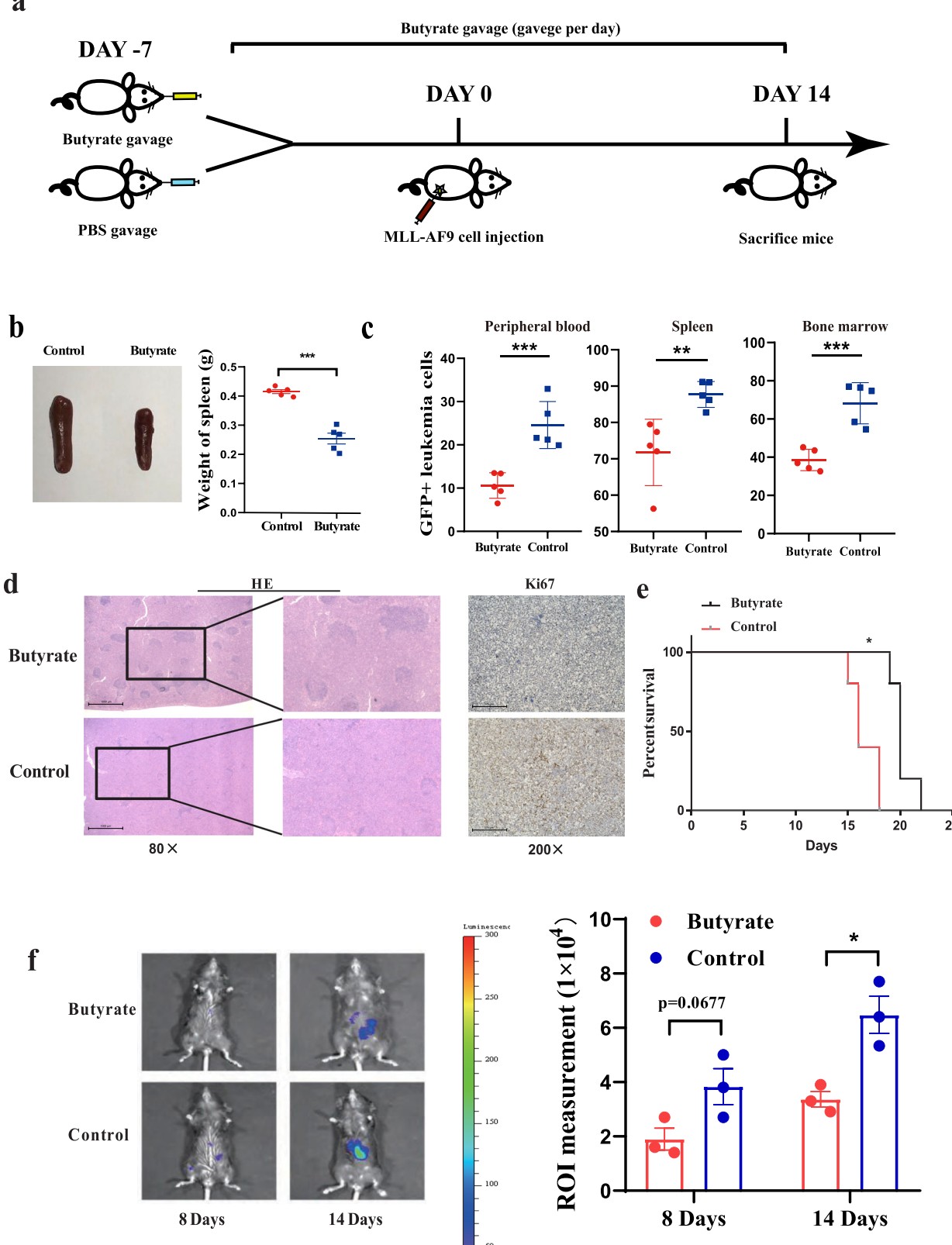

higher LPS concentrations. In vitro experiments showed that LPS increased leukaemia cell proliferation, and in vivo studies revealed that LPS exacerbated the leukaemia burden and shortened OS in AML mice. We also clarified that its leukaemia-promoting effects might function through decreased BAX and cleaved caspase-3 and upregulated bcl-2 levels. These results suggest that gut metabolites are another independent contributing source of elevated plasma LPS levels and that the absence of butyrate and elevated LPS play important roles in AML progression. Though our results demonstrated that LPS promoted AML progression in the experiments of in vitro cell culture and murine model, whether the small increase in LPS concentration plays a vital effect in

**Fig. 5 Microbiota-derived butyrate gavage delays AML progression. a** Schematic diagram of the mouse experimental process, including butyrate gavage. **b** Photographs of spleens from butyrate-treated AML mice ($n = 5$) and control AML mice ($n = 5$). **c** Leukaemia cells (GFP+ cells) in the bone marrow, peripheral blood, and spleen from butyrate-treated AML mice ($n = 5$) and control AML mice ($n = 5$). Details of the gating strategy are described in Supplementary Fig. 11b. **d** HE histopathology sections and Ki67 immunohistochemical staining of a representative spleen in butyrate-treated and control AML mice. All microscopic analyses were performed at an original magnification of ×80 or ×200, scale bar = 1000 and 275 μm. **e** Kaplan–Meier survival curve of AML mice ($n = 5$ per group). **f** On 8 and 14 days after injection of MLL-AF9 cells, the load of Luc-expressing MLL-AF9 cells in mice was analysed by IVIS ($n = 3$ per group). $P$ values were determined using unpaired two-tailed $t$-test and error bars represent mean ± SEM in **b**, **c**, **f**. $P$ values were determined Gehan–Breslow–Wilcoxon test and error bars represent mean ± SEM in **e**. ***$P < 0.0001$ (**b**), ***$P < 0.0001$ PB, **$P = 0.0066$ SP, ***$P = 0.0006$ BM (**c**), *$P = 0.0137$ (**f**), **$P = 0.0026$ (**e**). Source data are provided as a Source Data file.

AML patients is unknown. In addition, leukaemia patients need to take ABX to prevent infection, especially during chemotherapy. During this period, whether the use of antibiotics will cause the disorder of intestinal flora and the content of LPS in the serum of AML patients, and then have adverse effects on the disease, is an interesting question worthy of our further research.

Importantly, while our findings relate to treatment-naive AML patients, they cannot yet be generalised as an approach for patients with relapsed and refractory disease. Future studies are needed to elucidate how the gut microbiota and other SCFAs affect recurrent or refractory leukaemia. In addition, our research is currently limited to horizontal clinical studies, and more clinical samples are needed for longitudinal clinical studies. Finally, our study can only prove that gut microbiota dysbiosis and GI damage can promote the progression of existing AML, but there is no evidence to prove whether GI damage can cause the occurrence of AML. Therefore, we intend to gradually address these problems and provide further insight into the relationship between AML and the gut microbiota.

In conclusion, our results suggest that AML caused compromised intestine integrity, disordered bacteria diversity and the change of bacterial-derived factors. Decreased barrier function, loss of butyrate and higher blood LPS are cancer-promoting factors in AML and butyrate regulates the concentration of LPS in peripheral blood by affecting the intestinal barrier (Fig. 8). The regulation of gut microbiota metabolites, especially those targeting butyrate, may provide an approach for AML therapy.

## Methods

**Patient recruitment and specimens.** Sixty-one adults were enrolled from 2018 to 2020. Stool, peripheral blood and BM samples included in this study were obtained from 31 ND AML patients and 30 healthy controls. These specimens were collected at Qilu Hospital of Shandong University. The exclusion criteria were as follows: age <15 and >70 years, diarrhoea, receiving antibiotics or hormone treatment in the last 10 weeks, and blood pressure anomalies. The patients were divided into an AML patient group ($n = 31$) and a healthy control (Con) group ($n = 30$). Their characteristics are summarised in Supplementary Table 1, and the risk stratification standards are shown in Supplementary Table 2. Forty-three blood samples collected without fasting were used for metabolic profiling analysis (21 AML patients and 22 healthy controls), and another 28 blood and faeces samples were utilised for cell culture or functional studies. Ten BM samples were randomly chosen for metabolic profiling analysis ($n = 5$ per group). Among the 61 individuals, all were chosen for stool 16S rRNA sequencing. Our study was approved by the Medical Ethical Committee of Qilu Hospital of Shandong University (KYLL-2018-137) and informed consent was obtained from all participants (Including patients, healthy participants and minor participants). Our study did not involve any interventional clinical trial.

**Cell culture and reagents.** MLL-AF9, a murine AML cell line labelled with a GFP reporter, was obtained from the spleen of the successfully established MLL-AF9 murine AML model. Primary leukaemia cells were cultured in IMDM culture medium (Gibco, USA) supplemented with 10% foetal bovine serum (FBS, Gibco, USA) and 1% penicillin/streptomycin (Gibco, USA). MLL-AF9 cells were cultured in DMEM containing 10% FBS, cytokines (IL-3, IL-6, GM-SCF, and SCF) and 1% penicillin/streptomycin. C1498 cells were obtained from ATCC (Cat. TIB-49) and cultured in DMEM medium containing 10% FBS and 1% penicillin/streptomycin. Cells were incubated at 37 °C in a humidified atmosphere with 5% $CO_2$. LPS, dimethyl sulfoxide and FITC-dextran were purchased from Sigma–Aldrich (USA). Adriamycin (ADR), daunorubicin (DNR) and cytosine arabinoside (Ara-C) were

dissolved in normal saline and diluted in RPMI 1640 medium immediately before use. Primary antibodies against cleaved caspase-3 (#9661), claudin-2 (#48120) and Bcl-2 (#3498) used for western blot were obtained from Cell Signaling Technology (Beverly, MA, USA), and antibodies against ZO-1 (ab221547), occludin (ab167161), claudin-1 (ab180158), BAX (ab199677), GAPDH (ab8245), β-tubulin (ab6046) and β-actin (ab227387) were purchased from Abcam (Cambridge, UK). Anti-Ki67 antibody (DAKO, MIB-1, 1:200) used for immunohistochemistry. ZO-1 (Servicebio, GB111402,1:500), claudin-1 (Servicebio, GB11032,1:500) and claudin-2 (Abcam, ab53032,1:500) used for immunofluorescence. Secondary antibodies were obtained from ZSGB-BIO(China). All reagents were dissolved and preserved following the manual's instructions.

**Faecalibacterium culture.** *Faecalibacterium* (*F. prausnitzii* A2–165, DSM 17677) was grown at 37 °C in LYHBHI medium (brain–heart infusion medium supplemented with 0.5% yeast extract (Difco) and 5 mg/L) supplemented with cellobiose (1 mg/mL; Sigma–Aldrich), maltose (1 mg/mL; Sigma), and cysteine (0.5 mg/mL; Sigma) in an anaerobic chamber.

**Murine AML models.** Specific pathogen-free (SPF) mice on a C57BL/6J background were purchased from the Laboratory Animal Center of Shandong University. Female mice (6–8 weeks) were maintained in a 12-h dark/light cycle at ambient temperature (72 ± 2 F) with controlled humidity (~45%). A retrovirus vector containing the intracellular domain of MLL-AF9 (MSCV-MLL-AF9-IRES-GFP) was kindly provided by Dr Hui Cheng (Institute of Hematology, Chinese Academy of Medical Sciences). To establish a murine AML model, C-kit+ cells from the BM of WT mice transfected with MSCV-MLL-AF9-IRES-GFP ($10^7$ cells/host), together with BM mononuclear cells from C57BL/6J mice ($10^7$ cells/host), were transplanted into lethally irradiated C57BL/6J recipients. On day 30 after transplantation, AML mice were verified by detecting the GFP frequency in peripheral blood from the lateral tail vein with a FACS Aria II sorter (BD Biosciences). Then, we collected spleen cells from successfully established MLL-AF9 AML mice and froze them as murine MLL-AF9 AML cells at −80 °C. For subsequent murine experiments, we injected MLL-AF9 AML cells into WT mice through the tail vein without irradiation, which was used as a murine AML model. The MLL-AF9 murine AML model is commonly applied in many studies as it relates to human AML[52,53]. AML1-ETO murine model was established in the same way as the MLL-AF9 murine model and AML1-ETO cells were kindly provided by Dr Jianxiang Wang (Institute of Hematology and Blood Diseases Hospital, Chinese Academy of Medical Sciences & Peking Union Medical College). All mice were 6–8 weeks old and were maintained in an SPF environment. Animal protocols were approved by the Animal Ethics Committee of Qilu Hospital, Shandong University.

**Preparation of faecal hydration liquid.** Fresh faeces from patients or healthy controls were weighed and diluted in normal saline to achieve a final concentration of 150 mg/mL. The patients were selected based on their adequate faeces, balanced sex and risk stratification. Then, the faecal suspension was filtered with a 40-μm filter sieve to remove large particles, the filtrate was collected, glycerol was added to a concentration of 20%, and the samples were stored at −80 °C. The faecal hydration liquid was thawed for oral gavage when FMT was performed.

**Faecal microbiota transplantation (FMT) and antibiotics.** For FMT, mice were randomly grouped according to the experimental requirements. First, an ABX was administered by oral gavage. Briefly, the antibiotic was used as follows: on days −21 to −7 before MLL-AF9 cell injection, mice received a daily gavage of metronidazole (100 mg/kg), while the ABX (ampicillin (1 g/L), vancomycin (0.5 g/L) and neomycin (1 g/L)) was added to the drinking water. Then, the frozen faecal hydration liquid was thawed and mixed for the same group of AML patients or controls. FMT was carried out via oral gavage in a final volume of 250 μL. In order to keep the time consistent, we performed faecal bacteria transplantation at 8 am every day. FMT was performed daily from 14 to 35 days after the start of the antibiotic regimen. We also collected stool and faecal hydration liquid on day 0 and day 7 after transplantation for 16S rRNA analysis to prove the success of transplantation. See the Supplementary materials for detailed patient information (Supplementary Table 3).

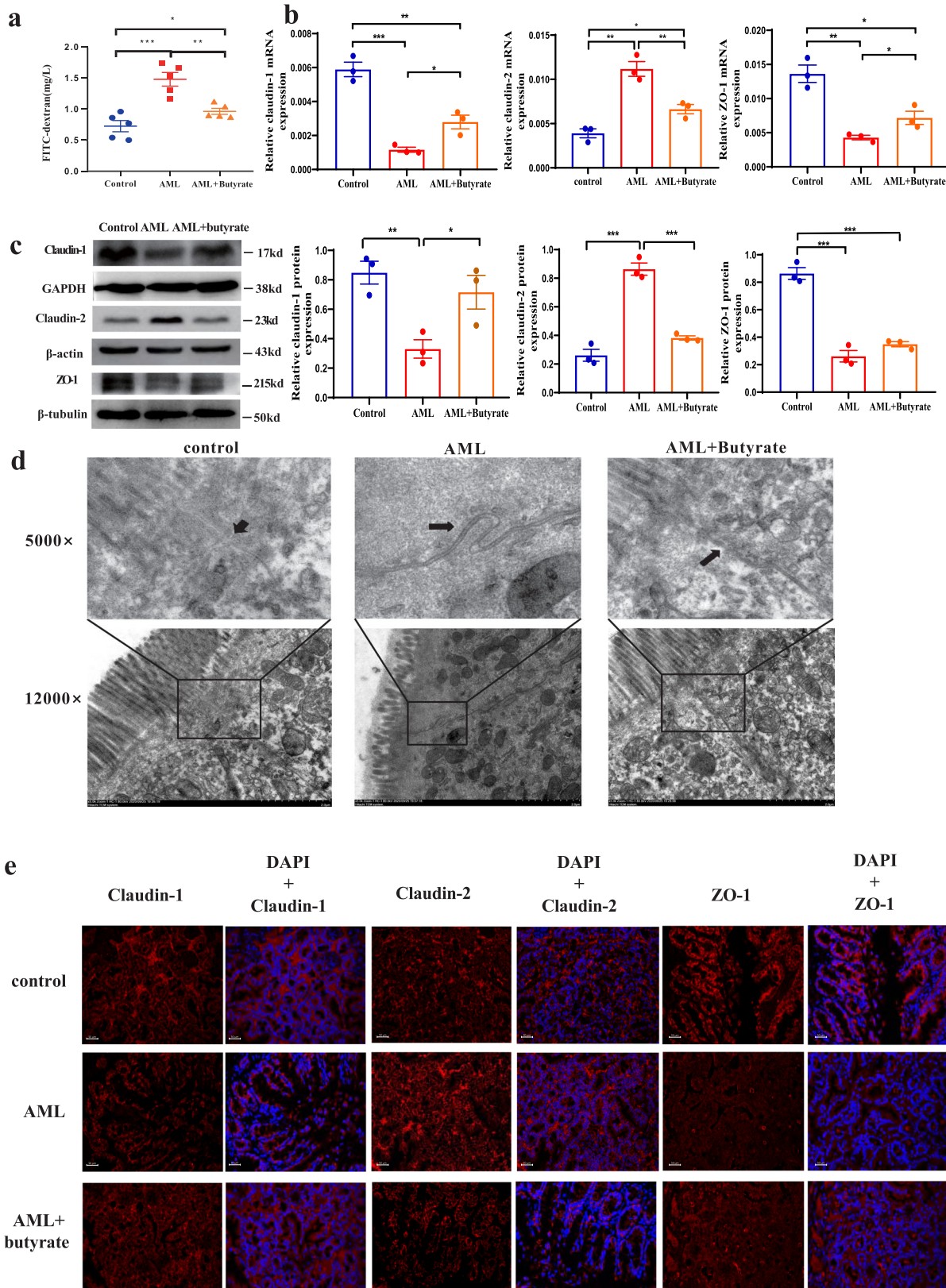

**Butyrate, LPS and DSS treatment**. Seven days before the injection of MLL-AF9 cells, mice received sodium butyrate (15 mg/kg) or carrier (sterile PBS) through a 20G-1.5-inch flexible intragastric gavage needle (Braintree Scientific; Braintree, MA) every day. Continue until the mouse is sacrificed. LPS was injected into mice at a concentration of 1 μg/mL in 200 μL via the tail vein together with MLL-AF9 cells. Next, it will be injected every 3 days during the experiment. Mice were given 1.9% DSS (MW 36,000–50,000 Da, Lot-No. S5036; MP Biomedicals, Santa Ana,

CA, USA) in autoclaved drinking water (w/v) for 3 days, followed by other days of DSS free drinking water.

**Dual-luciferase activity assay**. MLL-AF9 cells were transfected with a luciferase (MCS-Luciferase-IRES-Puror) retrovirus vector. Firefly and Renilla luciferase activities were measured consecutively 24 h following transfection using a Dual-

**Fig. 6 Butyrate reverses intestinal barrier damage in mice with AML. a** The concentration of FITC-dextran in the peripheral blood after FITC-dextran gavage for 6 h. Data represent the mean ± SEM ($n = 5$ per group). **b** The mRNA expression levels of the tight junction protein components claudin-1, claudin-2, and ZO-1 in intestinal epithelial cells of AML, control, and butyrate-treated mice ($n = 3$ per group). **c** The protein levels of claudin-1, claudin-2, and ZO-1 in intestinal epithelial cells were determined by western blot. GAPDH was used as the control ($n = 3$ per group). **d** Transmission electron microscopy (TEM) of intestines isolated from normal, AML, and butyrate-treated AML mice for the duration of the experiment; arrows indicate the cell–cell interface, scale bar = 2 μm. **e** Immunofluorescence analysis of intestine tissue from normal, AML, and butyrate-treated mice, scale bar = 50 μm. **d, e** Three times each experiment was repeated independently with similar results. Cells were fixed and stained with a rabbit polyclonal antibody. APC (red) goat anti-rabbit IgG was used as a secondary antibody. Immunofluorescence indicated the expression quantity and localisation of claudin-1, claudin-2, and ZO-1. $P$ values were determined using unpaired two-tailed $t$-test and error bars represent mean ± SEM in (**a, b, c**). ***$P = 0.0007$, **$P = 0.0027$, *$P = 0.0482$ (**a**), ***$P = 0.0005$ AML vs Control (claudin-1), **$P = 0.0063$ AML vs AML + Butyrate (claudin-1), *$P = 0.0192$ AML + Butyrate vs Control (claudin-1), **$P = 0.0021$ AML vs Control (claudin-2), **$P = 0.0099$ AML vs AML + Butyrate (claudin-2), *$P = 0.0203$ AML + Butyrate vs Control (claudin-2), **$P = 0.0021$ AML vs Control (ZO-1), *$P = 0.0475$ AML vs AML + Butyrate (ZO-1), *$P = 0.0155$ AML + Butyrate vs Control (ZO-1) (**b**), **$P = 0.0065$ AML vs Control (claudin-1), *$P = 0.0418$ AML vs AML + Butyrate (claudin-1), ***$P = 0.0005$ AML vs Control (claudin-2), ***$P = 0.0004$ AML vs AML + Butyrate (claudin-2), ***$P = 0.0005$ AML vs Control (ZO-1), ***$P = 0.0004$ AML + Butyrate vs Control (ZO-1) (**c**). Source data are provided as a Source Data file.

Luciferase Reporter Assay (Promega, USA) according to the manufacturer's instructions. Moreover, to monitor tumour progression and metastasis in vivo, bioluminescent imaging was performed with an IVIS Lumina Series III (Perkin-nElmer, USA) on days 10 and 14 after injection and was used to confirm the dynamic process of tumour cell load in vivo.

**Cell proliferation assays**. After different treatments, the cells were incubated with 10 μL of CCK-8 (Beyotime, China) for 3 h. The absorbance was measured at 450 nm. Each sample was measured in triplicate. Details of the gating strategy are described in Supplementary Fig. 11.

**Apoptosis assays**. Apoptosis was assessed with an Annexin V/propidium iodide (PI) apoptosis detection kit (BestBio, Shanghai, China) according to the manufacturer's protocol. Cells were harvested after different treatments and washed twice with PBS. Then, the cells were resuspended in 400 μL of binding buffer and stained with 5 μL of Annexin V for 15 min and 10 μL of PI for another 5 min in the dark at 4 °C. The percentages of apoptotic cells were analysed immediately using a Galios flow cytometer (Beckman Coulter, CA, USA).

**Enzyme-linked immunosorbent assay (ELISA)**. Blood plasma samples were collected from EDTA-stabilised peripheral blood of 20 ND AML patients and 18 controls, and supernatants of mouse peripheral blood were collected and stored at −80 °C for determination of the LPS concentration. Human and mouse LPS ELISA kits were purchased from R&D Systems (Minneapolis, MN, USA). ELISAs were performed in accordance with the manufacturer's instructions.

**Western blot analysis**. Cells were collected, washed twice with PBS, and then lysed with RIPA buffer (Beyotime, China) containing protease inhibitor compound (Beyotime, China) on ice. A bicinchoninic acid protein assay kit (Beyotime, China) was applied to measure protein concentrations. Protein extracts (30 μg) were loaded into 10% SDS-PAGE gels and then electrotransferred onto nitrocellulose membranes (Millipore, Bedford, MA, USA). After blocking with 5% nonfat milk for 1 h at room temperature, the membranes were incubated overnight with specific primary antibodies at 4 °C followed by incubation with HRP-conjugated secondary antibodies at room temperature for 1 h. Protein bands were detected by a FluorChem E Chemiluminescent imaging system (ProteinSimple, San Jose, CA, USA) after washing. Uncropped and unprocessed scans of western blots are available in the Source Data file.

**SCFA measurement**. For SCFA determination in faeces or blood plasma, we first screened out significantly different microbial metabolites by LC-MS. Briefly, for untargeted metabolomics of polar metabolites, extracts were analysed using a quadrupole time-of-flight mass spectrometer (Sciex TripleTOF 6600) coupled to hydrophilic interaction chromatography via electrospray ionisation at Shanghai Applied Protein Technology Co., Ltd. LC separation was performed on an ACQUIY UPLC BEH Amide column (2.1 mm × 100 mm, 1.7 μm particle size (Waters, Ireland) using a gradient of solvent A (25 mM ammonium acetate and 25 mM ammonium hydroxide in water) and solvent B (acetonitrile). The gradient was 85% solvent B for 1 min and was linearly reduced to 65% in 11 min, reduced to 40% in 0.1 min and maintained for 4 min, and then increased to 85% in 0.1 min, with a 5 min re-equilibration period. The flow rate was 0.4 mL/min, the column temperature was 25 °C, the autosampler temperature was 5 °C, and the injection volume was 2 μL. The mass spectrometer was operated in both negative ion and positive ionisation modes. The ESI source conditions were set as follows: ion source gas 1 (Gas1) as 60, ion source gas 2 (Gas2) as 60, curtain gas (CUR) as 30, source temperature of 600 °C, and ion spray voltage floating of ±5500 V.

For targeted SCFA metabolism, we quantitatively determined the content of SCFAs by gas chromatography-mass spectrometry GC-MS. Samples were thawed on ice, and 100-μL aliquots were added to a 2-mL glass centrifuge tube and mixed with 50 μL of water with 15% phosphoric acid and 150 μL of 5 μg/mL 4-methyl valeric acid. The suspensions were homogenised with a vortex for approximately 1 min and centrifuged for 10 min at 12,000× g. Then, 1 μL of supernatant was collected for GC-MS analysis using an Agilent Model 7890A/5975C GC-MS system. To quantify SCFAs, a calibration curve for the concentration ranges from 0.1 to 100 μg/mL was constructed. The internal standard was used to correct for injection variability between samples and minor changes in the instrument response.

The samples were separated with an Agilent HP-INNOWAX capillary GC column (30 m × 0.25 mm ID × 0.25 μm). The initial temperature was 90 °C, which was increased to 120 °C at 10 °C/min and then to 150 °C at 5 °C/min and finally to 250 °C at 25 °C/min, where it remained for 2 min. The carrier gas was helium (1.0 mL/min). The temperatures of the injection port and transmission line were 250 and 230 °C, respectively. An electron bombardment ionisation source was used, selected ion monitoring scanning mode was applied, and the electron energy was 70 eV (Shanghai Applied Protein Technology Co., Ltd.).

**Microbial diversity analysis**. Microbial DNA was extracted from faecal samples using an E.Z.N.A.® soil DNA Kit (Omega Bio-tek, Norcross, GA, USA) according to the manufacturer's protocols. The final DNA concentration and purification were determined using a NanoDrop 2000 UV-vis spectrophotometer (Thermo Scientific, Wilmington, USA), and the DNA quality was checked by 1% agarose gel electrophoresis. The V3-V4 hypervariable regions of the bacterial 16S rRNA gene were amplified with primers 338 F (5'-ACT CCT ACG GGA GGC AGC AG-3') and 806 R (5'-GGA CTA CHV GGG TWT CTA AT-3') using a thermocycler PCR system (GeneAmp 9700, ABI, USA). PCR was conducted using the following programme: 3 min of denaturation at 95 °C; 27 cycles of 30 s at 95 °C, 30 s of annealing at 55 °C, and 45 s of elongation at 72 °C; and a final extension at 72 °C for 10 min. PCR was performed in triplicate in a 20-μL mixture containing 4 μL of 5× FastPfu Buffer, 2 μL of 2.5 mM dNTPs, 0.8 μL of each primer (5 μM), 0.4 μL of FastPfu Polymerase, and 10 ng of template DNA. The resulting PCR products were extracted from a 2% agarose gel and further purified using an AxyPrep DNA Gel Extraction Kit (Axygen Biosciences, Union City, CA, USA) and quantified using QuantiFluor™-ST (Promega, USA) according to the manufacturer's protocol. Purified amplicons were pooled in equimolar amounts and paired-end sequenced (2 × 300) on an Illumina MiSeq platform (Illumina, San Diego, USA) according to standard protocols by Majorbio Bio-Pharm Technology Co. Ltd. (Shanghai, China). OTUs were clustered with a 97% similarity cut-off using UPARSE (version 7.1 http://drive5.com/uparse/), and chimeric sequences were identified and removed using UCHIME. The taxonomy of each 16S rRNA gene sequence was analysed by the RDP Classifier algorithm (http://rdp.cme.msu.edu/) against the Silva (SSU123) 16S rRNA database using a confidence threshold of 70%.

**Processing of sequencing data**. Raw fastq files were demultiplexed, quality-filtered by Trimmomatic, and merged by FLASH with the following criteria: (i) reads were truncated at any site receiving an average quality score <20 over a 50-bp sliding window; (ii) primers were exactly matched, allowing two nucleotide mismatches, and reads containing ambiguous bases were removed; (iii) sequences overlapping more than 10 bp were merged according to their overlap sequence. Alpha diversity was measured with Shannon metrics, and beta diversity was calculated using UniFrac.

**Quantitative reverse transcriptase PCR**. Total RNA was extracted from cells using TRIzol (Invitrogen, USA). The total RNA concentration and purity were quantified using a spectrophotometer (Eppendorf, Germany). Reverse transcription

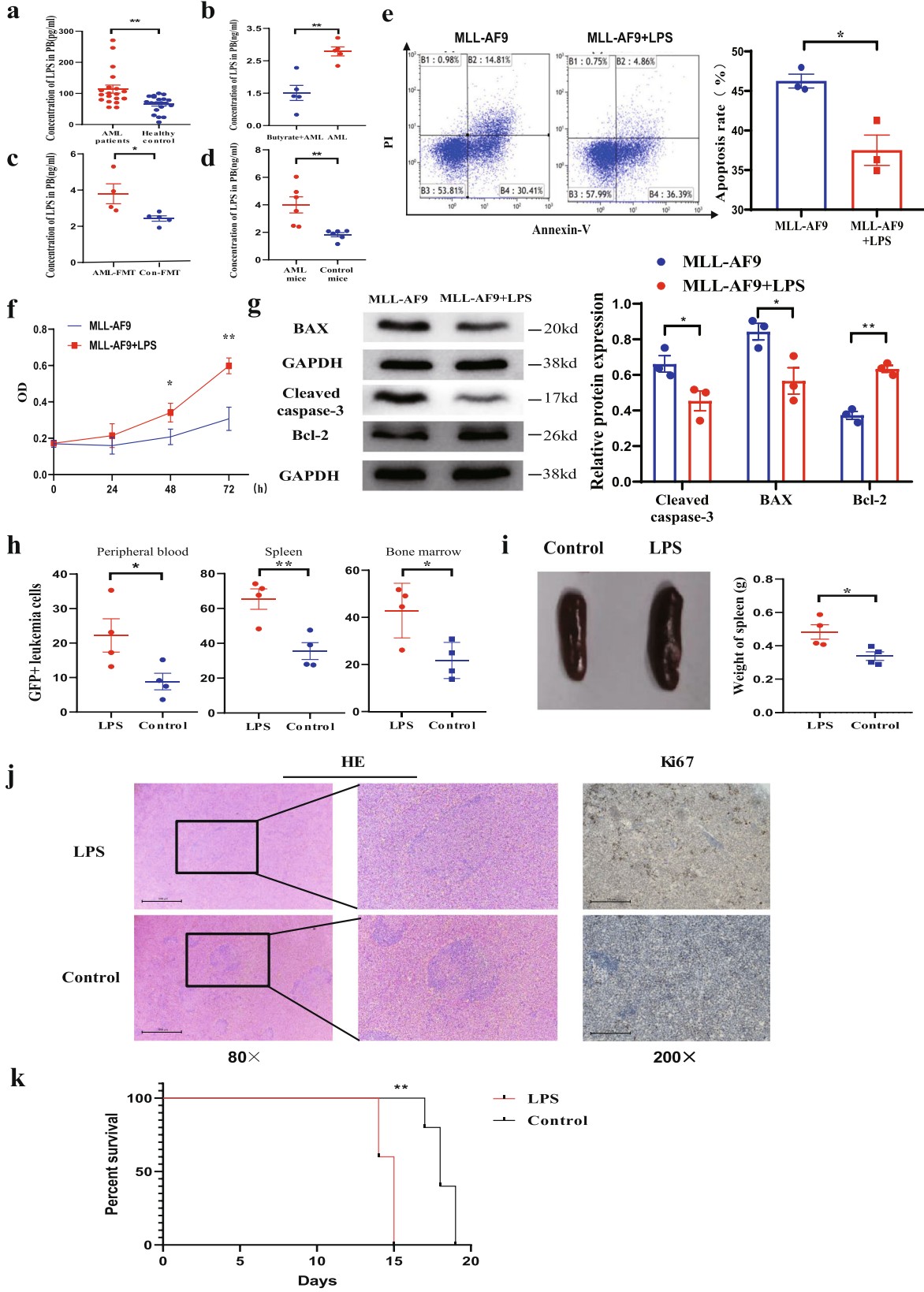

was performed at 37 °C for 15 min followed by 85 °C for 10 s using a Prime Script RT reagent Perfect Real Time Kit (Takara Bio Inc, Japan). Quantitative PCR was performed in duplicate on a Light Cycler 480II real-time PCR system (Roche, Switzerland) with a SYBR Green Real-time PCR Master Mix kit (Toyobo, Japan). The PCR contained 3.2 µL of ddH$_2$O, 5 µL of 2×SYBR Green Real-time PCR Master Mix, 0.4 µL of forward and reverse primers, and 1 µL of cDNA in a final volume of 10 µL. PCR conditions were as follows: 95 °C for 10 min, followed by 40 cycles of 95 °C for 20 s and 60 °C for 1 min. The primer sequences for the genes are shown in Supplementary Table 2. To determine the specificity of the PCRs, melting curves were routinely analysed. All experiments were conducted according to the manual's instructions. Relative gene expression was expressed relative to that of the endogenous control GAPDH and calculated using the $2^{-\Delta CT}$ method.

**Fig. 7 Damage to the intestinal barrier accelerates bacterial-derived LPS leakage into the blood, and LPS exacerbates the progression of AML. a** LPS concentrations in the peripheral plasma of AML patients and healthy controls ($n = 20$ per group, biologically independent samples). **b** LPS concentrations in butyrate-treated and control AML mice. **c** LPS concentrations in AML-FMT and Con-FMT mice. **d** LPS concentrations in AML mice and normal mice after LPS gavage. **b–d** All data are from animal's independent experiments. **e** Representative FACS graphs of apoptotic MLL-AF9 cells after culture with or without LPS for 48 h ($n = 3$ per group). Details of the gating strategy are described in Supplementary Fig. 11a. **f** CCK-8 analysis of the proliferation of MLL-AF9 cells with or without LPS for 24, 48, and 72 h ($n = 3$ per group). **g** The western blot results for Bcl-2, BAX, cleaved caspase-3, and GAPDH was used as controls ($n = 3$ per group). **h** Leukaemia cells (GFP+ cells) in the spleen, peripheral blood, and bone marrow from LPS-treated AML mice ($n = 4$) and control AML mice ($n = 4$). Details of the gating strategy are described in Supplementary Fig. 11b. **i** Representative photographs of spleens and the weight of spleens from LPS-treated AML mice ($n = 4$) and control AML mice ($n = 4$). **j** HE histopathology sections and Ki67 immunohistochemical staining of a representative spleen, LPS-treated AML mice, and control AML mice. All microscopic analyses were performed at an original magnification ×80 or ×200, scale bar = 1000 and 275 µm. **k** Kaplan–Meier survival curve of the mouse leukaemia model ($n = 4$ per group). $P$ values were determined using unpaired two-tailed $t$-test and error bars represent mean ± SEM in **a–i**. $P$ values were determined using Gehan–Breslow–Wilcoxon test and error bars represent mean ± SEM in (**k**). **$P = 0.0017$ (**a**), **$P = 0.015$ (**b**), *$P = 0.0322$ (**c**), **$P = 0.0051$ (**d**), *$P = 0.0113$ (**e**), *$P = 0.0261$, **$P = 0.0028$ (**f**), *$P = 0.00454$ Cleaved caspase-3, *$P = 0.0034$ BAX, ***$P = 0.0009$ BCL-2 (**g**), *$P = 0.0476$ PB, **$P = 0.0077$ SP, *$P = 0.025$ BM (**h**), *$P = 0.0276$ (**i**), **$P = 0.0043$ (**k**). Source data are provided as a Source Data file.

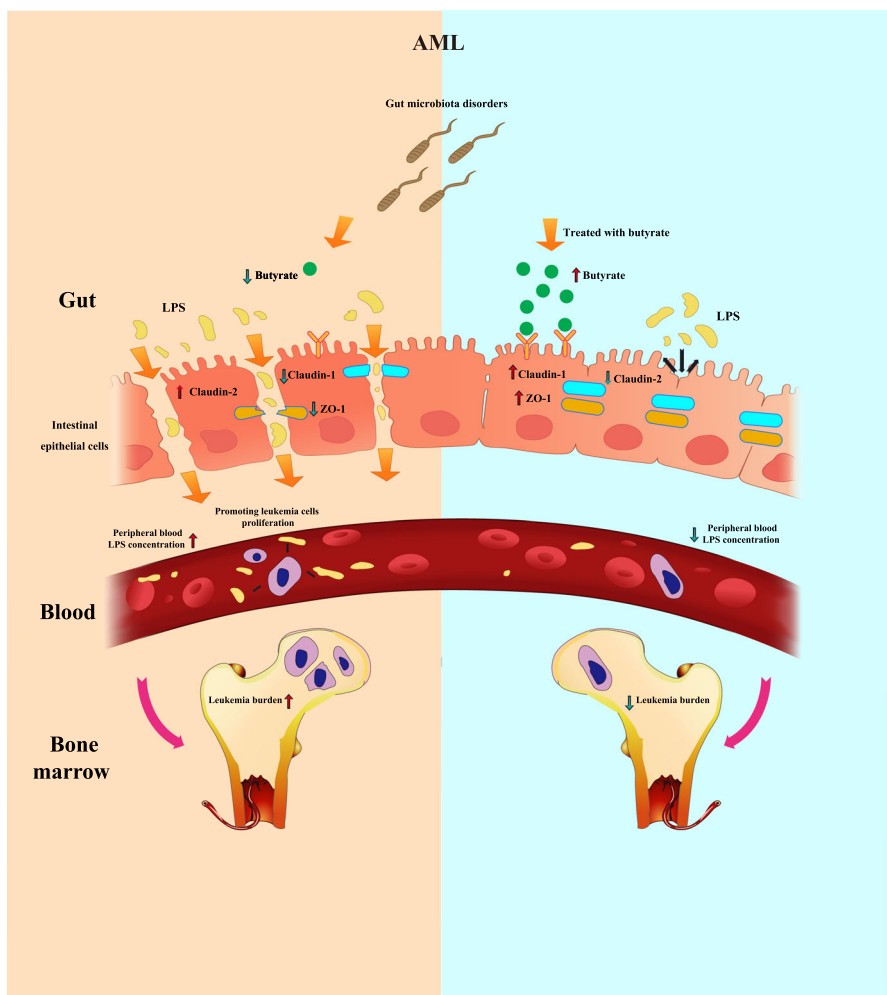

**Fig. 8 Gut microbiota regulates AML via alteration of intestinal barrier function and LPS blood concentration.** Gut microbiota disorders in AML lead to a decrease in intestinal butyrate. Decreased butyrate weakens gut barrier function by altering tight junction proteins (claudin-1, claudin-2, ZO-1). Dysregulation of intestinal barrier function allows LPS leakage into blood, which in turn promotes the proliferation of AML cells and accelerates AML progression. These effects could be reversed by treatment with butyrate or FMT.

**FITC-dextran assay.** Food and water were withheld from all mice for 4 h. FITC-dextran was administered by a 20G-1.5-inch flexible intragastric gavage needle at a concentration of 50 mg/mL in PBS. Mice received FITC-dextran at 800 mg/kg (~16 mg/mouse). Four hours later, serum was collected from peripheral blood, diluted 1:1 with PBS, and analysed on a plate reader at excitation/emission wavelengths of 485 nm/535 nm. Concentrations of FITC-dextran in experimental samples were determined based on a standard curve.

**Transmission electron microscopy.** Briefly, the intestines from control mice or AML mice treated with butyrate or vehicle were harvested 14 days after injection of MLL-AF9. Cross-sections were immediately sliced (2 mm wide) from the colon. The tissue was then rinsed three times with Sorensen's buffer containing 0.1% ruthenium red (RR) and postfixed for 1 h in 1% osmium tetroxide in the same buffer containing RR. The samples were again rinsed with Sorensen's buffer containing RR. Next, the tissue was dehydrated in ascending concentrations of ethanol, treated with propylene

oxide, and embedded in Epon epoxy resin. Semithin sections were stained with toluidine blue for tissue identification. Selected regions of interest were ultrathin sectioned (70 nm thick), mounted on copper grids, and poststained with uranyl acetate and lead citrate. The samples were examined using a Philips CM100 electron microscope at 60 kV. Images were recorded digitally using a Hamamatsu ORCA-HR digital camera system operated using AMT software (Servicebio, Wuhan, China).

**Statistical analysis**. GraphPad Prism 8.0 was applied for graph development. ImageJ V1.50 was applied for western blot image processing and Kaluza 2.1.1 was applied for flow cytometry analysis. Statistical analysis was conducted with raw data using SPSS 20.0 software. Logistic regression was used to analyse the causal relationship between *Faecalibacterium* and WBC. The correlation between *Faecalibacterium* and WBC was performed by multiple linear regression. The Shapiro–Wilk test was used for normality tests. Normally distributed data were analysed by Student's *t*-test or paired *t*-tests. Otherwise, comparisons between groups were performed using the Mann–Whitney $U$ test for nonpaired data and the Wilcoxon signed-rank test for paired data. Results are expressed as the mean ± SEM, except where stated otherwise. $*P < 0.05$, $**P < 0.01$ and $***P < 0.0001$ were considered statistically significant. All figures' details of the statistical analysis and the exact $P$ value are in the Supplementary Data.

**Reporting summary**. Further information on research design is available in the Nature Research Reporting Summary linked to this article.

## Data availability

Source Data are provided with this paper. The metabolic raw datasets generated in this study have been deposited in the metabolights database under accession code MTBLS3092; username: polly0719@163.com and password: 582f70] and 16S rRNA sequencing datasets were uploaded to the NCBI database with SRA, accession: PRJNA746137.

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

## Acknowledgements

This work was supported by the National Natural Science Foundation of China (Grant Nos. 81873439 (D.-X.M.), 91642110 (D.-X.M.), 81770159 (C.-Y.J.) and 81873425 (T.S.)), Distinguished Taishan Scholars in Climbing Plan (Grant No. tspd20210321 (C.-Y.J.)) and Young Taishan Scholars (Grant No. tsqn201812132 (T.S.)).

## Author contributions

D.-X.M. designed and funded the research. C.-Y.J. helped design the research. R.W. performed the research and wrote the manuscript. X.Y. and J.L. assisted with the research. F.Z. and C.Z. analysed the data. T.S. and Y.C. edited the paper.

## Competing interests

The authors declare no competing interests.
