## [Peer Review File · Nature Communications]

Gut microbiota regulates acute myeloid leukaemia via alteration of intestinal barrier function mediated by butyrateREVIEWER COMMENTS

Reviewer #1 (Remarks to the Author); expert on intestinal barrier:

The major claim of this paper is that the alteration in gut microbiota plays an important pathogenic role in the progression of AML, mainly that observed decrease in microbial (MB) diversity contributes to AML progression. The investigators performed series of interlinked studies to investigate the involvement of gut MB in a mouse model of AML. The studies are presented sequentially and suggest that a reduction in *Faecalibacterium* abundance may contribute to AML pathogenesis, that a fecal MB transplant from AML patients may further promote AML process, that bacterial derived butyrate may have a protective role by targeting the gut barrier and prevent gut-bacteria derived LPS from crossing the intestinal barrier. The hypothesis that the gut MB plays a role in the pathogenesis of AML is not particularly innovative but certainly important topic to study and the studies presented fill the gap in knowledge to the role of MB. Overall, the studies are novel and were successful in providing insights into the various aspects of MB alteration and its potential role in AML pathogenesis. The studies also generated a number of interesting hypotheses. The weakness mainly relates to the lack of detailed studies or experiments to establish a definitive cause-and-effect relationship as it relates to the gut MB and AML, role of *Faecalibacterium* and AML, and the role of gut barrier, butyrate, and LPS.

Specific points to consider:

1. It is unclear how valid the animal model of AML used in this study as it relates to the human AML. There were no discussions or reviews of the literature on this topic.
2. Although the studies showed a decrease in MB diversity and which likely has a modulating role in the pathogenesis, there were no definitive studies to show the cause and effect- key studies to show that the preservation of WT MB flora would inhibit the development of AML in the mouse model.
3. Although *Faecalibacterium* was identified as being reduced and an interesting and intriguing hypothesis generated that this could have an etiologic role. This intriguing possibility was not fully explored or validated and it is unclear what role if any *Faecalibacterium* has. More definitive study to address this cause and effect relationship would greatly increase the novelty and impact of this work. While it was proposed that *Faecalibacterium* can cause an increase in butyrate, the investigators did not validate this in their experiments. More detailed investigation of *Faecalibacterium* supplement on intestinal/stool butyrate rate would also provide an insight into the role *Faecalibacterium* plays in butyrate generation.
4. The studies with butyrate, barrier function, and LPS were very interesting and hypothesis generating but, in this reviewer's point of view, these studies made the overall work somewhat diffuse and superficial, providing some insight into the interconnected process and generating interesting hypotheses but lacking the detailed studies to fully validate the roles each play.
 - a. The studies with butyrate were somewhat descriptive and did not definitively prove their role. This also applies to butyrate having a potential beneficial mechanism of protection for *Faecalibacterium*-no data are provided to establish this causal link.
 - b. The studies with intestinal barrier are superficial at best and suggest alteration in intestinal permeability but there is no mechanistic insight into the increase in Dextran. The increase in claudin-2 would not explain this. What happens with occludin and other TJ protein expression.
 - c. The increase in LPS reported in AML patients is minimal at approximately 120 pg/ml and would not be expected to have any physiological relevance. In other studies, such as in liver diseases, LPS levels are elevated at much higher levels.
5. Overall, there were many interesting and novel studies that were carried out by the investigators- which helped generate several interesting hypotheses. In particular, the alteration in stool MB and the reduction in *Faecalibacterium* are quite interesting and novel. In this reviewer's view, it would have been much more impactful and paradigm shifting to focus on the etiologic role of *Faecalibacterium* in AML and to provide more definitive supporting data-establishing cause and effect relationship than to provide series of possibilities that are superficially examined.

Reviewer #2 (Remarks to the Author); expert on metagenomics and microbiome:

In this study, xxx and colleagues explored the role of the gut microbiota in AML. The authors observed difference in terms of fecal microbiota diversity and composition in AML patients in comparison to healthy controls. In mouse model, microbiota depletion with antibiotics worsens AML severity. FMT from microbiota of patient with AML worsens AML severity. The authors then observed that the microbiota of patients with AML produce less butyrate. The decreased concentration of butyrate was observed in the feces but not in the blood). Butyrate gavage was then showed to improve AML severity in mice. This effect was associated with an improvement of AML-induced increase in intestinal permeability and lower systemic LPS concentration. Finally, LPS administration IV worsens AML severity in mice.

Comments:

- Many confounding factors are likely to be present in the comparison between healthy subjects and patients with AML. A simple univariate comparison is not appropriate. A more sophisticated analysis taking into account potential bias is required.

- The FMT protocol is unclear.

- Strains and Culture information are not provided for *Faecalibacterium*. Moreover, the experiment with this bacterium cannot be compared with the FMT one. The *F. prausnitzii* experiment evaluate an improvement compared to baseline while the FMT one is looking at a worsening following FMT from AML patient. I suggest doing FMT experiment with feces from AML patient, supplemented or not with *F. prausnitzii*.

- It seems that only one AML and one healthy subject were used as donor for the FMT experiment. It seems not sufficient. Multiple donors should be tested.

Reviewer #3 (Remarks to the Author); expert on AML, metabolism and microbiome:

The current manuscript by Wang et al described that the pathogenesis of AML leads to microbiota dysbiosis, which causes a reduced level of SCFA including butyrate and thus a disordered barrier function of intestinal epithelia. They further reported that due to the damaged barrier function of intestinal epithelia in AML mice and patients, endotoxin such as LPS produced by microbiota can pass through and enter peripheral blood, which is utilized by leukemia cells as stimuli for both growth and survival. While the manuscript covered multiple lines of investigation, there are several concerns that needs to be addressed.

Major concerns:

1. The novelty of current study is largely comprised. The major phenotypes (AML results in dysbiosis and damaged epithelial barrier function) and mechanisms (reduced production of butyrate other SCFAs) described in current manuscript have already been reported by others (Ye et al, Cancer Cell 2018; <https://doi.org/10.1016/j.ccell.2018.08.016>). Although the authors used a different mouse model, it is inappropriate for them to claim that their finding is a 'breakthrough'.

2. The establishment of MLL-AF9 leukemia model used in the current manuscript requires lethal irradiation according to their method section. Irradiation causes full body damages including intestinal tissues, and therefore it raises the question that whether the phenotypes observed in this model is only under the condition of irradiation (also irradiation seen in AML patients is rare) ? For example, the authors showed that Abx treatment promotes leukemia progression in MLL-AF9 mice (Fig.2). Will Abx still have similar effects if no irradiation is performed? Similar question for Fig.3, 5, 6 and 7.

3. According to the authors, LPS produced by gut microbiota facilitates progression of MLL-AF9 leukemia (Fig.7), and Abx also promotes MLL-AF9 leukemia progression (Fig.2). Abx depletes gut flora and therefore LPS production should be greatly reduced. Can the authors explain the contradiction of their findings?

4. Although differences between groups were significant, most of the data sets of animal studies were collected at a quite late stage of the disease (more than 70% of leukemia blast in bone marrow and spleen) and differences between each group were very subtle. For example, in Fig.2 C tumor burden in bone marrow was about 91% vs. 88% (and similar cases in Fig. 3C and Fig.5C), which impairs the significance of their work. It will be helpful for the authors to examine leukemia burden at earlier stages of the disease.

5. In Fig.3, the authors showed that fecal samples from AML patients promotes MLL-AF9 leukemia progression. Which patient sample was used in this experiment? According to Fig.1 the risk favorable and unfavorable patients have different microbiota composition. Will mice show different leukemia burden when fecal samples from different risk groups are used?

6. Colon is the tissue with most flora and is more affected by microbiota and most of the authors' data were generated based on the analyses of fecal samples. However, in Fig.6 the authors generated data from small intestines. It will be helpful if they can generate these data from colon tissue.

Minor points:

1. In Fig.1F, the author showed that risk favorable and unfavorable patients have different microbiota composition. Does this result still hold true for the actual outcomes of these patients (patient showed in supplemental table1)?

2. Bacteroides is greatly increased in AML patient (Fig.1C). Could the authors comment on this observation?

3. What was the injection method for mice LPS treatment (Fig.7)?

4. Fig.4E needs more detailed explanation.

5. English writing might need some retouching.

Reviewer #4 (Remarks to the Author); expert on metabolomics and microbiome:

This is a very interesting manuscript, which studied the relationship between the gut microbiota, microbial metabolites and AML progression. Multiple mouse studies were used to further investigate the role of the bacteria or bacterial metabolites in AML. However, there are a few concerns to be addressed.

1. The antibiotics use in the patient group (no antibiotics use within the last 10 weeks) may still bias the finding since it has been reported that it can take between 1-12 months for the gut microbiota to normalize.

2. How were the blood samples chosen for metabolic analysis? Were they plasma or serum and how they were collected? Were patients fasting prior to the collection?

3. It is not clear if fecal hydration liquid was pooled from all patients or a subset. If latter, how were the donors selected?

4. How were the animals housed? Singly or grouped? If grouped, how were they grouped?

5. It would be helpful to have a graph showing the mouse experimental design.

6. Why was 70% confidence threshold chosen for determining taxonomy of 16S sequences?

7. Figure 1C and lines 340-342. Many of the p values shown here are not consistent between the text and the figure. Why? Were the p value corrected for multiple test correction?

8. Lines 353-355.: was the correlation performed based on all patients and healthy control samples? If so it is not surprising to see a strong correlation between Faecalibacterium and Roseburia, and WBC. Have the authors try to do the correlation only with samples from patients to further confirm this

correlation? It would be more convincing.

9. Figure 2A - n=3 per group is too small - any justification for the power calculation? How were the animals housed? This is crucial for evaluating the gut microbiota due to cage effects. While there is no significant differences in Sob index at day 0 between the Con and AML, it shows a significant difference in PCoA plot. Why is it? How about Shannon index? The label in the figure is confusing - if control is AML without antibiotics, maybe change them to 'AML_con' and 'AML_abx'. What is the rationale of using antibiotics before MLL-AF9 injection? If the hypothesis is that AML causes the bacterial shift, why not directly inject the cells and monitor the bacterial changes in these animals? It will also be essential to keep the control and AML mice in the same cage to avoid cage-effect. As it is the experiment should have had another control group with antibiotics treatment without AML to determine if the microbial shift is due to the antibiotics or AML. The data may support that the disturbed gut bacterial composition by the antibiotics accelerate AML progression but it does not support that AML causes bacterial dysbiosis.

10. Line 381, "Faecalibacterium correlated with a favorable prognosis" this correlation should be done within patients rather than with the inclusion of healthy control. Which Faecalibacterium was used? Isolated from patients and cultured or directly use the fecal slurry? If the latter, it is not a pure Faecalibacterium FMT and wording should be changed. Again could n=4 per group give a meaningful result?

11. Figure S4a, was before con-FMT time point after antibiotics? If so, this is not surprising. It would be helpful to show health control donors, AML patient donors, and after AML FMT and after Con-FMT in one figure. Again, the labels are confusing. Since Control has been used to described healthy control patients, it would be good to use a different abbreviation, e.g. AML_con or AML_nab (no antibiotics). Was it one donor corresponding to one recipient? If so it should be good to label the pair in the PCoA plot.

12. Figure 4a, use cross validated scores plot instead of OPLS-DA scores plot. Which sample type is this plot based on? Lines 420-422, were the MS/MS patterns matched with the database?

13. Line 422-424, in the method, it mentioned that SCFAs were analysed using GC-MS but here it indicated as LC-MS. Which one was used? Figure 4D, again can this correlation be performed in patient group and healthy control groups, separately?

14. Figure 4F keep the color of the groups consistent throughout the manuscript. Were these p values corrected for multiple test correction?

15. All the data including metabolic and 16S rRNA sequencing raw data should be made publicly available via data repositories.

Minor:

Line 31: "...observed a significant shift..." was it in feces or gut content? Clarify here.

Line 37-38 "Metabolome analyses showed that microbiota derived butyrate concentration obviously decreased in AML patient feces" should be written "Metabolic analyses showed that fecal concentrations of butyrate, produced by the gut microbiota, significantly decreased in AML patients,"

Line 40, add comma before 'which'

Line 89, replace "GVHD patients" to "patients with GVHD"

Lines 91 and 343, replace 'flora' with 'bacteria' r'microbiota'

Line 96, should be 'disordered'

Line 96-97 the past tense should be used.

Line 117, another 28 samples - which sample type?

Line 130, CO₂ where 2 should be subscripted.

Line 157-158: does it mean that "fresh feces from patients or healthy control were weighted and diluted with normal saline to achieve a final concentration of 150 mg/mL."?

Line 159, what filter was used?

Line 182: in vivo should be italicized

Line 214, which sample types were used to measure SCFAs?

Line 278 H₂O where 2 should be subscripted.

Line 326, should it be 'all fecal samples' rather than 'all samples'?

Figure 1B, variations should be labeled clearer, not overlapping with the axis.

Line 336 should change microbiota to bacteria.

Line 338 add 'compared to the healthy control' after 'AML patients'

Line 338 remove 'approximately'

Figure 1D, change blue to red for positive correlation.

Line 347 change 'depleted' to 'lower' and add 'compared to control group'.

Figure S1A, why are there 72 samples?

Line 357 change 'disordered' to 'low'

Line 356, please elaborate on 'favorable-risk'. How was favorable and non-favorable defined.

Line 363 should be 16S rRNA

Point by point responses to reviewers' comments

We appreciate the reviewers for their thoughtful and constructive comments and suggestions regarding our manuscript. We have addressed all of the questions and concerns through additional experimentation and clarification. The detailed point-by-point responses in blue font are given below.

REVIEWER COMMENTS

Reviewer #1 (Remarks to the Author); expert on intestinal barrier:

The major claim of this paper is that the alteration in gut microbiota plays an important pathogenic role in the progression of AML, mainly that observed decrease in microbial (MB) diversity contributes to AML progression. The investigators performed series of interlinked studies to investigate the involvement of gut MB in a mouse model of AML. The studies are presented sequentially and suggest that a reduction in Faecalibacterium abundancy may contribute to AML pathogenesis, that a fecal MB transplant from AML patients may further promote AML process, that bacterial derived butyrate may have a protective role by targeting the gut barrier and prevent gut-bacteria derived LPS from crossing the intestinal barrier. The hypothesis that the gut MB plays a role in the pathogenesis of AML is not particularly innovative but certainly important topic to study and the studies presented fill the gap in knowledge to the role of MB. Overall, the studies are novel and were successful in providing insights into the various aspects of MB alteration and its potential role in AML pathogenesis. The studies also generated a number of interesting hypotheses. The weakness mainly relates to the lack of detailed studies or experiments to establish a definitive cause -and-effect relationship as it relates to the gut MB and AML, role of Faecalibacterium and AML, and the role of gut barrier, butyrate, and LPS.

Response: Thank you very much for your overall positive comments and constructive suggestions. As you suggested, we performed new experiments to further clarify a definitive cause-and-effect relationship as it relates to the gut MB and AML, role of Faecalibacterium and AML, and the role of gut barrier, butyrate, and LPS. Our detailed responses are listed below.

Specific points to consider:

1. It is unclear how valid the animal model of AML used in this study as it relates to the human AML. There were no discussions or reviews of the literature on this topic.

Response: Thanks a lot for your comment. MLL-AF9 murine AML model, the animal model of AML used in our study, is valid as it relates to the human AML. The well-studied MLL-AF9 oncogene initiates myeloid leukaemia in both humans and mice [1]. The MLL-AF9 transgenic murine model that results in myeloid leukaemia has been described and studied in some detail [2, 3]. Similarly, in our study, MLL-AF9 murine AML model was established by transplanting C-kit⁺ cells transfected with MLL-AF9-GFP together with other supporting bone marrow mononuclear cells into C57BL/6J recipients. After being verified successfully establishment, their spleen cells were collected and frozen as MLL-AF9 murine AML cells. These murine AML cells were injected into SPF WT C57BL/6J mice which were used as AML mice for the subsequent experiments in our study. This AML model could imitate the progression of human AML, which is useful and valid for our research to explore the role of gut microbiota in AML progression. There are many studies used this animal model of AML to investigate their own purpose genes or pathways in the progression of AML *in vivo*. For example, Zheng et al. used this animal model to demonstrate that Dnmt3b plays a tumor suppressive role in MLL-AF9 AML progression [4]. Hayashi et al. applied this animal model to prove that antitumor immunity augments the therapeutic effects of p53 activation on acute myeloid leukaemia [5]. To address your comment and make it more clear, we added a sentence which indicated the literature on this topic in the new version

“The MLL-AF9 murine AML model is commonly applied in many studies as it relates to human AML [4-5].” (Lines 161-164)

2. Although the studies showed a decrease in MB diversity and which likely has a modulating role in the pathogenesis, there were no definitive studies to show the cause and effect- key studies to show that the preservation of WT MB flora would inhibit the development of AML in the mouse model.

Response: Thank you very much for your valuable comment. To address this point, we

performed new FMT experiments. We resumed and preserved WT MB flora of AML mice via oral gavage of WT mice faecal suspension while oral gavage of PBS as control, and then determined the murine leukaemia burden of AML. Our results showed that the frequencies of AML cells in BM, spleen, and blood of AML mice gavaged with WT mice faecal suspension were decreased significantly compared with AML mice gavaged with PBS control. It can also be verified from the decreased size and weight of the spleen and mitigated leukaemia cell infiltration in AML mice gavaged with normal mice faecal suspension (Line 419-429, Figure 2F-H). These results demonstrated that the preservation of WT MB flora could inhibit the development of AML in the mouse model.

3. Although *Faecalibacterium* was identified as being reduced and an interesting and intriguing hypothesis generated that this could have an etiologic role. This intriguing possibility was not fully explored or validated and it is unclear what role if any *Faecalibacterium* has. More definitive study to address this cause-and-effect relationship would greatly increase the novelty and impact of this work. While it was proposed that *Faecalibacterium* can cause an increase in butyrate, the investigators did not validate this in their experiments. More detailed investigation of *Faecalibacterium* supplement on intestinal/stool butyrate rate would also provide an insight into the role *Faecalibacterium* plays in butyrate generation.

Response: Thanks a lot for your valuable comments and suggestions. In our original version, we showed that *Faecalibacterium* correlated with a favorable prognosis and may be considered as a beneficial bacterium in AML, and we studied the function of *Faecalibacterium* through bacterial transplantation. The results showed that *Faecalibacterium* gavage obviously relieved the splenomegaly (Supplementary figure 3a-b). In terms of tumor load, we observed a significant reduction in the bone marrow and a reduced tendency in the spleen and peripheral blood (Supplementary figure 3c). The overall survival of the AML mice treated by *Faecalibacterium* gavage was not significantly different from that of control AML mice (Supplementary figure 3d).

To further address the cause-and-effect relationship between *Faecalibacterium* and AML, we performed new murine FMT experiment with faeces from AML patients (AML-FMT), supplemented with or without *Faecalibacterium*. Our results showed that the spleen sizes and the leukaemia cells frequencies in bone marrow and spleen were

significantly reduced in AML-FMT mice supplemented with Faecalibacterium compared with AML-FMT mice without Faecalibacterium. The leukaemia cells frequencies in peripheral blood also had a decreased trend, but did not reach significant difference. All these results demonstrated that Faecalibacterium could decrease the leukaemia burden in AML mice and AML-FMT mice, but the inhibiting effect of single Faecalibacterium supplement was weaker than the transplantation of an intact gut microbiota. We added these results in the new version (Supplementary figure 3e-g, line 441-452).

To provide more insight into the role Faecalibacterium plays in butyrate generation, we performed additional two experiments. Firstly, we cultured Faecalibacterium *in vitro*, and determined the content of butyrate in the cultured supernatant of Faecalibacterium. The result showed that the concentration of butyrate in the supernatant of Faecalibacterium was significantly higher than that in controls. Secondly, we administrated Faecalibacterium to mice via gavage and then detected the stool butyrate. Our results showed that the stool butyrate content was much higher in mice gavaged with Faecalibacterium than in controls. Therefore, we confirmed that butyrate is a direct metabolite of Faecalibacterium, and that Faecalibacterium could cause an increase in butyrate. We added these results in the new version. (Line 500-510, Supplementary figure 5 g-h).

4. The studies with butyrate, barrier function, and LPS were very interesting and hypothesis generating but, in this reviewer's point of view, these studies made the overall work somewhat diffuse and superficial, providing some insight into the interconnected process and generating interesting hypotheses but lacking the detailed studies to fully validate the roles each play.

Response: We thank you very much for your valuable and constructive comments. We performed a number of new experiments, and the detailed responses are listed below.

a. The studies with butyrate were somewhat descriptive and did not definitively prove their role. This also applies to butyrate having a potential beneficial mechanism of protection for Faecalibacterium- no data are provided to establish this causal link.

Response: Thanks a lot for your valuable comment. In our study, we first found that

there was decreased butyrate positively correlated with the decreased Faecalibacterium in the stools of AML patients (Figure 4B, C and D). Then, we further demonstrated the roles of Faecalibacterium and butyrate in AML mice.

As for the role of Faecalibacterium in AML, our results showed that Faecalibacterium gavage obviously relieved the splenomegaly, decreased leukaemia cells burden in AML mice (Supplementary figure 3a-c). Moreover, we performed new experiments and the results showed that the leukaemia burdens, including the splenomegaly, leukaemia cells frequencies in spleen, bone marrow and peripheral blood, were significantly reduced in AML FMT mice after supplementation with Faecalibacterium (Supplementary figure 3e-g, line 441-452). All these results demonstrated the important role of Faecalibacterium in AML.

As for the role of butyrate in AML, we administered butyrate to mice via gavage and determined the effect of butyrate on leukaemia load and survival time in AML mice. We found that butyrate treatment significantly alleviated the splenomegaly, decreased the percentage of leukaemia cells burden, mitigated the leukaemia cells infiltration and proliferation in the spleen, and alleviated AML progression in AML mice (Figure 5D, F). We further determined the effects of butyrate on intestinal integrity. Our results showed that butyrate treatment significantly ameliorated intestinal damage of AML mice (Figure 6A) and preserved the integrity of the intestinal epithelial cell's junction (Figure 6D, right panel). Furthermore, with the new data of additional experiments, our results mechanistically demonstrated that butyrate treatment restored the expressions of downregulated cell-cell junction protecting protein and upregulated cell-cell junction-inhibiting protein in AML mice (Figure 6B, C, Supplementary figure 6a). All these results definitively demonstrated the important role and mechanisms of butyrate in AML.

Then, as the relationship between butyrate and Faecalibacterium. There are many references reporting that Faecalibacterium could generate butyrate [6, 7]. As the above response mentioned, we performed new experiments and proved that Faecalibacterium generated butyrate in culture supernatant and Faecalibacterium gavage elevated the butyrate content in mice stool (Line 500-510, Supplementary figure 5 g-h). These references and our results demonstrated that Faecalibacterium plays a vital role in butyrate generation, and provide data to establish the causal link for potential beneficial mechanism of protection in AML.

b. The studies with intestinal barrier are superficial at best and suggest alteration in intestinal permeability but there is no mechanistic insight into the increase in Dextran. The increase in claudin-2 would not explain this. What happens with occludin and other TJ protein expression.

Response: We appreciate your helpful comment. With your constructive suggestion, to further provide the mechanistic insight into the increase in Dextran, we determined the expression of occludin and other TJ molecules (TJP2, Claudin-3, Claudin-4 and Claudin-8). Our results showed that butyrate treatment significantly increased the mRNA expressions of occludin and claudin-3 in intestinal epithelial cells though no significant change was found between normal mice and AML mice. We further detected the protein expression of occludin using Western blot method. The result showed that occludin expression was also increased significantly in butyrate treatment mice, which is consistent with the mRNA expression level. These findings suggest that intestinal barrier damage caused by AML might not be due to occludin expression change, but butyrate could protect intestinal barrier via further increasing occludin expression. Despite of our new findings, the detailed molecular mechanisms associated with damage or protection of intestinal barrier in AML need to be further studied in the future. (Line 563-566, 569-577, 582-584; Figure 6B, C, Supplementary figure 6a-b)

c. The increase in LPS reported in AML patients is minimal at approximately 120 pg/ml and would not be expected to have any physiological relevance. In other studies, such as in liver diseases, LPS levels are elevated at much higher levels.

Response: Thank you for your comment. It is a truth that the concentration of LPS in AML patients is not very high in our manuscript and may be lower than other diseases, which needs detailed explanation. There are many papers reporting that low concentration of LPS has its physiological relevance. Deopurkar et al. reported that plasma LPS concentrations in normal people increased to 90pg/ml-120pg/ml after cream intake, which increased the expression of TLR-4 expression and are relevant to the pathogenesis of atherosclerosis and insulin resistance [8]. Ghanim et al reported that high-fat, high-carbohydrate meal induced an increase in plasma LPS concentration (105pg/ml-130pg/ml) and the expression of SOCS-3, TLR2, and TLR4 protein, reactive oxygen species generation, and nuclear factor-kappaB binding activity [9]. Different

from inflammation or inflammatory diseases, the development of human acute myeloid leukaemia is a slow, long and gradual process. During this process, long-term low-concentration LPS may cause a series of changes of molecular pathways and play oncogenic role for haematopoietic cells. Therefore, the increase in LPS reported in AML patients in our study would have physiological and pathological relevance.

5. Overall, there were many interesting and novel studies that were carried out by the investigators- which helped generate several interesting hypotheses. In particular, the alteration in stool MB and the reduction in Faecalibacterium are quite interesting and novel. In this reviewer's view, it would have been much more impactful and paradigm shifting to focus on the etiologic role of Faecalibacterium in AML and to provide more definitive supporting data-establishing cause and effect relationship than to provide series of possibilities that are superficially examined.

Response: We appreciate your positive comments and constructive suggestions. As mentioned above, we performed additional experiments to provide more definitive supporting data for the cause-and-effect relationship between Faecalibacterium and AML. Firstly, using *in vitro* culture and *in vivo* gavage experiments, we confirmed that butyrate is a direct metabolite of Faecalibacterium, and that Faecalibacterium could cause an increase in butyrate (Supplementary figure 5 g-h). Secondly, to further address the cause-and-effect relationship between Faecalibacterium and AML, we performed new murine FMT experiment with faeces from AML patients, supplemented with or without Faecalibacterium. Our results showed that the spleen sizes and the leukaemia cells frequencies in bone marrow and spleen were significantly reduced in AML-FMT mice supplemented with Faecalibacterium compared with AML-FMT mice without Faecalibacterium. The leukaemia cells frequencies in peripheral blood also had a decreased trend, but did not reach significant difference. (Supplementary figure 3e-g). All these results demonstrated that Faecalibacterium could decrease the leukaemia burden in AML mice and AML-FMT mice, but the inhibiting effect of single Faecalibacterium supplement was weaker than the transplantation of an intact gut microbiota. We added these results in the new version.

Reference

1. Dobson, C.L., et al. The *mll*-AF9 gene fusion in mice controls myeloproliferation

- and specifies acute myeloid leukaemogenesis. *EMBO J*,1999 Jul 1;18(13):3564-74.
2. Johnson, J.J. et al. Prenatal and postnatal myeloid cells demonstrate stepwise progression in the pathogenesis of MLL fusion gene leukaemia. *Blood*,2003 Apr 15;101(8):3229-35.
 3. Kumar, A.R., et al. Hoxa9 influences the phenotype but not the incidence of MLL-AF9 fusion gene leukaemia. *Blood*,2004 Mar 1;103(5):1823-8
 4. Zheng et al. Loss of Dnmt3b accelerates MLL-AF9 leukaemia progression. *Leukaemia*, 2016 Dec; 30(12):2373-84
 5. Hayashi et al. Antitumor immunity augments the therapeutic effects of p53 activation on acute myeloid leukaemia. *Nat Commun*. 2019 Oct 25;10(1):4869
 6. Lenoir M, et al. Butyrate mediates anti-inflammatory effects of *Faecalibacterium prausnitzii* in intestinal epithelial cells through Dact3. *Gut Microbes*. 2020 Nov 9;12(1):1-16.
 7. Zhou L, et al. *Faecalibacterium prausnitzii* Produces Butyrate to Maintain Th17/Treg Balance and to Ameliorate Colorectal Colitis by Inhibiting Histone Deacetylase 1. *Inflamm Bowel Dis*. 2018 Aug 16;24(9):1926-40.
 8. Deopurkar R, et al. Differential effects of cream, glucose, and orange juice on inflammation, endotoxin, and the expression of Toll-like receptor-4 and suppressor of cytokine signaling-3. *Diabetes Care*. 2010 May;33(5):991-7.
 9. Ghanim H, et al. Increase in plasma endotoxin concentrations and the expression of Toll-like receptors and suppressor of cytokine signaling-3 in mononuclear cells after a high-fat, high-carbohydrate meal: implications for insulin resistance. *Diabetes Care*. 2009 Dec;32(12):2281-7.

Reviewer #2 (Remarks to the Author); expert on metagenomics and microbiome:

In this study, xxx and colleagues explored the role of the gut microbiota in AML. The authors observed difference in terms of fecal microbiota diversity and composition in AML patients in comparison to healthy controls. In mouse model, microbiota depletion with antibiotics worsens AML severity. FMT from microbiota of patient with AML worsens AML severity. The authors then observed that the microbiota of patients with AML produce less butyrate. The decreased concentration of butyrate was observed in the feces but not in the blood). Butyrate gavage was then showed to improve AML severity in mice. This effect was associated with an improvement of AML-induced

increase in intestinal permeability and lower systemic LPS concentration. Finally, LPS administration IV worsens AML severity in mice.

Comments:

- Many confounding factors are likely to be present in the comparison between healthy subjects and patients with AML. A simple univariate comparison is not appropriated. A more sophisticated analysis taking into account potential bias is required.

Response: We appreciate your valuable comments and constructive suggestion. We reanalyzed the data with more sophisticated analysis taking into account potential bias, such as age and gender, which might be confounding factors for the results of gut microbiota [1, 2]. Firstly, we stratified the subjects (patients and controls) into 3 groups, the young (Age < 30), the middle-aged ($30 \leq \text{Age} < 55$) and the old ($55 \leq \text{Age}$), and analyzed the gut microbiota diversity. The result showed that the Shannon index (α -diversity) is still decreased in the faeces of AML patients compared with that in healthy controls in any of the 3 groups, and reaches statistical significance in the middle-aged group. We also reanalyzed the microbiota data according to gender. The result showed that Shannon index (α -diversity) is still decreased in AML patients compared with that in healthy controls in either male or female group, and reaches significant difference in male group. In the age- or gender-stratified groups, the content of Faecalibacterium of AML patients is also lower than that of controls. Therefore, our results are still valid when taking accounting for other confounding factors. We added these results in the manuscript as supplementary data (Line 394-398; supplementary figure 6c, d).

We also used multiple linear regression method, instead of spearman correlation, to analyze the association between Faecalibacterium and WBC taking accounting of other factors such as Roseburia and age. Our results demonstrated that Faecalibacterium are significantly negatively correlated with the level of peripheral white blood cells. We added these results in the manuscript as supplementary data (Line 386-388; supplementary table 3).

- The FMT protocol is unclear.

Response: We described the FMT protocol in detail in our new version. For FMT, mice were randomly grouped according to the experimental requirements. First, an antibiotic mix (ABX) was administered by oral gavage. Briefly, the antibiotic was used as follows:

on days -21- -7 before MLL-AF9 cell injection, mice received a daily gavage of metronidazole (100 mg/kg), while the antibiotic mix (ampicillin (1 g/L), vancomycin (0.5 g/L) and neomycin (1 g/L)) was added to the drinking water. Then, the frozen faecal hydration liquid was thawed and mixed for the same group of AML patients or controls. FMT was carried out via oral gavage in a final volume of 250 μ L. In order to keep the time consistent, we performed faecal bacteria transplantation at 8 am every day. FMT was performed daily from 14 to 35 days after the start of the antibiotic regimen. We also collected stool and faecal hydration liquid on day 0 and day 7 after transplantation for 16s rRNA analysis to prove the success of transplantation. See the supplementary materials for detailed patients' information (Supplementary table 3). (Line 175-188).

- Strains and Culture information are not provided for Faecalibacterium. Moreover, the experiment with this bacterium cannot be compared with the FMT one. The *F. prausnitzii* experiment evaluate an improvement compared to baseline while the FMT one is looking at a worsening following FMT from AML patient. I suggest doing FMT experiment with feces from AML patient, supplemented or not with *F. prausnitzii*.

Response: As suggested, we provide the Strains and Culture information for Faecalibacterium in the new version (Line144-148). Moreover, with your valuable suggestion, we performed new murine FMT experiment with faeces from AML patients (AML-FMT), supplemented with or without *F. prausnitzii*. Our results showed that the spleen sizes and the leukaemia cells frequencies in bone marrow and spleen were significantly reduced in AML-FMT mice supplemented with Faecalibacterium compared with AML-FMT mice without Faecalibacterium. The leukaemia cells frequencies in peripheral blood also had a decreased trend, but did not reach significant difference. All these results demonstrated that Faecalibacterium could decrease the leukaemia burden in AML mice and AML-FMT mice, but the inhibiting effect of single Faecalibacterium supplement was weaker than the transplantation of an intact gut microbiota. We added these results in the new version (Line 441-452, Supplementary figure 3e-g).

- It seems that only one AML and one healthy subject were used as donor for the FMT experiment. It seems not sufficient. Multiple donors should be tested.

Response: Thanks a lot for your comment. We apologize for the misunderstanding caused by our description in the manuscript. We used 5 AML patients and 5 healthy controls as donors for the FMT experiments in our study. To reduce the differences between individuals, we mixed the fecal hydration liquid of individuals from the same group (AML patients or controls), and gavaged them into the mice. Then we determined the bacterial colonization using 16s rRNA sequencing (day 0 and day 7) and performed the following experiments. The patients' information was shown in Supplementary table 3. The results were shown in Figure 3 and supplementary figure 4a.

Reference

1. Santos-Marcos JA, et al. Sex Differences in the Gut Microbiota as Potential Determinants of Gender Predisposition to Disease. *Mol Nutr Food Res*. 2019;63(7): e1800870.
2. O'Toole PW, Jeffery IB. Gut microbiota and aging. *Science*. 2015;350(6265):1214-5

Reviewer #3 (Remarks to the Author); expert on AML, metabolism and microbiome:

The current manuscript by Wang et al described that the pathogenesis of AML leads to microbiota dysbiosis, which causes a reduced level of SCFA including butyrate and thus a disordered barrier function of intestinal epithelia. They further reported that due to the damaged barrier function of intestinal epithelia in AML mice and patients, endotoxin such as LPS produced by microbiota can pass through and enter peripheral blood, which is utilized by leukaemia cells as stimuli for both growth and survival. While the manuscript covered multiple lines of investigation, there are several concerns that needs to be addressed.

Major concerns:

1. The novelty of current study is largely comprised. The major phenotypes (AML results in dysbiosis and damaged epithelial barrier function) and mechanisms (reduced production of butyrate other SCFAs) described in current manuscript have already been reported by others (Ye et al, *Cancer Cell* 2018; <https://doi.org/10.1016/j.ccell.2018.08.016>). Although the authors used a different mouse model, it is inappropriate for them to claim that their finding is a 'breakthrough'.

Response: Thank you very much for raising this point. We are aware of the study by Ye et al, as it demonstrated that leukaemia cells hijack host glucose by inducing IGFBP1 production from adipose tissue to mediate insulin sensitivity and by inducing gut dysbiosis, serotonin loss, and incretin inactivation to suppress insulin secretion. As the reviewer said, different from our murine AML model, their findings used a different mouse model [blast crisis chronic myeloid leukaemia (bcCML) model (BNmodel)] for the research of leukaemia-associated microbiota, epithelial barrier and SCFAs. Therefore, as the reviewer suggested, to make our finding more rigorous, we softened our statement in the manuscript, and replaced “breakthrough” with “new findings” (Line 582-584).

2. The establishment of MLL-AF9 leukaemia model used in the current manuscript requires lethal irradiation according to their method section. Irradiation causes full body damages including intestinal tissues, and therefore it raises the question that whether the phenotypes observed in this model is only under the condition of irradiation (also irradiation seen in AML patients is rare)? For example, the authors showed that Abx treatment promotes leukaemia progression in MLL-AF9 mice (Fig.2). Will Abx still have similar effects if no irradiation is performed? Similar question for Fig.3, 5, 6 and 7.

Response: Thanks a lot for your careful comment. The description of murine leukaemia model in the method section of our study may be too simple, which makes you misunderstanding the role of irradiation in our experiments. We used lethal irradiation to establish MLL-AF9 leukaemia model in the first batch of mice, then no irradiation was applied in the subsequent batches of mice for murine MLL-AF9 AML experiments. Briefly, we firstly successfully established MLL-AF9 leukaemia model by transplanting C-kit+ cells transfected with MLL-AF9-GFP together with other supporting cells into C57BL/6J recipient mice, in which it requires lethal irradiation. Then, we collected their spleen MLL-AF9 AML cells and injected them into WT C57BL/6J mice as murine AML model, in which no irradiation was applied. We used these murine AML mice for our subsequent in vivo experiments. Therefore, there is no effect of irradiation on the results of Fig.2, 3, 5, 6 and 7.

3. According to the authors, LPS produced by gut microbiota facilitates progression of MLL-AF9 leukaemia (Fig.7), and Abx also promotes MLL-AF9 leukaemia progression

(Fig.2). Abx depletes gut flora and therefore LPS production should be greatly reduced. Can the authors explain the contradiction of their findings?

Response: Thanks a lot for your careful comment. It is a truth that Abx depletes LPS-producing gut flora of mice, and meanwhile LPS or Abx promotes MLL-AF9 leukaemia progression in our study. However, these two results are not contradictory. Firstly, Abx can't deplete all the gut bacteria as shown by our results [supplementary figure 2a] and other researchers [1, 2], and the remaining bacteria still could produce LPS. Secondly, Abx was shown to damage the gut epithelial integrity [3, 4], which facilitates more LPS leaking into blood, and this may compensate for the possible decreased LPS production caused by Abx depleted gut bacteria. Thirdly, more LPS could be released by the dead bacteria when ABx applied. However, the detailed mechanisms need further study in the future.

4. Although differences between groups were significant, most of the data sets of animal studies were collected at a quite late stage of the disease (more than 70% of leukaemia blast in bone marrow and spleen) and differences between each group were very subtle. For example, in Fig.2 C tumor burden in bone marrow was about 91% vs. 88% (and similar cases in Fig. 3C and Fig.5C), which impairs the significance of their work. It will be helpful for the authors to examine leukaemia burden at earlier stages of the disease.

Response: Thanks a lot for your valuable comment. The characteristics of MLL-AF9 AML mice model is that the injected leukaemia cells first infiltrate and proliferate in bone marrow and spleen, and then enter peripheral blood. Therefore, when leukaemia cells appear in peripheral blood, the leukaemia cells burden in bone marrow or spleen is much higher. Moreover, in our study, the generation of the effect of AML cells on gut microbiota or the effect of FMT on AML progression in vivo needs a slow and long process, so at the late stage of the disease the effect could be more apparent. In many other studies, the leukaemia cells burden is also heavy in bone marrow or spleen when they used MLL-AF9 mice as the disease model to explore their purpose signaling pathways [5, 6].

Considering the comment you raised is very helpful for us, according to your suggestion, we repeated our ABX gavage experiments to see the association between

AML and gut microbiota at early stage. As below shown, at early stage (on day 9), the results demonstrated that the peripheral blood, spleen and bone marrow of AML mice with ABX gavage had significantly increased leukaemia cell infiltration after tail vein injection of MLL-AF9 AML cells, and the degree of difference is consistent with our previous experiments about the late stage of AML model. Moreover, in the future we will find other AML mice model with longer disease period to better obtain early disease data under long-term experimental processing.

Sacrificing mice in the early stages of AML animal model (9days).

5. In Fig.3, the authors showed that fecal samples from AML patients promotes MLL-AF9 leukaemia progression. Which patient sample was used in this experiment? According to Fig.1 the risk favorable and unfavorable patients have different microbiota composition. Will mice show different leukaemia burden when fecal samples from different risk groups are used?

Response: Thanks for your comment. The information of patient samples used in Fig.3 were shown as below.

	Age(year)	Gender	Positive genes
Patient-1	34	Male	FLT3-ITD, NPM1, IDH2, SRSF2
Patient-2	62	Male	CEBPA
Patient-3	56	Male	FLT3-ITD, NPM1, DMMT3A
Patient-4	47	Female	CBFB-MYH11
Patient-5	49	Female	IDH2, NPM1

As the reviewer said, it is a valuable suggestion to perform FMT from patients of different risk groups. Therefore, we tried our best to collect the faeces samples from AML patients, divide them into favorable and unfavorable groups, and do FMT to AML mice with their faecal hydration liquid. Our results of these new experiments showed

that the leukaemia cells burden in bone marrow, spleen or peripheral blood of AML mice gavaged with faecal hydration liquid from favorable-risk patients had a decreased trend compared with that from unfavorable-risk patients, which is consistent with our hypothesis. Moreover, the weight of spleen from AML mice gavaged with faecal hydration liquid from favorable-risk patients also had a decreased tendency. However, all these differences did not reach statistical significance. The reason may be due to that: Many other factors, including cytogenetical and molecular changes, immunological status, also affect the risk stratification of AML. Though the risk favorable and unfavorable patients have different microbiota composition, these differences might not be strong enough to significantly affect the progress of AML in the body. The results of new experiments are shown as below.

Unfavorable-FMT Favorable-FMT

6. Colon is the tissue with most flora and is more affected by microbiota and most of the authors' data were generated based on the analyses of fecal samples. However, in Fig.6 the authors generated data from small intestines. It will be helpful if they can generate these data from colon tissue.

Response: Thanks a lot for your careful comment. We are very sorry for our mistake. We generated these data of Fig.6 from colon tissue but not from small intestines in our study. We corrected them in the new version.

Minor points:

1. In Fig.1F, the author showed that risk favorable and unfavorable patients have different microbiota composition. Does this result still hold true for the actual outcomes of these patients (patient showed in supplemental table1)?

Response: We followed up part of these patients for their actual outcomes. As most of them are alive until now, we used whether acquiring complete remission (CR) after first therapy as the outcome index. This result still holds true for the actual outcomes of these patients. The CR rate of the patients with favorable risk is higher than that of the patients with unfavorable risk (p value=0.015). We added them in the new version (Line 390-394, supplementary table 3).

2. Bacteroides is greatly increased in AML patient (Fig.1C). Could the authors comment on this observation?

Response: Thanks a lot for your comment. It is a truth that Bacteroides is greatly increased in AML patient (Fig.1C), which indicates that Bacteroides might be involved in the progression of AML. There are also several studies demonstrating the possible association and role of Bacteroides in leukaemia [7, 8]. We have not deepened the research of Bacteroides in our study mainly because: 1. The p value of Bacteroides (P=0.04332) between AML patients and healthy controls is higher than that of Faecalibacterium (P=0.0001), or Roseburia (P=0.000017). 2. Bacteroides is not significantly related to SCFAs especially butyrate.

3. What was the injection method for mice LPS treatment (Fig.7)?

Response: We include the injection method for mice LPS treatment in the new version (Line 193-195).

4. Fig.4E needs more detailed explanation.

Response: Thanks a lot for your suggestion. To make Fig.4E more detailed explanation, we rewrote the sentence as “The results showed that AML patients and healthy controls

had significant differences and identified abundant bacterial functions in the COG database. For example, DNA-binding transcriptional and fatty acid biosynthesis (COG0416, COG1309) were decreased enriched in AML patients than healthy people (Figure 4E).” (Line 515 - 520).

5. English writing might need some retouching.

Response: The revised manuscript has been edited by the American Journal Experts. (Certificate attached).

Reference

1. Lewis BB, Buffie CG, Carter RA, et al. Loss of Microbiota-Mediated Colonization Resistance to Clostridium difficile Infection With Oral Vancomycin Compared With Metronidazole. *J Infect Dis.* 2015;212(10):1656-1665. doi:10.1093/infdis/jiv256
2. Buffie CG, Jarchum I, Equinda M, et al. Profound alterations of intestinal microbiota following a single dose of clindamycin results in sustained susceptibility to Clostridium difficile-induced colitis. *Infect Immun.* 2012;80(1):62-73. doi:10.1128/IAI.05496-11.
3. Shi Y, Kellingray L, Zhai Q, et al. Structural and Functional Alterations in the Microbial Community and Immunological Consequences in a Mouse Model of Antibiotic-Induced Dysbiosis. *Front Microbiol.* 2018;9: 1948. Published 2018 Aug 21. doi:10.3389/fmicb.2018.01948
4. Fachi JL, Felipe JS, Pral LP, et al. Butyrate Protects Mice from Clostridium Difficile-Induced Colitis through an HIF-1-Dependent Mechanism. *Cell Rep.*

- 2019;27(3):750-761.e7. doi: 10.1016/j.celrep.2019.03.054
5. Ramakrishnan R, Peña-Martínez P, Agarwal P, et al. CXCR4 Signaling Has a CXCL12-Independent Essential Role in Murine MLL-AF9-Driven Acute Myeloid Leukaemia. *Cell Rep.* 2020;31(8):107684. doi: 10.1016/j.celrep.2020.107684
 6. Velasco-Hernandez T, Soneji S, Hidalgo I, Erlandsson E, Cammenga J, Bryder D. Hif-1 α Deletion May Lead to Adverse Treatment Effect in a Mouse Model of MLL-AF9-Driven AML. *Stem Cell Reports.* 2019;12(1):112-121. doi:10.1016/j.stemcr.2018.11.023
 7. Chua LL, Rajasuriar R, Lim YAL, Woo YL, Loke P, Ariffin H. Temporal changes in gut microbiota profile in children with acute lymphoblastic leukemia prior to commencement during and post-cessation of chemotherapy. *BMC Cancer.* 2020;20(1):151. Published 2020 Feb 24. doi:10.1186/s12885-020-6654-5
 8. Bindels LB, Neyrinck AM, Salazar N, et al. Non-Digestible Oligosaccharides Modulate the Gut Microbiota to Control the Development of Leukemia and Associated Cachexia in Mice. *PLoS One.* 2015;10(6):e0131009. Published 2015 Jun 22. doi:10.1371/journal.pone.0131009

Reviewer #4 (Remarks to the Author); expert on metabolomics and microbiome:

This is a very interesting manuscript, which studied the relationship between the gut microbiota, microbial metabolites and AML progression. Multiple mouse studies were used to further investigate the role of the bacteria or bacterial metabolites in AML. However, there are a few concerns to be addressed.

1. The antibiotics use in the patient group (no antibiotics use within the last 10 weeks) may still bias the finding since it has been reported that it can take between 1-12 months for the gut microbiota to normalize.

Response: We thank you for your overall positive comments. The criteria of patient inclusion in our study are no antibiotics use within the last 10 weeks, which is within the range between 1-12 months. In clinical practice, it may be not easy to recruit AML patients without using antibiotics within one year because the fever or other symptoms of AML might make the patients to take antibiotics. Many other researches also included patients without using antibiotics within 2 months for their gut microbiota associated study, and they also proved that the gut microbiota could normalize within 2

months. Ren Z, et al. reported that the patients included in their study were those who have not used antibiotics for 2 months [1]. In the study by Zhu J et al., the enrollment criteria of patients was that no antibiotics was used for 3 months [2]. Another study also reported that the gut microbiota was resumed 2-3 week after constitutively using 5-day antibiotics [3]. All these findings suggest that the criteria of no antibiotics use within the last 10 weeks is valid in our study. Your suggestions have given us more enlightenment, and we will further tighten our exclusion and inclusion criteria in future research.

2. How were the blood samples chosen for metabolic analysis? Were they plasma or serum and how they were collected? Were patients fasting prior to the collection?

Response: The blood samples for metabolic analysis were chosen from those patients who had been collected stool samples, so that we can have the paired data from blood and stool samples of AML patients. We used blood plasma for metabolic analysis, which were collected from EDTA-stabilized peripheral blood by centrifugation. The patients were not fasting prior to the collection.

3. It is not clear if fecal hydration liquid was pooled from all patients or a subset. If latter, how were the donors selected?

Response: The faecal hydration liquid was pooled from a subset of patients. The information of these patients was shown as below.

	Age(year)	Gender	Positive genes
Patient-1	34	Male	FLT3-ITD, NPM1, IDH2, SRSF2
Patient-2	62	Male	CEBPA
Patient-3	56	Male	FLT3-ITD, NPM1, DNMT3A
Patient-4	47	Female	CBFB-MYH11
Patient-5	49	Female	IDH2, NPM1

4. How were the animals housed? Singly or grouped? If grouped, how were they grouped?

Response: After being purchased, all the mice were randomly divided into different groups and housed in different group cages in an SPF environment according to the requirement of individual experiment, such as being grouped in antibiotic experiment (Fig. 2B-E), FMT experiment (Fig. 3), butyrate experiment (Fig. 5, 6), and other experiments in our study. During the experiment process, all the mice in the same group were housed in a cage without contacting with other group mice.

5. It would be helpful to have a graph showing the mouse experimental design.

Response: Thanks for your helpful suggestion. We reproduced a graph showing the mouse experimental design in the new version (Figure 2B, 3A, 5A).

6. Why was 70% confidence threshold chosen for determining taxonomy of 16S sequences?

Response: According to the associated methods in other published papers [4, 5], we chose 70% confidence threshold for determining taxonomy of 16S sequences. Moreover, inquiring other experienced sequencing companies (Shanghai Majorbio Biopharm Technology Co.,Ltd), they also used 70% confidence threshold.

7. Figure 1C and lines 340-342. Many of the p values shown here are not consistent between the text and the figure. Why? Was the p value corrected for multiple test correction?

Response: Thanks for your careful check. It's our mistake, and we corrected and made them consistent between the text and the figure in the new version. The p values were corrected for multiple test correction.

8. Lines 353-355.: was the correlation performed based on all patients and healthy control samples? If so, it is not surprising to see a strong correlation between

Faecalibacterium and Roseburia, and WBC. Have the authors tried to do the correlation only with samples from patients to further confirm this correlation? It would be more convincing.

Response: The correlation performed in lines 382-384 (new version) was based on only samples from patients. And, there is no significant correlation between Faecalibacterium and Roseburia, and WBC in healthy control samples.

9. Figure 2A - n=3 per group is too small - any justification for the power calculation? How were the animals housed? This is crucial for evaluating the gut microbiota due to cage effects. While there are no significant differences in Sob index at day 0 between the Con and AML, it shows a significant difference in PCoA plot. Why is it? How about Shannon index? The label in the figure is confusing - if control is AML without antibiotics, maybe change them to 'AML_con' and 'AML_abx'. What is the rationale of using antibiotics before MLL-AF9 injection? If the hypothesis is that AML causes the bacterial shift, why not directly inject the cells and monitor the bacterial changes in these animals? It will also be essential to keep the control and AML mice in the same cage to avoid cage-effect. As it is the experiment should have had another control group with antibiotics treatment without AML to determine if the microbial shift is due to the antibiotics or AML. The data may support that the disturbed gut bacterial composition by the antibiotics accelerate AML progression but it does not support that AML causes bacterial dysbiosis.

Response: Thanks very much for your helpful comments and constructive suggestions. Our detailed responses are listed below.

9.1 Figure 2A - n=3 per group is too small - any justification for the power calculation?

For Figure 2A, we performed another 5 mice per group, and added the data in the new Figure 2A. Moreover, we used the statistical program "proc power" of SAS software to calculate the power of Shannon index "n=5 per group" as below. The power is 0.780 when n=5, which means the high credibility.

Power of Shannon index

Power calculates - SAS

Program:

```
proc power;  
twosamplemeans test=diff_satt  
groupmeans=2.7592|3.9444  
groupstddevs=0.6864|0.4244  
ntotal = 10 to 20 by 2  
groupweights=(1 1)  
power=.;  
run;
```

Result:

Fixed Scenario Elements	
Distribution	Normal
Method	Exact
Group 1 Mean	2.7592
Group 2 Mean	3.9444
Group 1 Standard Deviation	0.6864
Group 2 Standard Deviation	0.4244
Group 1 Weight	1
Group 2 Weight	1
Number of Sides	2
Null Difference	0
Nominal Alpha	0.05

Computed Power			
Index	N Total	Actual Alpha	Power
1	10	0.0471	0.780

The total sample size of N total=10, the sample size of each group=10/2=5, at this time power is 0.780

9.2 How were the animals housed? This is crucial for evaluating the gut microbiota due to cage effects.

As mentioned above, all the mice fed in the same cage were randomly divided into different groups and housed in different group cages in an SPF environment according to the requirement of individual experiment. All the mice in the same group were housed in a cage without contacting with other group mice during the experiment process. The reason why we did not mix breeding the different group mice in a same cage is that the mice could eat the stools of other mice in the same cage, which would affect the gut microbiota difference among groups [6, 7]. Therefore, gut microbiota should not be affected due to cage effects.

9.3 While there are no significant differences in Sob index at day 0 between the Con and AML, it shows a significant difference in PCoA plot. Why is it? How about Shannon index?

As the reviewer said “n=3 per group is too small”, this cause the question “While there is no significant differences in Sob index at day 0 between the Con and AML, it shows a significant difference in PCoA plot”. Therefore, we performed another 5 mice per group and added the data in the new Figure 2A. The final result showed that there is no significant difference in PCoA plot at day 0 between the Con and AML. Moreover, the Shannon index and PCoA of a weighted UniFrac distance analysis were decreased in AML group compared with control group at day 14, indicating the decreased gut microbiota diversity.

9.4 the label in the figure is confusing - if control is AML without antibiotics, maybe change them to ‘AML_con’ and ‘AML_abx’.

In Figure 2A, the control group is the healthy mice without any treatment, and AML group is mice injected with AML cells without other treatment. Therefore, we cannot change them to “AML_con” and “AML_abx”. To make it not to be confusing, we labeled it clearly in the legend.

9.5 What is the rationale of using antibiotics before MLL-AF9 injection? If the hypothesis is that AML causes the bacterial shift, why not directly inject the cells and monitor the bacterial changes in these animals?

Thank you for your careful review. Our purpose of this part is to clarify the correlations between AML and bacterial dysbiosis. Firstly, for the effect of AML on bacteria, as you suggested, we directly injected AML cells without antibiotics and monitored the bacterial changes in these animals, and found that AML caused bacterial shift (Figure 2A). Secondly, for the effect of bacteria on AML, we ablated the gut microbiota using antibiotics before MLL-AF9 injection, and found that destroying the diversity of the gut microbiota accelerated AML progression (Figure 2B-E). Therefore, we conclude that AML causes bacterial dysbiosis and gut microbiota dysbiosis aggravate the progress of AML.

9.6 It will also be essential to keep the control and AML mice in the same cage to avoid cage-effect.

Thanks a lot for your comment. As mentioned above, the reason why we did not keep the control and AML mice in the same cage is that the mice could eat the stools of other mice in the same cage, which would affect the gut microbiota difference among groups [6, 7]. Therefore, we raised different group mice in different cages and feed with the same diet and drink at the same SPF environment.

9.7 As it is the experiment should have had another control group with antibiotics treatment without AML to determine if the microbial shift is due to the antibiotics or AML. The data may support that the disturbed gut bacterial composition by the antibiotics accelerate AML progression but it does not support that AML causes bacterial dysbiosis.

Thanks for your comment. We have included the control group with antibiotics treatment without AML which was compared with healthy mice, and the result showed that the antibiotics could shift the microbiota (supplementary figure 2a). We have also included mice group without antibiotics treatment with AML which was compared with healthy mice, and the result showed that AML could shift the microbiota (Figure 2A). These two results demonstrated that either antibiotics or AML could cause the bacteria dysbiosis. Moreover, we included the control group without antibiotics treatment with AML which was compared with that with antibiotics treatment with AML, and the result showed that the antibiotics accelerate AML progression (Figure 2B-E).

10. Line 381, "Faecalibacterium correlated with a favorable prognosis" this correlation should be done within patients rather than with the inclusion of healthy control. Which Faecalibacterium was used? Isolated from patients and cultured or directly use the fecal slurry? If the latter, it is not a pure Faecalibacterium FMT and wording should be changed. Again, could n=4 per group give a meaningful result?

Response: Thank you for your careful review. In our study, the correlation between Faecalibacterium and prognosis has been done within patients rather than with the

inclusion of healthy control. We declared it in the new version (line 433-434). The Faecalibacterium we used in the study was Faecalibacterium prausnitzii. Faecalibacterium prausnitzii is pure standard strain, not isolated from patients and cultured or directly used the faecal slurry. The individual point of weight or leukaemia cell frequency in the same group (n=4 per group) was tightly clustered but not scattered, which indicated a meaningful result (Figure 3C).

Moreover, we performed new murine FMT experiment with faeces from AML patients (AML-FMT), supplemented with Faecalibacterium (n=5). Our results showed that spleen sizes and the leukaemia cells frequencies in bone marrow and spleen were significantly reduced in AML-FMT mice supplemented with Faecalibacterium compared with AML-FMT mice without Faecalibacterium. The leukaemia cells frequencies in peripheral blood also had a decreased trend, but did not reach significant difference (Supplementary figure 3e-g). All these results demonstrated that Faecalibacterium could decrease the leukaemia burden in AML mice, but the inhibiting effect of single Faecalibacterium supplement was weaker than the transplantation of an intact gut microbiota. The inhibitory effect of Faecalibacterium on AML relies on an intact gut microbiota.

11. Figure S4a, was before con-FMT time point after antibiotics? If so, this is not surprising. It would be helpful to show health control donors, AML patient donors, and after AML FMT and after Con-FMT in one figure. Again, the labels are confusing. Since Control has been used to described healthy control patients, it would be good to use a different abbreviation, e. g. AML_con or AML_nab (no antibiotics). Was it one donor corresponding to one recipient? If so it should be good to label the pair in the PCoA plot.

Response: In Figure S4a, the “before con-FMT” time point was before antibiotics. The left figure is shown the bacteria PCoA results of FMT in which faeces from healthy people: PCoA results from healthy people (red points), from AML mice before FMT with healthy people faeces (blue triangles), from AML mice after FMT with healthy people faeces (green diamonds). The right figure is shown the bacteria PCoA results of FMT in which faeces from AML patients: PCoA results from AML patients (red points), from AML mice before FMT with AML patients’ faeces (blue triangles), from AML mice after FMT with AML patients’ faeces (green diamonds). To make it clearer, we

changed the labels with “Healthy people, AML mice before con-FMT, AML mice after con-FMT” and “AML patients, AML mice before AML-FMT, AML mice after AML-FMT”. It is not one donor corresponding to one recipient, and we did not label the pair in the PCoA plot.

12. Figure 4a, use cross validated scores plot instead of OPLS-DA scores plot. Which sample type is this plot based on? Lines 420-422, were the MS/MS patterns matched with the database?

Response: Thanks for your comment. We have used cross validated scores plot along with OPLS-DA scores plot in Figure 4a. In detail, the cross validated randomly changes the arrangement order of the categorical variable Y, and establishes 200 OPLS-DA models to obtain the R2 and Q2 values of the random model. The abscissa represents the permutation retention of the permutation test, and the ordinate represents the value of R2 or Q2. The Q2 point of is lower than the original blue Q2 point on the far right from left to right, indicating that the model is robust and reliable without overfitting [8]. This plot was based on the blood plasma. The MS/MS patterns were matched with the database in the new version (lines 485-488).

13. Line 422-424, in the method, it mentioned that SCFAs were analysed using GC-MS but here it indicated as LC-MS. Which one was used? Figure 4D, again can this correlation be performed in patient group and healthy control groups, separately?

Response: Thanks a lot for your careful review. For SCFAs determination in our study, we first screened out significantly-different propionic acid and butyric acid by LC-MS, then used targeted SCFAs metabolisms to quantitatively determine the content of SCFAs by GC-MS. We are sorry only introduced GC-MS in the method, and now we added the method of LC-MS in the new version (Line 234-249).

According to your suggestion, we performed this correlation in patient group and healthy control group, separately. The results are shown below. The Faecalibacterium was positively correlated with butyric acid in both AML patient group and healthy control group, and more obviously in AML patients. Moreover, in our new experiments, we proved that Faecalibacterium produces butyric acid in vitro and in vivo (Line 500-510, Supplementary figure 5 g-h).

Spearman Correlation Heatmap

Spearman Correlation Heatmap

14. Figure 4F keep the color of the groups consistent throughout the manuscript. Were these p values corrected for multiple test correction?

Response: Thanks for your valuable suggestion. We keep the color of the groups in Figure 4F consistent throughout the manuscript. These p values were corrected for multiple test correction.

15. All the data including metabolic and 16S rRNA sequencing raw data should be made publicly available via data repositories.

Response: We submitted all the data including metabolic and 16S rRNA sequencing

raw data to data repositories. All metabolic raw datasets were uploaded to the metabolights database, accession: MTBLS3092, and 16s rRNA sequencing datasets were uploaded to the NCBI database with SRA, accession: PRJNA746137.

Minor:

Line 31: "...observed a significant shift..." was it in feces or gut content? Clarify here.

Response: We clarified it as "Here, we observed a significant shift in the gut microbiota in the faeces of AML patients".

Line 37-38 "Metabolome analyses showed that microbiota derived butyrate concentration obviously decreased in AML patient feces" should be written "Metabolic analyses showed that fecal concentrations of butyrate, produced by the gut microbiota, significantly decreased in AML patients,"

Response: Thank you. We have replaced with "Metabolic analyses showed that fecal concentrations of butyrate, produced by the gut microbiota, significantly decreased in AML patients,".

Line 40, add comma before 'which'

Response: Thank you. We added comma before "which" in line 41(new version).

Line 89, replace "GVHD patients" to "patients with GVHD"

Response: Thank you. We have replaced with "patients with GVHD" in line 91(new version).

Lines 91 and 343, replace 'flora' with 'bacteria' r' microbiota'

Response: Thank you. We have replaced with "bacteria" or "microbiota" in line 93 and 376 (new version).

Line 96, should be 'disordered'

Response: Thank you. We have changed to “disordered” in line 98 (new version).

Line 96-97 the past tense should be used.

Response: Thank you. We have changed to the past tense in line 98-99 (new version).

Line 117, another 28 samples - which sample type?

Response: The 28 samples were blood and faeces.

Line 130, CO₂ where 2 should be subscripted.

Response: Thank you. We have subscripted 2 of CO₂.

Line 157-158: does it mean that “fresh feces from patients or healthy control were weighted and diluted with normal saline to achieve a final concentration of 150 mg/mL.”?

Response: Yes. We replaced with “Fresh feces from patients or healthy control were weighted and diluted with normal saline to achieve a final concentration of 150 mg/mL.”

Line 159, what filter was used?

Response: We used 40 µm sieve to filter the faecal suspension.

Line 182: *in vivo* should be italicized

Response: Thank you. We have made “*in vivo*” italicized.

Line 214, which sample types were used to measure SCFAs?

Response: We used faeces and blood plasma to measure SCFAs.

Line 278 H₂O where 2 should be subscripted.

Response: Thank you. We have subscripted 2 of H₂O.

Line 326, should it be 'all fecal samples' rather than 'all samples'?

Response: Thank you. We have replaced with "all faecal samples".

Figure 1B, variations should be labeled clearer, not overlapping with the axis.

Response: We relabeled the variations to make it clearer and no overlapping.

Line 336 should change microbiota to bacteria.

Response: Thank you. We have changed to "bacteria".

Line 338 add 'compared to the healthy control' after 'AML patients'

Response: Thank you. We have added "compared to the healthy controls" after "AML patients".

Line 338 remove 'approximately'

Response: Thank you. We have removed "approximately".

Figure 1D, change blue to red for positive correlation.

Response: Thank you. We have changed blue to red for positive correlation.

Line 347 change 'depleted' to 'lower' and add 'compared to control group'.

Response: Thank you. We have changed "depleted" to "lower" and add "compared to control group".

Figure S1A, why are there 72 samples?

Response: There should be 61 samples, consistent with the result of manuscript. We are sorry for this mistake. We corrected it in the new Figure S1A.

Line 357 change 'disordered' to 'low'

Response: Thank you. We have changed "disordered" to "low".

Line 356, please elaborate on 'favorable-risk'. How was favorable and non-favorable defined.

Response: We elaborate the definition of favorable and non-favorable in the new version according to NCCN guideline (supplementary table 2).

Line 363 should be 16S rRNA

Response: Thank you. We have replaced with "16S rRNA".

Reference

1. Ren Z, Li A, Jiang J, et al. Gut microbiome analysis as a tool towards targeted non-invasive biomarkers for early hepatocellular carcinoma. *Gut*. 2019;68(6):1014-1023. doi:10.1136/gutjnl-2017-315084
2. Zhu J, Liao M, Yao Z, et al. Breast cancer in postmenopausal women is associated with an altered gut metagenome. *Microbiome*. 2018;6(1):136. Published 2018 Aug 6. doi:10.1186/s40168-018-0515-3
3. Ng KM, Aranda-Díaz A, Tropini C, et al. Recovery of the Gut Microbiota after Antibiotics Depends on Host Diet, Community Context, and Environmental Reservoirs [published correction appears in *Cell Host Microbe*. 2020 Oct 7;28(4):628]. *Cell Host Microbe*. 2019;26(5):650-665.e4. doi:10.1016/j.chom.2019.10.011
4. Zhong L, Lai CY, Shi LD, et al. Nitrate effects on chromate reduction in a methane-based biofilm. *Water Res*. 2017;115: 130-137. doi:10.1016/j.watres.2017.03.003
5. Quast C, Pruesse E, Yilmaz P, et al. The SILVA ribosomal RNA gene database

project: improved data processing and web-based tools. *Nucleic Acids Res.* 2013;41(Database issue): D590-D596. doi:10.1093/nar/gks1219

6. Wang X, Sun G, Feng T, et al. Sodium oligomannate therapeutically remodels gut microbiota and suppresses gut bacterial amino acids-shaped neuroinflammation to inhibit Alzheimer's disease progression. *Cell Res.* 2019;29(10):787-803. doi:10.1038/s41422-019-0216-x
7. Caruso R, Ono M, Bunker ME, Núñez G, Inohara N. Dynamic and Asymmetric Changes of the Microbial Communities after Cohousing in Laboratory Mice. *Cell Rep.* 2019;27(11):3401-3412.e3. doi: 10.1016/j.celrep.2019.05.042
8. Lu L, Li H, Wu X, et al. HJC0152 suppresses human non-small-cell lung cancer by inhibiting STAT3 and modulating metabolism. *Cell Prolif.* 2020;53(3): e12777. doi:10.1111/cpr.12777

REVIEWER COMMENTS

Reviewer #1 (Remarks to the Author):

The major claim of this paper is that the alteration in gut microbiota plays an important pathogenic role in the progression of AML, mainly that observed decrease in microbial (MB) diversity contributes to AML progression. In response to the various concerns raised by the reviewers, the investigators have performed number of new experiments and added additional data to address the many issues that were raised, especially as it relates to the cause-and-effect relationship. The manuscript has been improved by these additions. The strengths of the manuscript relates to the detailed series of experiment to demonstrate the potential role of MB alteration in the pathogenesis of AML and delineating the various components, including gut bacteria, Faecalibacterium, butyrate, intestinal barrier, and LPS, that mediate the progression on AML. A particularly important strength exists as it relates to the potential therapeutic significance of gut MB modulation and short chain fatty acids for the treatment of AML. Although there are number of important strengths, there are some weaknesses or concerns that persist, including the overstatement of the importance and novelty of these studies and not recognizing other works related to this topic (e.g. prior works related to AML and MB, butyrate and intestinal barrier), the lack of discussion of the recently published paper on AML and gut MB (Yu, FEBS) and the contradictory findings related to MB diversity, and the studies related to the use of antibiotics and its potential effects on the gut barrier and LPS generation. Overall, strengths of this manuscript outweigh the weaknesses.

Specific concerns or points to consider:

1. Although the investigators provide some supporting data about other investigators using this model system to study AML, it still remains unclear to this reviewer how valid the animal model used in this study relates to the human AML.
2. As noted by the reviewer 3 and acknowledged by the investigators, there are published studies suggesting the involvement of butyrate and gut barrier in AML. However, the investigators continue to overstate the novelty of their studies throughout.
3. Recent study by Yu et al (FEBS) suggest that the bacterial diversity is increased in AML and they did not report Faecalibacterium to be depleted. How do the investigators reconcile these differences since this is the major premise of their studies.
4. In response to the role of Faecalibacterium, investigators studied the effects of adding *F. prausnitzii* in butyrate generation and on AML. What was the rationale for using this species- was this based on experimental data on species abundance.
5. Although intestinal damage of AML is assumed by the investigators as an important contributing mechanism in AML progression, no data or histology demonstrating the existence of intestinal damage is provided. If present, it remains unclear if this is a driving etiologic factor or a secondary finding.
6. There continues to be some concern related to the role of intestinal barrier dysfunction and LPS. An important focus of these studies relate to the pathogenic role of gut barrier defect and increased intestinal penetration of LPS and the subsequent AML promoting effects of LPS. The investigators utilized antibiotic treatment to demonstrate that the depletion of bacterial flora contributes to the development of AML. One potential problem with this approach is that the antibiotic treatment causes a dramatic eradication of all bacteria and is not a good mimic of AML associated change in MB diversity. Since the antibiotic cocktail used kills all Gram negative (GN) bacteria in the mouse gut, the primary source of LPS, it is difficult to reconcile that LPS would be the primary bacterial derived factor responsible for the AML progression following antibiotic treatment. If LPS was the primary pathogenic factor as suggested, it would follow that antibiotic depletion of GN bacteria should be protective against AML progression. The small increase in LPS concentration reported in AML patients is highly unlikely to be responsible promoting AML- this topic has been well-studied previously. In general, the

antibiotic depletion of gut bacteria has been shown to be protective against barrier disruption by variety of agents (including NSAIDs) and in various animal models of intestinal inflammation by depleting the gut bacteria and bacterial derived factors, the current findings with antibiotic therapy contradicts the protective role of gut bacteria depletion (especially the GN bacteria), how do the investigators explain this potentially discrepant impact of gut MB.

Reviewer #2 (Remarks to the Author):

the authors addressed my concerns

Reviewer #3 (Remarks to the Author):

The authors performed quite amount of experiments for the revision of current manuscript, which improves the quality of their story. However, the reviewer still have concerns for their responses to reviewer#3's major points 3 and 4:

For response to point 3:

The authors claimed that it is the bacteria that survived the antibiotics treatment produces LPS. However, antibiotics should depletes most of gut microbiota and therefor, there should be several logs magnitude drop of bacterial amount and thus LPS. Comparison of LPS with and without antibiotics treatment should be performed to support their hypothesis. The authors also claimed that antibiotics damages gut epithelial integrity. Under the condition of leukaemia, antibiotics effects on gut epithelial integrity should be minimal. Comparison of gut epithelial integrity in leukaemia mice with or without antibiotics treatment should be performed.

For response to point 4:

The authors still have NOT answer the concern raised by the reviewer: the differences between treated and control groups are so subtle (e.g. Fig.2C with less than 3% difference) that it affects the biological significance of current study.

Reviewer #4 (Remarks to the Author):

The authors have sufficiently addressed the majority of the comments. There are a few remaining ones:

1. Point 2 regarding the blood collection. The blood metabolome is affected by the diet. In the methodology, it should be made clear that the blood samples were collected without fasting.
2. Point 3 why were these patient selected and the justification should be added into the Methodology.
3. The gut microbiota from mice housed in the same cage tends to be similar compared to that of mice from different cages. As authors explained, the mice were housed based on the experiments/grouping, which could induce bias to the results. Linking with Point 9.6, in my opinion, animals should be singly housed.
4. Point 9.1 I understand the authors carried out the experiments with another 2 mice per group and added the data together to make a n=5. Is it correct?

Point by point responses to reviewers' comments

We appreciate the reviewers for their thoughtful and constructive comments and suggestions regarding our manuscript. We have addressed all of the questions and concerns through additional experimentation and clarification. The detailed point-by-point responses in blue font are given below.

REVIEWER COMMENTS

Reviewer #1 (Remarks to the Author):

The major claim of this paper is that the alteration in gut microbiota plays an important pathogenic role in the progression of AML, mainly that observed decrease in microbial (MB) diversity contributes to AML progression. In response to the various concerns raised by the reviewers, the investigators have performed number of new experiments and added additional data to address the many issues that were raised, especially as it relates to the cause-and-effect relationship. The manuscript has been improved by these additions. The strengths of the manuscript relate to the detailed series of experiment to demonstrate the potential role of MB alteration in the pathogenesis of AML and delineating the various components, including gut bacteria, Faecalibacterium, butyrate, intestinal barrier, and LPS, that mediate the progression on AML. A particularly important strength exists as it relates to the potential therapeutic significance of gut MB modulation and short chain fatty acids for the treatment of AML. Although there are number of important strengths, there are some weaknesses or concerns that persist, including the overstatement of the importance and novelty of these studies and not recognizing other works related to this topic (e.g. prior works related to AML and MB, butyrate and intestinal barrier), the lack of discussion of the recently published paper on AML and gut MB (Yu, FEBS) and the contradictory findings related to MB diversity, and the studies related to the use of antibiotics and its potential effects on the gut barrier and LPS generation. Overall, strengths of this manuscript outweigh the weaknesses.

Response: Thank you very much for your overall positive comments and constructive suggestions. As you suggested, we modified our manuscript to soften the importance and novelty of our studies, included more references of other works related to this topic, and added the explanation of the contradictory findings related to MB diversity in the

discussion part. Moreover, we performed new experiments to further clarify the use of antibiotics and its potential effects on the gut barrier and LPS generation. Our detailed responses are listed below.

Specific concerns or points to consider:

1. Although the investigators provide some supporting data about other investigators using this model system to study AML, it still remains unclear to this reviewer how valid the animal model used in this study relates to the human AML.

Response: Thanks a lot for your comment. The animal model we used is valid as it relates to the human AML. Firstly, in terms of pathogenesis, the murine AML model we used in our manuscript was established by over-expressing MLL-AF9 oncogene in murine haematopoietic stem cells through retrovirus vector transfection, which is similar to human AML pathogenesis in which MLL-AF9 is overexpressed in human haematopoietic stem cells [1, 2]. Secondly, in terms of biology, murine leukaemia cells highly expressed myeloid leukaemia markers Gr-1 and Mac-1 which were also found higher in AML patients, and AML cells from murine MLL-AF9 model and human AML patients activated the same down signaling pathway signal molecules, such as Hoxa9 and Meis1 [2, 3]. Additionally, the genes related to prognosis and invasion, such as Erg and Evil, were simultaneously increased both in MLL-AF9 murine model and AML patients, which indicated that MLL-AF9 murine model may be similar to AML patients as related to the disease progression and invasion [5]. Thirdly, in terms of symptoms, the murine AML model we used presents typical symptoms of AML disease with extensive infiltration of leukaemia cells in bone marrow, peripheral blood, and spleen, which is similar to AML patients [2]. In addition, due to the reliable, safe and simple nature, this murine AML model becomes the most frequently in vivo model to study the biology of human AML [3]. There are many studies which used this murine AML model to investigate the clinical drug screening, treatment responses and other applications related to human AML [4].

2. As noted by the reviewer 3 and acknowledged by the investigators, there are published studies suggesting the involvement of butyrate and gut barrier in AML. However, the investigators continue to overstate the novelty of their studies throughout.

Response: Thanks for your comment. With your suggestion, we modified our manuscript to soften the claim of novelty and included other reference of other works related to this topic (line 106, line 590-592, line 647-648, line 690-691; reference 48)

3. Recent study by Yu et al (FEBS) suggest that the bacterial diversity is increased in AML and they did not report Faecalibacterium to be depleted. How do the investigators reconcile these differences since this is the major premise of their studies.

Response: Thanks a lot for your comment. During the revised process of our manuscript, Yu et al published their study about the profile of gut microbial dysbiosis in adults with myeloid leukaemia in *FEBS Open Bio* on Jun 24, 2021. We read this paper carefully and found that their results are what the reviewer said “the bacterial diversity is increased in AML and they did not report Faecalibacterium to be depleted.” We retrieved many associated references to reconcile these differences between our and Yu’s results. Firstly, many papers reported that regional differences could significantly affect the gut bacterial diversity [6,7]. The AML patients in the study by Yu et al live in Southern China, where the weather was hot and the dietary habit was sweet food; otherwise, our AML patients live in Northern China, 1300 km away from Yu et al, where the weather was cold and the dietary habit was salty food. These differences might lead to regional-dependent or diet-dependent changes in gut microbiota. Secondly, the gender is a very important factor to affect the bacterial diversity, which has been reported by other researches [8,9]. The gender ratio in our study is 15 males to 16 females while it is 19 males to 10 females in the study by Yu et al, which might cause the differences between our and Yu’s results. We included the explanation in the discussion part as below “Recent study by Yu et al³⁵ reported that the bacterial diversity was increased and no depletion of Faecalibacterium was found in AML, which is different from our results. These differences may be due to the regional and gender diversities which have been reported by other studies³⁶⁻³⁹”. (Line 666-670, reference 35-39).

Moreover, we guarantee ours are the true results. The standard for selecting patients and controls and the determination of bacterial diversity are rigorous in our study. The detailed information of our patients is shown as the following table and supplementary table 1, and the results of gut bacteria can be accessed via the link “<https://www.ncbi.nlm.nih.gov/sra/?term=PRJNA746137>”.

Furthermore, our animal experiments proved that the gut microbiota diversity of AML mice decreased significantly compared with that of healthy control mice (Figure 2A), and recovering the WT gut microbiota of AML mice via oral gavage of faecal suspension from normal mice reduced the leukaemia burden in AML mice (Figure 2G). All these data make our results convincing.

The detailed information of AML patients in our study

No.	gender	age	FAB subtype	Faecalibacterium abundance	OUT number
AML1	male	65	M5	140	60797
AML2	male	46	other	1966	57528
AML3	female	31	M5	3475	52087
AML4	female	63	M4b	7856	52514
AML5	female	55	M5	21948	54088
AML6	female	26	M3	22	53956
AML7	female	44	M5	9303	62032
AML8	female	69	M5	317	34971
AML9	female	60	M4a	4292	42511
AML10	female	46	M3	2847	54822
AML11	female	68	M5	5	54014
AML12	male	15	other	15166	54105
AML13	male	50	M5	1200	60173
AML14	male	31	M3	8	52516
AML15	male	43	M3	34	53792
AML16	female	55	M5	226	54891
AML17	female	46	M5	25465	101106
AML18	female	24	M5	5926	36266
AML19	female	57	M3	4787	59597
AML20	male	30	M2	5783	52264
AML21	male	51	M5	1166	35061
AML22	female	47	M5	420	45058
AML23	male	69	other	1631	41358

AML24	male	30	M5	708	36150
AML25	male	55	M5	2823	59061
AML26	male	48	M2	566	50581
AML27	male	33	M3	976	53822
AML28	female	23	M4b	71	42220
AML29	female	53	M3	132	33509
AML30	male	46	M1	8866	50342
AML31	male	45	other	9636	50520

4. In response to the role of Faecalibacterium, investigators studied the effects of adding *F. prausnitzii* in butyrate generation and on AML. What was the rationale for using this species- was this based on experimental data on species abundance.

Response: Thank you for your comment. The rationale for using *F. prausnitzii* was based on the species abundance in our experimental data. As shown below, at the species level, the abundance of *F. prausnitzii* was TOP 1 in control, and significantly decreased in AML ($p=0.00042$). Therefore, we used *F. prausnitzii* to study the effect of *Faecalibacterium* in butyrate generation and on AML. We included this result in the new version (Line 434-437; Supplementary figure 2b).

5. Although intestinal damage of AML is assumed by the investigators as an important contributing mechanism in AML progression, no data or histology demonstrating the existence of intestinal damage is provided. If present, it remains unclear if this is a driving etiologic factor or a secondary finding.

Response: Thanks a lot for your comment. The indexes of intestinal damage mainly include the histology changes (microstructure level and ultrastructure level), molecular changes, and functional changes. For histology changes of ultrastructure level detected by transmission electron microscopy, our results demonstrated the existence of intestinal damage in AML mice (Figure 6D). For molecular levels by quantitative reverse transcriptase PCR, Western blot analysis, and immunofluorescence assay, our results demonstrated the molecular changes of intestinal damage (Figure 6B, C, E). For functional changes by FITC-dextran assay, our results showed that the intestinal permeability to FITC-dextran was significantly increased in AML mice, suggesting leukaemia-induced intestinal damage (Figure 6A). Moreover, we performed new murine experiments to determine the intestinal microstructure histology in AML mice using hematoxylin-eosin staining under optical microscope. The results showed that there was no apparent microstructure change except the intestinal recess became a little shallow in AML mice. Another new experiment using Ki67 staining showed that the Ki67 expression of intestinal epithelial cells in bottom of intestinal recess was lower in AML mice, suggesting the inhibition of intestinal epithelial cells proliferation in AML mice. We included the new results (line 560-565) and Supplementary figure 6e and 6f in the new version.

In addition, all of our results showed that the AML mice were present with intestinal damage compared with control mice, which indicates AML could cause intestinal damage and intestinal damage was a secondary finding of AML (Figure 6). Furthermore, as butyrate is mainly absorbed by intestinal epithelial cells as an energy source, which promotes intestinal barrier integrity, we used butyrate to repair intestinal damage (Figure 6). After the recovery of intestinal damage, we found the progression of AML was inhibited (Figure 5), which indicates that intestinal damage could be a promoting factor for AML progression.

6. There continues to be some concern related to the role of intestinal barrier dysfunction and LPS. An important focus of these studies relates to the pathogenic role

of gut barrier defect and increased intestinal penetration of LPS and the subsequent AML promoting effects of LPS. The investigators utilized antibiotic treatment to demonstrate that the depletion of bacterial flora contributes to the development of AML. On potential problem with this approach is that the antibiotic treatment causes a dramatic eradication of all bacteria and is not a good mimic of AML associated change in MB diversity. Since the antibiotic cocktail used kills all Gram negative (GN) bacteria in the mouse gut, the primary source of LPS, it is difficult to reconcile that LPS would be the primary bacterial derived factor responsible for the AML progression following antibiotic treatment. If LPS was the primary pathogenic factor as suggested, it would follow that antibiotic depletion of GN bacteria should be protective against AML progression. The small increase in LPS concentration reported in AML patients is highly unlikely to be responsible promoting AML- this topic has been well-studied previously. In general, the antibiotic depletion of gut bacteria has been shown to be protective against barrier disruption by variety of agents (including NSAIDs) and in various animal models of intestinal inflammation by depleting the gut bacteria and bacterial derived factors, the current findings with antibiotic therapy contradicts the protective role of gut bacteria depletion (especially the GN bacteria), how do the investigators explain this potentially discrepant impact of gut MB.

Response: We thank you very much for your valuable and constructive comments. We performed a number of new experiments, and the detailed responses are listed below.

a. There continues to be some concern related to the role of intestinal barrier dysfunction and LPS. An important focus of these studies relates to the pathogenic role of gut barrier defect and increased intestinal penetration of LPS and the subsequent AML promoting effects of LPS. The investigators utilized antibiotic treatment to demonstrate that the depletion of bacterial flora contributes to the development of AML. On potential problem with this approach is that the antibiotic treatment causes a dramatic eradication of all bacteria and is not a good mimic of AML associated change in MB diversity.

Response: Thanks a lot for your valuable comment. In our study, we first ablated the gut microbiota in mice with antibiotic treatment, then stopped antibiotics and injected leukaemia cells into mice to evaluate their AML progression. The ablated gut

microbiota would be recovered during the AML process [10]. Moreover, to reduce the gut microbiota eradication degree caused by antibiotics and better mimics the gut microbiota dysbiosis *in vivo*, we performed new experiments in which we applied mice with antibiotics only for 1 week and stopped for 3 days, and then established AML mice, which shortened the duration of antibiotics treatment (new Figure 2B). The 16S rRNA results showed that the diversity and composition of the gut microbiota were still decreased but higher than previous data (new Supplementary figure 2a), and the FACS and histology results showed that the percentage of leukaemia cells was much higher in antibiotic-treated AML mice than in control PBS AML mice (new Figure 2C, D). We replaced these figures in the new version. In addition, to mimic the AML-associated change of MB diversity *in vivo*, FMT was performed via gavaging faecal hydration liquid of AML patients to clarify the causal relationship between the gut microbiota imbalance and AML progression, while gavaging faecal hydration liquid of healthy controls served as control.

b. Since the antibiotic cocktail used kills all Gram negative (GN) bacteria in the mouse gut, the primary source of LPS, it is difficult to reconcile that LPS would be the primary bacterial derived factor responsible for the AML progression following antibiotic treatment. If LPS was the primary pathogenic factor as suggested, it would follow that antibiotic depletion of GN bacteria should be protective against AML progression.

Response: Thank you very much for your valuable comment. Truly, the antibiotic cocktail could kill Gram negative bacteria in the mouse gut, but the amount of live bacteria was not equal to the amount of LPS at some time. Many studies reported that LPS concentration in blood is sharply increased after using antibiotics, as the bacteria lysis caused by antibiotics releases much bacteria components including LPS into blood [11,12]. For this reason, we performed additional experiments to determine the change tendency of bacteria diversity or LPS concentration in blood after using antibiotics. As shown below, the bacteria diversity of AML mice was gradually decreased after using antibiotics treatment. However, the blood LPS concentration was increased after using antibiotics, reached the peak at 48h, and remained in high level even at 144h. Therefore, the change trend was inconsistent between bacteria diversity and LPS concentration after using antibiotics, and contrary at some time. Therefore, usage of antibiotics in short time could induce the increase of LPS in blood and maybe decrease LPS

concentration in a long-time usage of antibiotics. We included this result in the new version (Line 608-611; Supplementary figure 7).

the bacteria diversity and abundance after using antibiotics

The blood LPS concentration after using antibiotics

c. The small increase in LPS concentration reported in AML patients is highly unlikely to be responsible promoting AML- this topic has been well-studied previously.

Response: Thank you for your valuable comment. Truly, in our study, LPS concentration in AML patients was higher than that in healthy controls (mean 113.5 pg/ml vs 65.4 pg/ml, $p=0.0017$), and the increase of LPS concentration is not very high.

Though in several *in vitro* studies [13, 14], LPS played important roles in AML at higher concentrations than our detected concentrations, whether the long-term small increase of LPS concentration in AML patients play vital effect is unknown, which is a limitation of our study and needs future study. We included this limitation in the discussion part of our new version “Though our results demonstrated LPS promoted AML progression in the experiments of *in vitro* cell culture and murine model, whether the small increase in LPS concentration play vital effect in AML patients is unknown, which is a limitation of our study and needs future study.” (Line 712-715).

d. In general, the antibiotic depletion of gut bacteria has been shown to be protective against barrier disruption by variety of agents (including NSAIDs) and in various animal models of intestinal inflammation by depleting the gut bacteria and bacterial derived factors, the current findings with antibiotic therapy contradicts the protective role of gut bacteria depletion (especially the GN bacteria), how do the investigators explain this potentially discrepant impact of gut MB.

Response: Thank you for your comment. In inflammatory diseases using variety of agents (including NSAIDs) or in various animal models of intestinal inflammation, the intestinal barrier is in the condition of inflammatory overreaction in which the bacteria or the bacterial derived factors are excessive. These excessive gut bacteria and bacterial derived factors were harmful and aggravated the diseases. Therefore, the antibiotic depletion of gut bacteria in these diseases or conditions has been shown to be protective by inhibiting the over-activated intestinal inflammation [15,16]. In some other diseases, the intestinal barrier was not in inflammatory hyperactivation state, and the gut bacteria and bacterial derived factors play protective roles; therefore, the antibiotic depletion of gut bacteria in these diseases would cause further dysbacteriosis and aggravate the diseases [17, 18]. The disease of AML should belong to the latter condition.

Reference

1. Wei J, Wunderlich M, Fox C, et al. Microenvironment determines lineage fate in a human model of MLL-AF9 leukemia. *Cancer Cell*. 2008;13(6):483-495. doi: 10.1016/j.ccr.2008.04.020
2. Xu SM, Yang Y, Zhou M, et al. *Zhongguo Shi Yan Xue Ye Xue Za Zhi*. 2013;21(5):1126-1132. doi: 10.7534/j.issn.1009-2137.2013.05.008

3. Schwaller J. Learning from mouse models of MLL fusion gene-driven acute leukemia. *Biochim Biophys Acta Gene Regul Mech.* 2020;1863(8):194550. doi: 10.1016/j. bbagrm. 2020.194550
4. Stubbs MC, Krivtsov AV. Murine Retrovirally-Transduced Bone Marrow Engraftment Models of MLL-Fusion-Driven Acute Myelogenous Leukemias (AML). *Curr Protoc Pharmacol.* 2017 Sep 11; 78:14.42.1-14.42.19. doi: 10.1002/cpph.28. PMID: 28892146;
5. Stavropoulou V, Peters AHFM, Schwaller J. Aggressive leukemia driven by MLL-AF9. *Mol Cell Oncol.* 2017;5(3): e1241854. Published 2017 Oct 23. doi:10.1080/23723556.2016.1241854
6. Adak A, Khan MR. An insight into gut microbiota and its functionalities. *Cell Mol Life Sci.* 2019;76(3):473-493. doi:10.1007/s00018-018-2943-4
7. Fontana A, Panebianco C, Picchianti-Diamanti A, et al. Gut Microbiota Profiles Differ among Individuals Depending on Their Region of Origin: An Italian Pilot Study. *Int J Environ Res Public Health.* 2019;16(21):4065. Published 2019 Oct 23. doi:10.3390/ijerph16214065
8. Rizzetto L, Fava F, Tuohy KM, Selmi C. Connecting the immune system, systemic chronic inflammation and the gut microbiome: The role of sex. *J Autoimmun.* 2018; 92:12-34. doi: 10.1016/j.jaut.2018.05.008
9. Santos-Marcos JA, Haro C, Vega-Rojas A, et al. Sex Differences in the Gut Microbiota as Potential Determinants of Gender Predisposition to Disease. *Mol Nutr Food Res.* 2019;63(7): e1800870. doi:10.1002/mnfr.201800870
10. Ng KM, Aranda-Díaz A, Tropini C, et al. Recovery of the Gut Microbiota after Antibiotics Depends on Host Diet, Community Context, and Environmental Reservoirs [published correction appears in *Cell Host Microbe.* 2020 Oct 7;28(4):628]. *Cell Host Microbe.* 2019;26(5):650-665.e4. doi: 10.1016/j.chom.2019.10.011
11. Van Langevelde P, Kwappenberg KM, Groeneveld PH, Mattie H, van Dissel JT. Antibiotic-induced lipopolysaccharide (LPS) release from *Salmonella typhi*: delay between killing by ceftazidime and imipenem and release of LPS. *Antimicrob Agents Chemother.* 1998;42(4):739-743. doi:10.1128/AAC.42.4.739
12. Wang C, Schaefer L, Bian F, et al. Dysbiosis Modulates Ocular Surface Inflammatory Response to Liposaccharide. *Invest Ophthalmol Vis Sci.* 2019;60(13):4224-4233. doi:10.1167/iovs.19-27939

13. Laouedj M, Tardif MR, Gil L, et al. S100A9 induces differentiation of acute myeloid leukemia cells through TLR4. *Blood*. 2017;129(14):1980-1990. doi:10.1182/blood-2016-09-738005
14. Baakhlagh S, Kashani B, Zandi Z, et al. Toll-like receptor 4 signaling pathway is correlated with pathophysiological characteristics of AML patients and its inhibition using TAK-242 suppresses AML cell proliferation. *Int Immunopharmacol*. 2021; 90:107202. doi: 10.1016/j.intimp.2020.107202
15. Li X, He C, Li N, et al. The interplay between the gut microbiota and NLRP3 activation affects the severity of acute pancreatitis in mice. *Gut Microbes*. 2020;11(6):1774-1789. doi:10.1080/19490976.2020.1770042
16. Zhao Z, Cheng W, Qu W, Shao G, Liu S. Antibiotic Alleviates Radiation-Induced Intestinal Injury by Remodeling Microbiota, Reducing Inflammation, and Inhibiting Fibrosis. *ACS Omega*. 2020;5(6):2967-2977. Published 2020 Feb 5. doi:10.1021/acsomega.9b03906
17. Shi Y, Kellingray L, Zhai Q, et al. Structural and Functional Alterations in the Microbial Community and Immunological Consequences in a Mouse Model of Antibiotic-Induced Dysbiosis. *Front Microbiol*. 2018;9: 1948. Published 2018 Aug 21. doi:10.3389/fmicb.2018.01948
18. Fachi JL, Felipe JS, Pral LP, et al. Butyrate Protects Mice from Clostridium Difficile-Induced Colitis through an HIF-1-Dependent Mechanism. *Cell Rep*. 2019;27(3):750-761.e7. doi: 10.1016/j.celrep.2019.03.054

Reviewer #2 (Remarks to the Author)

The authors addressed my concerns.

Response: Thank you very much for your comments. We appreciate your kind help and valuable suggestions.

Reviewer #3 (Remarks to the Author):

The authors performed quite amount of experiments for the revision of current manuscript, which improves the quality of their story. However, the reviewer still has concerns for their responses to reviewer#3's major points 3 and 4:

For response to point 3:

The authors claimed that it is the bacteria that survived the antibiotics treatment produces LPS. However, antibiotics should deplete most of gut microbiota and therefore, there should be several logs magnitude drop of bacterial amount and thus LPS. Comparison of LPS with and without antibiotics treatment should be performed to support their hypothesis. The authors also claimed that antibiotics damages gut epithelial integrity. Under the condition of leukaemia, antibiotics effects on gut epithelial integrity should be minimal. Comparison of gut epithelial integrity in leukaemia mice with or without antibiotics treatment should be performed.

Response: Thanks a lot for your positive comments. With your suggestion, we performed two additional experiments. Firstly, we performed new experiment to compare LPS concentration in mice with or without antibiotics treatment. As shown below, the blood LPS concentration was increased after using antibiotics, reached the peak at 48h, and remained in high level even at 144h. Several other researches have also reported that LPS concentration in blood is sharply increased after using antibiotics, as the bacteria lysis caused by antibiotics releases much bacteria components including LPS into blood [1,2]. These may explain the inconsistency between the amount of bacteria and LPS after using antibiotics. Secondly, we performed experiment to compare gut epithelial integrity in leukaemia mice with or without antibiotics treatment. The results showed that compared with mice without antibiotics, the mice treated with antibiotics were present with more intestinal damage, such as higher FITC-dextran permeability, decreased expression of cell-cell junction protecting proteins (as shown below). Though the effects on gut epithelial integrity by antibiotics might not be very high under the condition of leukaemia, antibiotics treatment could release large amount of LPS by killing and lysating bacteria and cause the increase of LPS into blood. We included this result in the new version (Line 608-611; Supplementary figure 7)

The result of LPS concentration in mice with or without antibiotics treatment

The result of intestinal permeability to FITC-dextran in AML mice with or without antibiotics treatment

The expressions of cell-cell junction protecting proteins in AML mice with or without antibiotics treatment

For response to point 4:

The authors still have NOT answer the concern raised by the reviewer: the differences between treated and control groups are so subtle (e.g., Fig.2C with less than 3% difference) that it affects the biological significance of current study.

Response: Thanks a lot for your valuable comment. The reason for the subtle differences between treated and control AML groups is that we sacrificed the AML mice too late and the leukaemia burden in both of the two groups is near the platform or saturation. Therefore, we repeated these experiments with sacrificing the AML mice at earlier time-point. The results are consistent with before, while the differences between two groups became more apparent. We replace with the new Fig.2B, 2C, 2D in the new version.

References

1. Van Langevelde P, Kwappenberg KM, Groeneveld PH, Mattie H, van Dissel JT. Antibiotic-induced lipopolysaccharide (LPS) release from *Salmonella typhi*: delay

between killing by ceftazidime and imipenem and release of LPS. *Antimicrob Agents Chemother.* 1998;42(4):739-743. doi:10.1128/AAC.42.4.739

2. Wang C, Schaefer L, Bian F, et al. Dysbiosis Modulates Ocular Surface Inflammatory Response to Liposaccharide. *Invest Ophthalmol Vis Sci.* 2019;60(13):4224-4233. doi:10.1167/iovs.19-27939

Reviewer #4 (Remarks to the Author):

The authors have sufficiently addressed the majority of the comments. There are a few remaining ones:

1. Point 2 regarding the blood collection. The blood metabolome is affected by the diet. In the methodology, it should be made clear that the blood samples were collected without fasting.

Response: With suggestion, we claimed that the blood samples were collected without fasting in the methodology of the new version (Line 117).

2. Point 3 why were these patients selected and the justification should be added into the Methodology.

Response: Thanks for your comment. We added the selected rationale and justification of these patients in the methodology of the new version (Line 169-170).

3. The gut microbiota from mice housed in the same cage tends to be similar compared to that of mice from different cages. As authors explained, the mice were housed based on the experiments/grouping, which could induce bias to the results. Linking with Point 9.6, in my opinion, animals should be singly housed.

Response: Thanks a lot for your valuable and strict suggestion for our animal experiments. To address your comment about cage-effect, with your suggestion, we performed new experiments to singly house the AML mice or normal mice, that is, one mouse was housed in one cage. We collected their feces to determine 16S rRNA sequencing. As shown below, no significant difference of the bacteria diversity was found among singly-housed mice in the same treatment group (AML or control), and

the statistical difference was found between AML and normal control groups with decreased diversity (Chao1 and Shannon indices). These results were similar to our previous group-housed results. We replaced this result in the new version (new Figure 2A, n=5, power=0.819 Chao1, power=0.883 Shannon).

4. Point 9.1 I understand the authors carried out the experiments with another 2 mice per group and added the data together to make a n=5. Is it correct?

Response: We carried out the experiments with another 5 mice per group, and replaced these data in the new Figure 2A.

REVIEWER COMMENTS

Reviewer #3 (Remarks to the Author):

The authors have performed experiments to address some of reviewers' #1 and #3 concerns.

However, some concerns still remain:

Response to reviewer #1 Point 1

The author is exaggerating the mouse leukemia model relevance to human AML. The MLL-AF9 translocation only represents a very small subtype of human adult leukemia whereas it is more common in pediatric leukemias (which is excluded in the current patient cohort). This model differs from other AML mouse models in several aspects, including leukemia stem cell populations, metabolism and signal transduction pathways. Therefore, many labs always incorporate at least two AML mouse models to validate the generalizability of their findings.

Response to reviewer #1 Point 3

For experiment performed in Fig2. F-H, the comparison should be AML+ FMT (fecal from normal mice) VS. AML+FMT (fecal from leukemic mice). The current experiment compared AML+FMT (fecal from normal mice) VS. AML+PBS, which is not a valid comparison.

Response to reviewer #1 Point 5

The author did not answer the question "if this is a driving etiologic factor or a secondary finding". The authors finding present that there are GI damage in AML mice. It could be a systemic/local immune response induced by AML or it could be a result of leukemia cell invasion to GI tract or it could be other mechanisms. Most likely, GI damage is secondary to AML. However, dysbiosis caused by GI damage may alter the niche for pre-leukemia cells and facilitate the leukemogenesis, which GI damage becomes the driving etiologic factor for AML. The current studies only present that GI damage is secondary to AML. Additionally, the current model used in this study is not suitable to investigate whether GI damage is a driving etiologic factor for AML.

Response to reviewer #1 Point 6b and reviewer #3 Point3

The authors' results shows that blood LPS level is elevated at first and then decreased after Abx treatment. At day 6 post Abx treatment, blood LPS level is comparable to that of control group. The leukemic transplantation was performed 10 days post first Abx treatment, at a point that the blood LPS level should be comparable or decreased to controls, which contradicts to the authors' claim that LPS is the contributor for AML progression.

Response to reviewer #1 Point 6d

The authors claim that in AML intestinal barrier was not in inflammatory hyperactivation state and that in AML gut bacteria and bacterial derived factors play protective roles. They also claim that LPS is one major driving factor for AML progression. LPS is a well-known powerful inflammatory inducer. Further, how could gut bacteria and bacterial derived factors (for example LPS, loss of butyrate) play protective roles under a compromised intestine integrity?

Reviewer #4 (Remarks to the Author):

The authors have addressed all the comments. One minor point, perhaps 'gender' should be replaced with 'sex' in Line 171.

Point by point responses to the reviewers' comments

Thanks for your valuable suggestions and remaining comments, which further have led to improvement of our manuscript. We have carefully addressed all your remaining concerns, with detailed point-by-point responses provided below.

REVIEWER COMMENTS

Reviewer #3 (Remarks to the Author):

The authors have performed experiments to address some of reviewers' #1 and #3 concerns. However, some concerns still remain.

Response: Thanks for your acknowledgements regarding our previous responses. Our detailed point-by-point responses to your remaining comments are provided below.

Response to reviewer #1 Point 1

The author is exaggerating the mouse leukemia model relevance to human AML. The MLL-AF9 translocation only represents a very small subtype of human adult leukemia whereas it is more common in pediatric leukemia (which is excluded in the current patient cohort). This model differs from other AML mouse models in several aspects, including leukemia stem cell populations, metabolism and signal transduction pathways. Therefore, many labs always incorporate at least two AML mouse models to validate the generalizability of their findings.

Response: Thank you very much for your constructive comment. The editor also raises the similar comment and suggests us to add additional murine models to support our proposal that bacterial diversity decrease in AML. In the revised manuscript, we have established another two commonly used types of AML murine models, AML1-ETO murine model [1-2] and C1498 murine model [3-4], to validate the generalizability of our findings. Then, we applied 16s rRNA sequencing to determine the gut microbiota diversity of the two additional AML murine models. Consistent with the results from MLL-AF9, the results from the two newly added AML mouse models showed that the diversity of gut microbiota significantly decreased in AML. We have added these results (Supplementary figure 2c-d) as well as the corresponding statements in the revision

(Line 132-134, 167-170 and Line 418-420).

Following your comment, we further used the AML1-ETO murine model to perform several key *in vivo* functional experiments to deeply investigate the role of gut microbiota in AML. Firstly, we used antibiotics to treat AML1-ETO mice by gavage, and the results showed that the treatment of antibiotics accelerated the progress of AML. Secondly, we administered butyrate to AML1-ETO mice, which delayed the progression of AML. Finally, we injected LPS into AML1-ETO mice through the tail vein, and the results showed that LPS significantly promoted the progression of AML. Overall, all these functional results using AML1-ETO murine model were consistent with those from MLL-AF9 murine model. We have added these results (see the newly added Supplementary figure 8) as well as the corresponding statements in the revision (line 659-663).

Response to reviewer #1 Point 3

For experiment performed in Fig2. F-H, the comparison should be AML+ FMT (fecal from normal mice) VS. AML+FMT (fecal from leukemic mice). The current experiment compared AML+FMT (fecal from normal mice) VS. AML+PBS, which is not a valid comparison.

Response: Thanks for pointing this out. As you suggested, we performed additional experiment for the comparison of AML+FMT (fecal from normal mice) VS. AML+FMT (fecal from leukemic mice). As expected, compared with FMT with the feces from normal mice, FMT with the feces from AML mice can accelerate the progression of AML *in vivo*. We have added these new results (Supplementary figure 9) and the corresponding statements in the revision (line 491-492).

Response to reviewer #1 Point 5

The author did not answer the question “if this is a driving etiologic factor or a secondary finding”. The authors finding present that there are GI damage in AML mice. It could be a systemic/local immune response induced by AML or it could be a result of leukemia cell invasion to GI tract or it could be other mechanisms. Most likely, GI damage is secondary to AML. However, dysbiosis caused by GI damage may alter the niche for pre-leukemia cells and facilitate the leukemogenesis, which GI damage becomes the driving etiologic factor for AML. The current studies only present that GI

damage is secondary to AML. Additionally, the current model used in this study is not suitable to investigate whether GI damage is a driving etiologic factor for AML.

Response: Thank you for the acknowledgement with our proposal that GI damage is secondary to AML. Based on your comment, we note that it is indeed of great significance to clarify the complex causal relationship between GI damage and AML. Actually, our previous results showed that antibiotics treatment caused more GI damage (see the three Figures below, which we have extracted from our previous responses to Reviewer #3 for better illustration here), which further accelerated the progression of AML and shortened AML mice survival (Figure 2C, D, E in the previous manuscript). In addition, the progression of AML was alleviated after repairing GI damage with gavaging butyrate (Figure 5 and 6 in the previous manuscript). All these previous results may to some extent support that GI damage is a promoting factor for AML.

However, as you mentioned, additional evidence should also be provided to further valid this point. We, therefore, additionally performed new experiment. Briefly, we damaged intestinal barrier using low-dose and short-term dextran sulfate sodium salt (DSS) (1.9%) by gavaging mice for 3 days, followed by injecting AML cells to establish AML murine model. The HE staining showed that DSS treatment caused slight GI damage. Results showed that the AML progression in DSS treatment mice accelerated than that in control AML group, which can further validate that GI damage promotes AML progression. We have added this new result (supplementary figure 10) as well as the corresponding statements in the revision (Line 202-204, Line 576-580).

Specially, our study can only prove that gut microbiota dysbiosis and GI damage promote the progression of existing AML, but there is no evidence to prove whether GI damage can cause the occurrence of AML. Whether the gut microbiota dysbiosis and GI damage can alter the niche for pre-leukemia cells and facilitate the leukemogenesis is what we need to study further. We have added the statement about this point in the new version (Line 758-761).

The result of intestinal permeability to FITC-dextran in AML mice with or without antibiotics treatment

The expressions of cell-cell junction protecting proteins in AML mice with or without antibiotics treatment

Response to reviewer #1 Point 6b and reviewer #3 Point3

The authors' results shows that blood LPS level is elevated at first and then decreased after Abx treatment. At day 6 post Abx treatment, blood LPS level is comparable to that

of control group. The leukemic transplantation was performed 10 days post first Abx treatment, at a point that the blood LPS level should be comparable or decreased to controls, which contradicts to the authors' claim that LPS is the contributor for AML progression.

Response: Thanks a lot. The editor also raises the similar comment. We apologize very much for bringing you such confusion which may be presumably due to the insufficient data and the explanation. In order to provide stronger evidence to support our conclusion, we repeated this experiment with the same design shown in Figure 2B, in which we continuously determined the murine serum LPS concentration throughout the whole experiment process. As shown in the newly added supplementary figure 7, the results demonstrated that the blood level of LPS was increased temporarily after using ABX which may be presumably due to bacterial lysis, then decreased slightly with the continued use of antibiotics, and increased again after stopping antibiotics possibly because of the damage of the intestinal barrier. In addition, it should be noted that, the blood levels of LPS in ABX-gavaging AML mice were significantly higher than those of control AML mice during the whole experiment. And LPS was shown to promote AML cell proliferation and inhibit AML cell apoptosis *in vitro*, and accelerate AML progression and shorten AML mice survival *in vivo* (Figure 7). All these results provide a better support for the role of LPS in AML progression. (Line 632-637).

Response to reviewer #1 Point 6d

The authors claim that in AML intestinal barrier was not in inflammatory hyperactivation state and that in AML gut bacteria and bacterial derived factors play protective roles. They also claim that LPS is one major driving factor for AML progression. LPS is a well-known powerful inflammatory inducer. Further, how could gut bacteria and bacterial derived factors (for example LPS, loss of butyrate) play protective roles under a compromised intestine integrity?

Response: Thanks a lot. Reading through your comment, we speculate that our insufficient interpretations may bring you such confusion. We are sorry for making you confused due to our vague explanation. In brief, our results showed that AML caused a compromised intestine integrity (Figure 6), disordered bacteria diversity (Figure 1, Figure 2A and supplementary figure 2c-d) and the change of bacterial derived factors

(loss of butyrate and higher blood LPS) (Figure 4 and 7). Under the compromised intestine integrity, the abnormal gut bacteria and the change of bacterial derived factors (loss of butyrate and higher blood LPS) play promoting roles in AML progression, while the normal gut bacteria and bacterial derived factors (normal levels of butyrate) by FMT from healthy people or normal mice play protective roles through repairing intestine integrity (Figure 3 and supplementary figure 9). Moreover, supplementation of bacterial derived factor butyrate delayed AML progression by repairing intestine integrity (Figure 5 and 6). Overall, the disordered gut bacteria and abnormal bacterial derived factors (elevated LPS and loss of butyrate) are risk factors for the progression of AML, and do not play a protective role. Following your comment, we have added these explanations in the Discussion to better explain the main message that we want to convey. (Line 758-761).

Reviewer #4 (Remarks to the Author):

The authors have addressed all the comments. One minor point, perhaps 'gender' should be replaced with 'sex' in Line 171.

Response: With suggestion, we replaced 'gender' with 'sex' in the new version (Line 177).

Reference

1. DeKelver RC, Yan M, Ahn EY, Shia WJ, Speck NA, Zhang DE. Attenuation of AML1-ETO cellular dysregulation correlates with increased leukemogenic potential. *Blood*. 2013;121(18):3714-3717. doi:10.1182/blood-2012-11-465641
2. Zhou B, Ye H, Xing C, et al. Targeting miR-193a-AML1-ETO- β -catenin axis by melatonin suppresses the self-renewal of leukaemia stem cells in leukaemia with t(8;21) translocation. *J Cell Mol Med*. 2019;23(8):5246-5258. doi:10.1111/jcmm.14399
3. Zhang L, Gajewski TF, Kline J. PD-1/PD-L1 interactions inhibit antitumor immune responses in a murine acute myeloid leukemia model. *Blood*. 2009;114(8):1545-1552. doi:10.1182/blood-2009-03-206672
4. Ruzicka M, Koenig LM, Formisano S, et al. RIG-I-based immunotherapy enhances survival in preclinical AML models and sensitizes AML cells to checkpoint

blockade. *Leukemia*. 2020;34(4):1017-1026. doi:10.1038/s41375-019-0639-x

REVIEWER COMMENTS

Reviewer #3 (Remarks to the Author):

The authors have done lots of works to improve their manuscript. The reviewer just has one more suggestion: in the experiments with ABX treatments, ABX was given before leukemic transplantation and stopped right after. If ABX was given to leukemic mice during the whole time of disease progression (to maintain a LPS level), is it able to repress leukemia? Because leukemia patients need to take ABX to prevent infection, especially during chemotherapy, this experiment will make the study more clinical relevant.

Point by point responses to the reviewers' comments

Thanks for your valuable suggestions and remaining comments, which we have carefully addressed. The detailed point-by-point responses are provided below.

REVIEWERS' COMMENTS

Reviewer #3 (Remarks to the Author):

The authors have done lots of works to improve their manuscript. The reviewer just has one more suggestion: in the experiments with ABX treatments, ABX was given before leukemic transplantation and stopped right after. If ABX was given to leukemic mice during the whole time of disease progression (to maintain a LPS level), is it able to repress leukemia? Because leukemia patients need to take ABX to prevent infection, especially during chemotherapy, this experiment will make the study more clinical relevant.

Response: Thanks for your acknowledgements and the remaining comments you suggested. At present study, we mainly focus on the role of intestinal microbes in AML. Following the previous literatures [1-2], here the main purpose of using ABX is to destroy intestinal microbes. In addition, the direct effect of ABX on tumor cells has also been reported in many studies [3-6]. Therefore, ABX was preferred to be given before leukemic transplantation and stopped right after, so as to exclude the direct impact of ABX on leukemia in vivo.

However, as you mentioned, the effect of ABX treatment to prevent infection in leukemia patients should not be ignored, especially during chemotherapy. It is indeed of great significance to explore the effect and mechanism of ABX treatment for AML therapy. However, the ABX application in clinical practice may be different from that in our study in the dosage, drug delivery route (oral, intravenous, hypodermal or others) as well as the administration together with chemotherapy. Leukemia can be influenced by ABX itself, ABX-induced microbiota dysbiosis and ABX combined with chemotherapy, even the complicated interaction among them. Your valuable suggestion here provides a novel direction for future study, and we have incorporated your comments in the revised manuscript in the Discussion (line 476-480).

References

1. Zhu Z, Huang J, Li X, et al. Gut microbiota regulate tumor metastasis via circRNA/miRNA networks. *Gut Microbes.* 2020;12(1):1788891. doi:10.1080/19490976.2020.1788891
2. Vicentini FA, Keenan CM, Wallace LE, et al. Intestinal microbiota shapes gut physiology and regulates enteric neurons and glia. *Microbiome.* 2021;9(1):210. Published 2021 Oct 26. doi:10.1186/s40168-021-01165-z
3. Friedman GD, Oestreicher N, Chan J, Quesenberry CP Jr, Udaltsova N, Habel LA. Antibiotics and risk of breast cancer: up to 9 years of follow-up of 2.1 million women. *Cancer Epidemiol Biomarkers Prev.* 2006;15(11):2102-2106. doi: 10.1158/1055-9965.EPI-06-0401
4. Cao Y, Wu K, Mehta R, et al. Long-term use of antibiotics and risk of colorectal adenoma. *Gut.* 2018;67(4):672-678. doi:10.1136/gutjnl-2016-313413
5. Wang Y. Effects of salinomycin on cancer stem cell in human lung adenocarcinoma A549 cells. *Med Chem.* 2011;7(2):106-111. doi:10.2174/157340611794859307
6. Mondal ER, Das SK, Mukherjee P. Comparative evaluation of antiproliferative activity and induction of apoptosis by some fluoroquinolones with a human non-small cell lung cancer cell line in culture. *Asian Pac J Cancer Prev.* 2004;5(2):196-204.